# Statistical Estimation in the Spiked Tensor Model via the Quantum Approximate Optimization Algorithm

**Leo Zhou**
Walter Burke Institute for Theoretical Physics and the IQIM
California Institute of Technology, Pasadena, CA 91125
Electrical and Computer Engineering Department
University of California, Los Angeles, CA 90095
leoxzhou@ucla.edu

**Joao Basso**
Department of Mathematics
University of California, Berkeley, CA 94720
Quantum Artificial Intelligence Lab (QuAIL)
NASA Ames Research Center, Moffett Field, CA 94035
Research Institute for Advanced Computer Science (RIACS)
USRA, Mountain View, CA 94043
joao.basso@berkeley.edu

**Song Mei**
Department of Statistics and Department of EECS
University of California, Berkeley, CA 94720
songmei@berkeley.edu

## Abstract

The quantum approximate optimization algorithm (QAOA) is a general-purpose algorithm for combinatorial optimization that has been a promising avenue for near-term quantum advantage. In this paper, we analyze the performance of the QAOA on the spiked tensor model, a statistical estimation problem that exhibits a large computational-statistical gap classically. We prove that the weak recovery threshold of 1-step QAOA matches that of 1-step tensor power iteration. Additional heuristic calculations suggest that the weak recovery threshold of $p$-step QAOA matches that of $p$-step tensor power iteration when $p$ is a fixed constant. This further implies that multi-step QAOA with tensor unfolding could achieve, but not surpass, the asymptotic classical computation threshold $\Theta(n^{(q-2)/4})$ for spiked $q$-tensors. Meanwhile, we characterize the asymptotic overlap distribution for $p$-step QAOA, discovering an intriguing sine-Gaussian law verified through simulations. For some $p$ and $q$, the QAOA has an effective recovery threshold that is a constant factor better than tensor power iteration. Of independent interest, our proof techniques employ the Fourier transform to handle difficult combinatorial sums, a novel approach differing from prior QAOA analyses on spin-glass models without planted structure.

## 1 Introduction

We study statistical estimation in the spiked tensor model, where we observe a $q$-tensor $\boldsymbol{Y} \in \mathbb{R}^{n^q}$ in $n^q$ dimensions given by

$$\boldsymbol{Y} = (\lambda_n/n^{q/2}) \cdot \boldsymbol{u}^{\otimes q} + (1/\sqrt{n}) \cdot \boldsymbol{W} \in \mathbb{R}^{n^q}. \tag{1.1}$$

38th Conference on Neural Information Processing Systems (NeurIPS 2024).

Here $\boldsymbol{u} \sim \mathrm{Unif}(\{+1, -1\}^n)$ is some hidden signal,[1] and $\boldsymbol{W} \in (\mathbb{R}^n)^{\otimes q}$ is a noise tensor whose entries are i.i.d. standard Gaussian $\mathcal{N}(0, 1)$. The parameter $\lambda_n > 0$ is the *signal-to-noise ratio* (SNR). The goal is to estimate $\boldsymbol{u}$ given only access to $\boldsymbol{Y}$. That is, we seek an estimator $\hat{\boldsymbol{u}} \colon (\mathbb{R}^n)^{\otimes q} \to \mathbb{S}^{n-1}(\sqrt{n})$ achieving nontrivial overlap with the signal:

$$\liminf_{n \to \infty} \mathbb{E}[\langle \hat{\boldsymbol{u}}(\boldsymbol{Y}), \boldsymbol{u} \rangle^2 / n^2] > 0. \tag{1.2}$$

This task is known as *weak recovery* in the spiked tensor model.

The spiked tensor model is a famous problem because it exhibits a huge computational-statistical gap, referring to regimes of SNR where the statistical estimation problem is information-theoretically solvable, but no efficient algorithm has been found. For example, it is known that the Bayes optimal estimator achieves non-trivial overlap with the signal $\boldsymbol{u}$ when $\lambda_n > \lambda_{\mathrm{IT}}$ for some constant threshold $\lambda_{\mathrm{IT}} = \Theta(1)$, whereas the problem is information-theoretically impossible when $\lambda_n \leq \lambda_{\mathrm{IT}}$ [1]. Furthermore, the maximum likelihood estimator also achieves non-trivial overlap with the signal when $\lambda_n > \lambda_{\mathrm{MLE}}$ for some $\lambda_{\mathrm{MLE}} = \Theta(1)$. However, the best-known polynomial-time classical algorithms for computing a non-trivial estimator require a much higher SNR of $\lambda_n = \Theta(n^{(q-2)/4})$. These include tensor power iteration, gradient descent, approximate message passing, and spectral methods with tensor unfolding [2–12]. Indeed, assuming the secret leakage planted clique conjecture, Ref. [13] proves an $\Omega(n^{(q-2)/4})$ lower bound on the SNR needed by any polynomial-time classical algorithm. See Fig. 1 for an illustration of the different SNR thresholds and Section 2.1 for more background.

On the other hand, quantum algorithms are widely believed to have computational advantages over classical algorithms for many problem classes. In particular, we focus on the Quantum Approximate Optimization Algorithm (QAOA) [14], a general-purpose quantum optimization algorithm that can be applied to optimize any objective function on bit-strings. The QAOA has received an enormous amount of attention in the quantum computing community for several reasons. First, the QAOA is simple and allows efficient implementation on near-term quantum hardware with many applications [15–19]. Additionally, the QAOA is computationally universal [20], and its generalization can realize other powerful algorithms such as the quantum singular value transform [21]. Under common complexity-theoretic assumptions, no classical device can efficiently simulate the output distribution of the QAOA even at shallow depth [22, 23]. Furthermore, the QAOA is guaranteed to find optimal solutions when its number of steps (or depth) diverges [14]. Nevertheless, analyzing the asymptotic performance of QAOA remains challenging: classical simulation of the algorithm is limited to small problem dimension $n$, and analytical computations are often highly non-trivial [24–28]. Given the enormous $\Omega(n^{(q-2)/4})$ computational-statistical gap in the spiked tensor model (compared to e.g., the constant factor gap in spin-glass optimization [29]), it is an interesting open question whether the QAOA, as a realistic quantum algorithm with asymptotic convergence guarantees, can provide any computational advantages.

In this work, we investigate the performance of QAOA for the spiked tensor model. In particular, we choose the log-likelihood objective of spiked tensor $C(\boldsymbol{z}) = \langle \boldsymbol{Y}, \boldsymbol{z}^{\otimes q} \rangle / n^{(q-2)/2}$. Its maximizer, the maximum likelihood estimator, achieves non-trivial overlap with the signal whenever $\lambda_n > \lambda_{\mathrm{MLE}} = \Theta(1)$. While the infinite-step QAOA could compute the maximizer, we are interested in the performance of QAOA when the depth is polynomial in the problem size, and hope that it can surpass the $\Theta(n^{(q-2)/4})$ classical threshold. Although some limitations of the QAOA are known for certain random optimization problems in the low-depth regime [26, 30–32], these negative results do not apply to the spiked tensor model since they rely on either sparse connectivity or concentration, both of which are absent in the current setting. Here, as a first attempt to bridge the gap in understanding how well a popular quantum algorithm may perform on a classically hard statistical estimation problem, we study the asymptotic behavior of the QAOA on the spiked tensor model in the constant-depth regime, where we are able to obtain rigorous and analytical results.

**Our contribution.**  In this paper, we analyze the signal-to-noise ratio threshold of $p$-step QAOA for weak recovery in the spiked tensor model, in the regime of fixed $p$ and $n$ approaching infinity. For $p = 1$, we prove the weak recovery threshold is $\lambda_n = \Theta(n^{(q-1)/2})$, matching that of 1-step tensor power iteration. For $p > 1$, heuristic calculations suggest the threshold is $\lambda_n = \Theta(n^{(q-2+\varepsilon_p)/2})$

---

[1]Another commonly studied prior is the uniform distribution over the $n$-sphere, $\mathrm{Unif}(\mathbb{S}^{n-1}(\sqrt{n}))$. In this work, we choose the Rademacher prior for the convenience of the QAOA analysis.

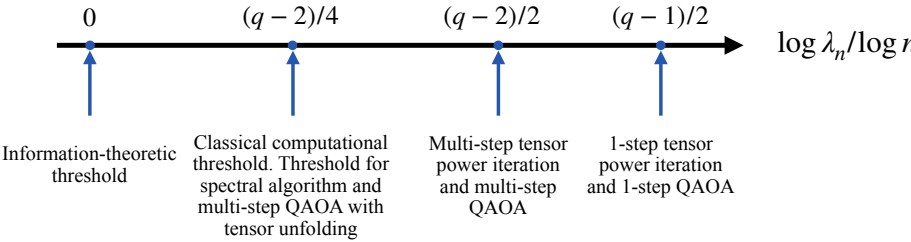

Figure 1: Different thresholds for the spiked tensor model.

where $\varepsilon_p = (q-2)/[(q-1)^p - 1]$, again matching $p$-step tensor power iteration. Additionally, given an initialization vector with $n^c/n$ correlation to the signal for $1/2 < c < 1$, we prove the weak recovery threshold for 1-step QAOA is $\lambda_n = \Theta(n^{(1-c)(q-1)})$, identical to 1-step tensor power iteration. These results indicate that constant-step QAOA has the same asymptotic recovery threshold as tensor power iteration in the spiked tensor model. Meanwhile, further heuristic analysis suggests that QAOA with tensor unfolding could achieve the classical computation threshold $\Theta(n^{(q-2)/4})$.

Furthermore, we derive the asymptotic distribution of the overlap for $p$-step QAOA, revealing an intriguing sine-Gaussian law distinct from $p$-step tensor power iteration. Analyzing the second moment, we see that, for certain $(p, q)$ pairs, the QAOA effectively has a recovery threshold that is a constant factor better than tensor power iteration. Since there are classical algorithms that achieve better recovery thresholds than power iteration, it remains an interesting open question whether quantum advantage over the state-of-the-art classical algorithms may be obtained at larger QAOA depths that grow with system size. To our current knowledge, our work is the first to obtain analytical results using the QAOA for a statistical inference problem.

The proof of the sine-Gaussian distribution adopted novel techniques, including using discrete Fourier transforms and the central limit theorem to handle combinatorial summations. The Fourier transform technique also allows us to replace nonlinear polynomials in the exponents with dual variables, leaving linear exponents that become easy in combinatorial sums. These techniques are of independent interest and could be useful for analyzing the QAOA in other models.

## 2   Background and related work

### 2.1   Spiked tensor model and prior algorithms

The spiked tensor model (1.1) was first introduced as a statistical model for tensor principal component analysis in [2], where it was studied with a spherical prior $u \in \mathbb{S}^{n-1}(\sqrt{n})$. The information-theoretic threshold for weak recovery under this model with the spherical prior [3, 7, 8] and the Rademacher prior $u \in \{\pm 1\}^n$ [1] are both $\lambda_n = \Theta(1)$.

**Tensor power iteration.**    A well-studied classical algorithm for the spiked tensor model is tensor power iteration [2, 10, 33]. Starting from a uniform random initialization $\hat{u}_0 \sim \mathrm{Unif}(\mathbb{S}^{n-1})$, the $k$-th iteration is given by $\hat{u}_k$, where

$$\hat{u}_k = \sqrt{n} Y[\hat{u}_{k-1}^{\otimes(q-1)}]/\big\|Y[\hat{u}_{k-1}^{\otimes(q-1)}]\big\|_2, \quad k \geq 1, \quad \hat{u}_0 \sim \mathrm{Unif}(\mathbb{S}^{n-1}). \tag{2.1}$$

Here, $Y[\hat{u}^{\otimes(q-1)}] \in \mathbb{R}^n$ denotes contracting the order-$q$ tensor $Y \in \mathbb{R}^{n^q}$ with the order-$(q-1)$ tensor $\hat{u}^{\otimes(q-1)} \in \mathbb{R}^{n^{q-1}}$. It is shown that with $(\log n)$ iterations, weak recovery is possible if the SNR satisfies $\lambda_n = \Omega(n^{(q-2)/2}/\operatorname{polylog}(n))$ [10, 33]. However, tensor power iteration does not match the best-known classical algorithms. Furthermore, we remark that rounding the tensor power iteration to $\operatorname{sign}(\hat{u}_k) \in \{\pm 1\}^n$ does not give a better threshold.

**Other classical algorithms and related results.**    [2] showed that the tensor power iteration and approximate message passing algorithms with random initialization can recover the signal provided $\lambda_n = \Omega(n^{(q-1)/2})$. This SNR threshold was later improved to $\lambda_n = \Omega(n^{(q-2)/2})$ by [3, 10, 33] for these same methods. The same threshold $\lambda_n = \Omega(n^{(q-2)/2})$ could also be achieved by gradient

descent and Langevin dynamics as proved in [9]. On maximum likelihood estimation for the spiked tensor model with a spherical prior, [5, 6] studied the loss landscape, providing intuition that it contains many saddle points and local minima near the equator, but no bad critical points off the equator.

The best currently known polynomial-time algorithms can achieve a sharp threshold of $\lambda_n = \Omega(n^{(q-2)/4})$. These include spectral methods with tensor unfolding [2, 11], sum-of-squares algorithms [34–36], sophisticated iteration algorithms [37–39], and gradient descent on the smoothed landscape [40, 41].

Another line of research has attempted to prove computational lower bounds in restricted computational models, including low-degree polynomials and statistical query algorithms [42, 43]. Under the secreted leakage planted clique conjecture, [13] proved that any classical polynomial-time algorithm requires $\lambda_n = \Omega(n^{(q-2)/4})$ for weak recovery of the signal.

**A quantum algorithm by Ref. [44].** To the best of our knowledge, the only prior quantum algorithm proposed for the spiked tensor model with provable guarantees is by Hastings in Ref. [44]. Hastings' algorithm is based on a spectral method for a Hamiltonian on $M$ bosons over $n$ modes, living in a Hilbert space of dimension $n^M$, where $M \gg [n^{(q-2)/4}/\lambda_n]^{4/(q-2)} \times \text{polylog}(n)$. Finding the dominant eigenvector of this Hamiltonian allows for weak recovery in the regime where $\lambda_n = \Theta(n^{(q-2)/4})$. In this regime, where $M = \Omega(\text{polylog } n)$, the standard classical matrix power iteration algorithm can extract the dominant eigenvector and recover the signal in $\tilde{O}(n^M)$ time. For the proposed quantum algorithm, Ref. [44] uses a combination of quantum phase estimation, amplitude amplification, and clever state initialization to recover the signal in $\tilde{O}(n^{M/4})$ time, achieving a quartic speedup. (A few months after our paper appeared online, a related work [45] emerged, simplifying Hastings' algorithm and generalizing it to another planted inference problem.)

We remark that Hastings' algorithm runs in superpolynomial time $n^{\Omega(\text{polylog } n)}$ and does not improve over the asymptotic computational threshold in SNR for recovery (although a constant factor improvement is possible). For comparison, the classical spectral method based on tensor unfolding [2, 11] achieves recovery when $\lambda_n > n^{(q-2)/4}$ in polynomial time $O(\text{poly}(n^q))$. In this work, we study the QAOA in the constant-step regime, where the gate complexity grows only linearly in the problem size $O(n^q)$.

## 2.2  Quantum approximate optimization algorithm

The quantum approximate optimization algorithm (QAOA) was introduced by [14] as a quantum algorithm for finding approximate solutions to combinatorial optimization problems. The QAOA can be applied to optimize any cost function on bit-strings, $C : \{\pm 1\}^n \to \mathbb{R}$. In the spiked tensor model, we consider optimizing the log-likelihood function given by

$$\hat{\boldsymbol{u}}_{\text{MLE}} = \arg\max_{\boldsymbol{\sigma} \in \{\pm 1\}^n} \left\{ C(\boldsymbol{\sigma}) = \langle \boldsymbol{Y}, \boldsymbol{\sigma}^{\otimes q} \rangle / n^{(q-2)/2} \right\}. \tag{2.2}$$

The maximum likelihood estimator $\hat{\boldsymbol{u}}_{\text{MLE}}$ achieves non-trivial correlation with the signal when $\lambda_n > \lambda_{\text{MLE}}$ for some constant $\lambda_{\text{MLE}} = \Theta(1)$. However, classical algorithms cannot efficiently compute the MLE unless $\lambda_n = \Omega(n^{(q-2)/4})$ [13]. This paper investigates whether QAOA could compute $\hat{\boldsymbol{u}}_{\text{MLE}}$, or an approximate estimator, for smaller values of $\lambda_n$.

The inputs to the QAOA algorithm are a cost function $C : \{\pm 1\}^n \to \mathbb{R}$ and parameter vectors $\boldsymbol{\gamma}, \boldsymbol{\beta} \in \mathbb{R}^p$. The initial QAOA state $|s\rangle = 2^{-n/2} \sum_{\boldsymbol{z}} |\boldsymbol{z}\rangle$ is the rescaled all-one vector $2^{-n/2} \mathbf{1}_{2^n} \in \mathbb{C}^{2^n}$, assigning equal probability to measuring each possible bit-string upon quantum measurement. See Appendix A.1 for a review of quantum computing terminology, where we also define the Pauli operators $\{X_k, Y_k, Z_k\}_{k=1}^n$ acting on the $k$-th qubit. The cost function $C$ associates with a $2^n \times 2^n$ diagonal matrix, where the $|\boldsymbol{z}|$'th diagonal gives $C(\boldsymbol{z})$. For the spiked tensor model with cost function $C(\boldsymbol{z}) = \langle \boldsymbol{Y}, \boldsymbol{z}^{\otimes q} \rangle / n^{(q-2)/2}$, this matrix is $C = \sum_{j_1,\ldots,j_q=1}^n Y_{j_1 \cdots j_q} Z_1 \cdots Z_q / n^{(q-2)/2} \in \mathbb{C}^{2^n \times 2^n}$. Letting $B = \sum_{j=1}^n X_j \in \mathbb{C}^{2^n \times 2^n}$, for any parameter $(\gamma, \beta)$, the unitary matrices $e^{-i\gamma C}, e^{-i\gamma B} \in \mathbb{C}^{2^n \times 2^n}$ are matrix exponents of $-i\gamma C$ and $-i\gamma B$. Given $\boldsymbol{\gamma}, \boldsymbol{\beta} \in \mathbb{R}^p$, the $p$-step QAOA state is

$$|\boldsymbol{\gamma}, \boldsymbol{\beta}\rangle = e^{-i\beta_p B} e^{-i\gamma_p C} \cdots e^{-i\beta_1 B} e^{-i\gamma_1 C} |s\rangle \in \mathbb{C}^{2^n}. \tag{2.3}$$

One can verify $|\gamma, \beta\rangle$ is a unit vector since $|s\rangle \in \mathbb{C}^{2^n}$ is unit and $e^{-i\beta_k B} \in \mathbb{C}^{2^n \times 2^n}$ and $e^{-i\gamma_k C} \in \mathbb{C}^{2^n \times 2^n}$ are unitary matrices. After preparing the quantum state $|\gamma, \beta\rangle$, QAOA samples a bit string $z \sim |\gamma, \beta\rangle$ in $\{\pm 1\}^n$ by quantum measurement. In our main results, we will analyze the distribution of the overlap $\mathcal{R}_{\text{QAOA}}$ of this quantum measurement $z$ with respect to the signal $u$:

$$\mathcal{R}_{\text{QAOA}} \equiv z^\top u / n = \frac{1}{n} \sum_{i=1}^{n} z_i u_i \in [-1, 1]. \tag{2.4}$$

For any function $f(z) = \sum_{k=0}^{n} \sum_{(j_1, \cdots, j_k)} \hat{f}_{j_1 \cdots j_k} z_{j_1} \cdots z_{j_k}$, its expectation under the QAOA state $|\gamma, \beta\rangle$ is given by $\langle \gamma, \beta | f(Z) | \gamma, \beta \rangle$, where $\langle \gamma, \beta | \in \mathbb{C}^{1 \times 2^n}$ is the conjugate transpose of $|\gamma, \beta\rangle \in \mathbb{C}^{2^n \times 1}$, and $f(Z) = \sum_{k=0}^{n} \sum_{(j_1, \cdots, j_k)} \hat{f}_{j_1 \cdots j_k} Z_{j_1} \cdots Z_{j_k} \in \mathbb{R}^{2^n \times 2^n}$ for Pauli-Z matrices $Z_j$. To simplify the notations, we denote $\langle \cdot \rangle_{\gamma, \beta}$ by the expectation with the quantum measurement from $|\gamma, \beta\rangle$, so that

$$\langle f(Z) \rangle_{\gamma, \beta} = \langle \gamma, \beta | f(Z) | \gamma, \beta \rangle. \tag{2.5}$$

In the main theorems of this paper, we will focus on the second moment of the overlap of QAOA, denoted as $\langle \mathcal{R}_{\text{QAOA}}^2 \rangle_{\gamma, \beta} = \langle \gamma, \beta | \hat{\mathcal{R}}^2 | \gamma, \beta \rangle$, where $\hat{\mathcal{R}} \equiv \frac{1}{n} \sum_{i=1}^{n} u_i Z_i$. We defer further related literature on theoretical analyses of the QAOA to Appendix A.2.

In terms of experimental realizations, the QAOA has been implemented in quantum computing platforms such as trapped ions [16, 19], superconducting qubits [17], and neutral atoms [18], for optimization problems with up to 179 bit variables. Implementing the QAOA for the spiked tensor model, however, poses additional challenges due to the all-to-all connectivity in its cost function, leading to a higher overhead in the number of quantum gates and circuit compilation costs. Currently, the largest experimental implementations for problems with dense connectivity include 17-bit Sherrington-Kirkpatrick spin-glass models on superconducting qubits [17], and an 18-bit LABS problem on trapped-ion quantum processors [19]. We expect larger problems can be implemented as quantum hardware matures, but quantum error-correction is likely necessary to observe any quantum advantage at scale [46].

## 3 Main results

### 3.1 Weak recovery threshold and overlap distribution for 1-step QAOA

We first consider the general 1-step QAOA for weak recovery in the spiked tensor model. Consider the spiked tensor model $Y$ (1.1) with planted signal $u \sim \text{Unif}(\{\pm 1\}^n)$ and the 1-step QAOA quantum state $|\gamma_n, \beta_n\rangle = e^{-i\beta_n B} e^{-i\gamma_n C} |s\rangle$ (see Section 2.2) with parameters $(\gamma_n, \beta_n) \in \mathbb{R}_{>0} \times [0, 2\pi]$. The quantum state $|\gamma_n, \beta_n\rangle$ depends randomly on $Y$ through $C(\sigma) = \langle Y, \sigma^{\otimes q} \rangle / n^{(q-2)/2}$. Our main results characterize the distribution of the overlap $\mathcal{R}_{\text{QAOA}} = \hat{u}^\top u / n$ between a sample $\hat{u} \sim |\gamma_n, \beta_n\rangle$ and the signal vector $u$.

**Theorem 1** (Weak recovery threshold and overlap distribution for 1-step QAOA). *Consider the spiked tensor model (1.1) and the 1-step QAOA overlap as defined above. Then the following hold.*

(a) *Take any sequence of $\{\gamma_n\}_{n \geq 1} \subseteq \mathbb{R}$, $\{\beta_n\}_{n \geq 1} \subseteq [0, 2\pi]$, and any sequence of $\{\lambda_n\}_{n \geq 1} \subseteq [0, \infty)$ with $\lim_{n \to \infty} \lambda_n / n^{(q-1)/2} = 0$. We have*

$$\lim_{n \to \infty} \mathbb{E}_Y[\langle \mathcal{R}_{\text{QAOA}}^2 \rangle_{\gamma_n, \beta_n}] = 0. \tag{3.1}$$

(b) *Take any sequence of $\{\gamma_n\}_{n \geq 1}$, $\{\beta_n\}_{n \geq 1}$, and $\{\lambda_n\}_{n \geq 1}$ which satisfies*

$$\lim_{n \to \infty} (\gamma_n, \beta_n, \lambda_n / n^{(q-1)/2}) = (\gamma, \beta, \Lambda). \tag{3.2}$$

*Then, over the randomness of $Y$ and the quantum measurement, the overlap $\mathcal{R}_{\text{QAOA}}$ of the 1-step QAOA converges in distribution to a sine-Gaussian law as*

$$\mathcal{R}_{\text{QAOA}} \xrightarrow{d} e^{-2q\gamma^2} \sin(2\beta) \sin(2q\Lambda\gamma G^{q-1}), \quad \text{where } G \sim \mathcal{N}(0, 1). \tag{3.3}$$

(c) *As a corollary of (b), under the asymptotic limit of (3.2) with $\Lambda > 0$, $\gamma > 0$, and $\beta \notin \{k\pi/2 : k \in \mathbb{Z}\}$, we have*

$$\lim_{n \to \infty} \mathbb{E}_Y[\langle \mathcal{R}_{\text{QAOA}}^2 \rangle_{\gamma_n, \beta_n}] > 0. \tag{3.4}$$

The full proof of Theorem 1 is contained in Appendix C.

**Remark 3.1** (Weak recovery threshold). Theorem 1(c) implies that when $\lambda_n = \Theta(n^{(q-1)/2})$, the overlap will be non-zero with non-trivial probability over both the random draw of the tensor and the quantum randomness. In contrast, Theorem 1(a) shows that when $\lambda_n = o(n^{(q-1)/2})$ the overlap will be zero with high probability. This establishes that $\lambda_n = \Theta(n^{(q-1)/2})$ is the weak recovery threshold of 1-step QAOA in the spiked tensor model.

**Remark 3.2** (Overlap distribution). Theorem 1 does not show that the overlap distribution for a typical instance $\boldsymbol{Y}$ converges to the same sine-Gaussian law. In Section 4, we perform numerical simulations that provide evidence that the overlap distribution will concentrate over the random draw of $\boldsymbol{Y}$, which would imply that the overlap distribution is indeed sine-Gaussian for any typical $\boldsymbol{Y}$.

**Comparison with classical tensor power iteration.** The 1-step tensor power iteration estimator (Eq. (2.1)) is redefined here for the reader's convenience: $\hat{\boldsymbol{u}}_1 = \sqrt{n}\boldsymbol{Y}[\hat{\boldsymbol{u}}_0^{\otimes(q-1)}]/\|\boldsymbol{Y}[\hat{\boldsymbol{u}}_0^{\otimes(q-1)}]\|_2$, where $\hat{\boldsymbol{u}}_0 \sim \mathrm{Unif}(\mathbb{S}^{n-1})$ is a random initialization vector. In the following proposition, we show that the weak recovery threshold for the 1-step power iteration estimator is also $\lambda_n = \Theta(n^{(q-1)/2})$, and we provide the distribution of the overlap $\mathcal{R}_{\mathrm{PI}} \equiv \hat{\boldsymbol{u}}_1^\top \boldsymbol{u}/n$ between the power iteration estimator $\hat{\boldsymbol{u}}_1$ and the signal $\boldsymbol{u}$.

**Proposition 3.3** (Weak recovery threshold for 1-step tensor power iteration). *Assume that the rescaled signal-to-noise ratio has a limit $\lim_{n\to\infty} \lambda_n/n^{(q-1)/2} = \Lambda$. Then over the randomness of $\boldsymbol{W}$ and initialization $\hat{\boldsymbol{u}}_0$, the overlap $\mathcal{R}_{\mathrm{PI}}$ of the power iteration estimator with the signal converges in distribution to*

$$\mathcal{R}_{\mathrm{PI}} \xrightarrow{d} \sin[\arctan(\Lambda G^{q-1})], \quad \text{where } G \sim \mathcal{N}(0,1). \tag{3.5}$$

*As a corollary, when $\lim_{n\to\infty} \lambda_n/n^{(q-1)/2} = 0$, we have $\mathcal{R}_{\mathrm{PI}} \xrightarrow{p} 0$.*

The proof of Proposition 3.3 is contained in Appendix H.1.

**Remark 3.4** (Comparing the overlaps). Theorem 1 and Proposition 3.3 show that both 1-step QAOA and 1-step power iteration have the same weak recovery threshold $\lambda_n = \Theta(n^{(q-1)/2})$. To compare the two algorithms more precisely, we take the limit $\lim_{n\to\infty} \lambda_n/n^{(q-1)/2} = \Lambda$ for some small $\Lambda > 0$. Eq. (3.3) and Eq. (3.5) give the limiting squared overlap distributions for 1-step QAOA and 1-step power iteration, respectively:

$$\lim_{\Lambda\to 0} \Lambda^{-2} \lim_{n\to\infty} \mathbb{E}_{\boldsymbol{Y}}[\langle \mathcal{R}_{\mathrm{QAOA}}^2 \rangle_{\gamma,\beta}] = e^{-4q\gamma^2} 4q^2\gamma^2 \sin^2(2\beta)\, \mathbb{E}_{G\sim\mathcal{N}(0,1)}[G^{2q-2}],$$
$$\lim_{\Lambda\to 0} \Lambda^{-2} \lim_{n\to\infty} \mathbb{E}_{\boldsymbol{Y}}[\mathcal{R}_{\mathrm{PI}}^2] = \mathbb{E}_{G\sim\mathcal{N}(0,1)}[G^{2q-2}]. \tag{3.6}$$

This gives

$$\max_{\gamma,\beta} \left\{ \lim_{\Lambda\to 0+} \lim_{n\to\infty} \mathbb{E}_{\boldsymbol{Y}}[\langle \mathcal{R}_{\mathrm{QAOA}}^2 \rangle_{\gamma,\beta}]/\, \mathbb{E}_{\boldsymbol{Y}}[\mathcal{R}_{\mathrm{PI}}^2] \right\} = e^{-4q\gamma_\star^2} 4q^2\gamma_\star^2 \sin^2(2\beta_\star) = q/e, \tag{3.7}$$

where the maximizer is $(\gamma_\star, \beta_\star) = (\frac{1}{2\sqrt{q}}, \pi/4)$. Thus, for $q > e$, 1-step QAOA gives better overlap than 1-step power iteration.

**Remark 3.5** (Rounding via $\mathrm{sign}(\hat{\boldsymbol{u}})$ will not improve the overlap). The readers may wonder whether the overlap of tensor power iteration will be improved by rounding the estimator via $\bar{\boldsymbol{u}}_1 = \mathrm{sign}(\hat{\boldsymbol{u}}_1) \in \{\pm 1\}^n$, outputting an estimator in the signal space. Defining $\overline{\mathcal{R}}_{\mathrm{PI}} = \bar{\boldsymbol{u}}_1^\top \boldsymbol{u}/n$, it is straightforward to show that as $\lim_{n\to\infty} \lambda_n/n^{(q-1)/2} = \Lambda$,

$$\overline{\mathcal{R}}_{\mathrm{PI}} \xrightarrow{d} \Phi(\Lambda G^{q-1}), \quad \text{where } G \sim \mathcal{N}(0,1), \;\; \Phi(t) = 2 \times \mathbb{P}_{Z\sim\mathcal{N}(0,1)}(Z \le t) - 1. \tag{3.8}$$

Hence, the computational threshold has the same exponent by rounding, but the overlap becomes smaller:

$$\lim_{\Lambda\to 0} \Lambda^{-2} \lim_{n\to\infty} \mathbb{E}_{\boldsymbol{Y}}[\overline{\mathcal{R}}_{\mathrm{PI}}^2] = (2/\pi) \cdot \mathbb{E}_{G\sim\mathcal{N}(0,1)}[G^{2q-2}]. \tag{3.9}$$

**Remark 3.6** (Sine-Gaussian law versus sine-arctan-Gaussian law). The sine-Gaussian law of QAOA is particularly interesting in that the overlap will not concentrate as $\Lambda \to \infty$. Instead, it will satisfy a sine-uniform distribution, i.e., $\sin(2q\Lambda\gamma G^{q-1}) \xrightarrow{d} \sin(U)$ for $U \sim \mathrm{Unif}([0, 2\pi])$. In contrast, the sine-arctan-Gaussian law of tensor power iteration will concentrate at $\{\pm 1\}$ as $\Lambda \to \infty$.

In Appendix E, we also study the scenario where prior information about the signal may be leveraged to recover the signal with a smaller SNR. There, we rigorously analyzed the 1-step QAOA applied to boost the signal in a weak estimator in Theorem 2, and compared it classical power iteration in Proposition E.2. Our result shows that the 1-step QAOA has the same asymptotic computational efficiency as 1-step power iteration, albeit with a constant-factor better overlap in the $\Lambda \ll 1$ regime when $q > e$.

## 3.2 Weak recovery threshold and overlap distribution for $p$-step QAOA

We next consider the general $p$-step QAOA for weak recovery in the spiked tensor model. Although it is known that the QAOA is able to output the MLE that weakly recovers the signal when $p$ grows unboundedly with $n$, here we focus on a more analytically tractable regime where $p$ is an arbitrary fixed constant in the $n \to \infty$ limit. Using a physics-style derivation, we show that the $p$-step QAOA can achieve weak recovery when the signal-to-noise ratio satisfies

$$\lambda_n = \Omega\left(n^{(q-2+\varepsilon_p)/2}\right), \quad \text{where} \quad \varepsilon_p = \begin{cases} \frac{q-2}{(q-1)^p - 1}, & q > 2, \\ 1/p, & q = 2. \end{cases} \tag{3.10}$$

Observe that $0 < \varepsilon_p \leq 1$ and $\lim_{p\to\infty} \varepsilon_p = 0$. Hence, the $p$-step QAOA can recover the signal with a progressively weaker SNR as $p$ increases. Moreover, we are able to characterize the overlap distribution $\mathcal{R}_{\text{QAOA}} \equiv \hat{u}^\top u / n$ of $p$-step QAOA between a sample $\hat{u} \sim |\gamma, \beta\rangle$ (see Eq. (2.3)) and the signal $u$ as follows:

**Claim 3.7** ($p$-step QAOA for weak recovery). *Consider the $p$-step QAOA with parameters $\{(\gamma_n, \beta_n)\}_{n\geq 1}$ applied to the spiked tensor model* (1.1) *with signal-to-noise ratio $\{\lambda_n\}_{n\geq 1}$. Suppose*

$$\lim_{n\to\infty} \left(\gamma_n, \beta_n, \lambda_n / n^{(q-2+\varepsilon_p)/2}\right) = (\gamma, \beta, \Lambda). \tag{3.11}$$

*Then, there are parameter-dependent coefficients $(a_p(\gamma, \beta), b_p(\gamma, \beta))$ such that over the randomness of $Y$ and the quantum measurement, the overlap $\mathcal{R}_{\text{QAOA}}$ of the $p$-step QAOA converges in distribution to a sine-Gaussian law as*

$$\mathcal{R}_{\text{QAOA}} \xrightarrow{d} a_p \sin(b_p \Lambda^{1/\varepsilon_p} G^{(q-1)^p}), \quad \text{where} \quad G \sim \mathcal{N}(0,1). \tag{3.12}$$

The derivation of Claim 3.7 is contained in Appendix D. We remark that our derivation uses non-rigorous heuristics from physics such as the Dirac delta function and its Fourier transform to linearize exponents in combinatorial sums (see Appendix D.1 for a sketch). Analytical expressions for the coefficients $a_p(\gamma, \beta)$ and $b_p(\gamma, \beta)$ can be found in Appendix D.5.

**Remark 3.8** (Weak recovery threshold). As $\Lambda \to 0$, Eq. (3.12) implies that $\mathcal{R}_{\text{QAOA}} \xrightarrow{p} 0$. Thus, Claim 3.7 implies that $\lambda_n = \Theta(n^{(q-2+\varepsilon_p)/2})$ is the weak recovery threshold by the $p$-step QAOA in the spiked tensor model in the regime of fixed QAOA parameter. We believe this scaling is also the weak recovery threshold for the QAOA with any sequence of parameters $(\gamma_n, \beta_n)$, but proving this requires ruling out better performance of the QAOA when $(\gamma_n, \beta_n)$ is allowed to depend strongly on $n$ as we have done in Theorem 1(a); we leave this as future work. Since $\varepsilon_p \to 0$ as $p \to \infty$, this means $\lambda_n = \Theta(n^{(q-2)/2})$ is the recovery threshold given a diverging number of QAOA steps (but constant with respect to $n$). However, this does not achieve the $\Theta(n^{(q-2)/4})$ computational threshold for classical algorithms.

**Comparison with classical tensor power iteration.** We now compare the overlap from the $p$-step QAOA to that from the classical $p$-step tensor power iteration algorithm. We show that the weak recovery threshold for the $p$-step power iteration estimator is also $\lambda_n = \Theta(n^{(q-2+\varepsilon_p)/2})$, and we provide the distribution of the overlap $\mathcal{R}_{\text{PI}} \equiv \hat{u}_p^\top u / n$ between the $p$-step power iteration estimator $\hat{u}_p$ (see Eq. (2.1)) and the signal $u$.

**Proposition 3.9** (Corollary of Lemma 3.2 of [33]). *Consider a random instance of the spiked tensor model with $\lim_{n\to\infty} \lambda_n / n^{(q-2+\varepsilon_p)/2} = \Lambda$. The overlap $\mathcal{R}_{\text{PI}}$ of the $p$-step tensor power iteration algorithm converges in distribution as*

$$\mathcal{R}_{\text{PI}} \xrightarrow{d} \sin\left[\arctan(\Lambda^{1/\varepsilon_p} G^{(q-1)^p})\right], \quad \text{where} \quad G \sim \mathcal{N}(0,1). \tag{3.13}$$

The proof of Proposition 3.9 is contained in Appendix H.3.

**Remark 3.10** (Comparing the overlaps). In the small $\Lambda \ll 1$ regime, we have

$$\mathcal{R}_{\text{QAOA}} \asymp (|a_p b_p|^{\varepsilon_p} \Lambda)^{1/\varepsilon_p} G^{(q-1)^p} \quad \text{and} \quad \mathcal{R}_{\text{PI}} \asymp \Lambda^{1/\varepsilon_p} G^{(q-1)^p}. \tag{3.14}$$

When $|a_p b_p| > 1$, the QAOA has a constant factor advantage over the classical power iteration algorithm in the overlap achieved, assuming the conjectured Claim 3.7 based on heuristic derivations. To quantify this advantage, we consider the quantum enhancement factor, $|a_p b_p|^{\varepsilon_p}$, which is the factor that the signal-to-noise ratio can shrink for the QAOA while maintaining the same overlap as the power iteration algorithm. Effectively, this factor $|a_p b_p|^{\varepsilon_p}$ corresponds to a quantum improvement in the recovery threshold by the QAOA over classical power iteration. We numerically optimize $|a_p b_p|^{\varepsilon_p}$ with respect to the QAOA parameters $(\boldsymbol{\gamma}, \boldsymbol{\beta})$, and present the optimized values in Table 1.

Table 1: **The quantum enhancement factor** $|a_p b_p|^{\varepsilon_p}$ **of the $p$-step QAOA over the $p$-step tensor power iteration**, for spiked $q$-tensors when $\lambda_n = \Lambda n^{(q-2+\varepsilon_p)/2}$ in the $\Lambda \ll 1$ regime. Note in the first row, which corresponds to $p = 1$ with $\varepsilon_1 = 1$, we know the optimal value $|a_1 b_1| = \sqrt{q/e}$ from Eq. (3.7). The remaining values are optimized via a quasi-Newton method starting with 1000 heuristic initial guesses of $(\boldsymbol{\gamma}, \boldsymbol{\beta})$ and keeping the best value; hence, they currently should be considered as lower bounds on the best possible enhancement factors.

| $p$ \ $q$ | 2 | 3 | 4 | 5 | 6 | 7 |
|---|---|---|---|---|---|---|
| 1 | 0.8578 | 1.0505 | 1.2131 | 1.3562 | 1.4857 | 1.6047 |
| 2 | 0.9663 | 1.0505 | 1.1916 | 1.2882 | 1.4167 | 1.5162 |
| 3 | 1.0204 | 1.0314 | 1.1615 | 1.2555 | 1.3844 | 1.4917 |
| 4 | 1.0487 | 1.0144 | 1.1419 | 1.2447 | 1.3795 | 1.4858 |
| 5 | 1.0631 | 1.0063 | 1.1327 | 1.2411 | 1.3770 | 1.4845 |
| 6 | 1.0697 | 1.0013 | 1.1297 | 1.2399 | 1.3743 | 1.4842 |
| 7 | 1.0719 | | | | | |

**Remark 3.11** (Weak recovery threshold for QAOA with tensor unfolding). Although neither the constant-step QAOA nor the tensor power iteration matches the $\Theta(n^{(q-2)/4})$ recovery threshold for the best polynomial-time classical algorithms, we can achieve this threshold using the idea of tensor unfolding. When $q$ is even, the tensor $\boldsymbol{Y} \in \mathbb{R}^{n^q}$ can be unfolded into a matrix $\overline{\boldsymbol{Y}}$:

$$\overline{\boldsymbol{Y}} = (\lambda_n/n^{q/2}) \cdot \bar{\boldsymbol{u}} \bar{\boldsymbol{u}}^\top + (1/\sqrt{n}) \cdot \overline{\boldsymbol{W}} \in \mathbb{R}^{n^{q/2} \times n^{q/2}}. \tag{3.15}$$

Here $\overline{Y}_{(j_1,\ldots,j_{q/2}),(j_{q/2+1},\ldots,j_q)} = Y_{j_1 \cdots j_q}$, $\overline{W}_{(j_1,\ldots,j_{q/2}),(j_{q/2+1},\ldots,j_q)} = W_{j_1 \cdots j_q}$, and $\bar{\boldsymbol{u}} = \text{vec}(\boldsymbol{u}^{\otimes(q/2)}) \in \{\pm 1\}^{n^{q/2}}$. Existing work [2, 47] have demonstrated that the leading eigenvector $\bar{z}$ of $\overline{\boldsymbol{Y}}$ has non-vanishing correlation with the signal $\bar{\boldsymbol{u}}$ as soon as $\lambda_n > n^{(q-2)/4}$. Furthermore, for the eigenvector $\bar{z}$ in such a regime, standard analysis as in [2] implies that the top singular vector of $\text{mat}(\bar{z}) \in \mathbb{R}^{n \times n^{q/2-1}}$ will have non-trivial overlap with the signal $\boldsymbol{u}$, achieving the $\Theta(n^{(q-2)/4})$ weak recovery threshold for the spectral method with tensor-unfolding.

A similar tensor-unfolding pre-processing could be applied to the QAOA to improve the computational threshold. Indeed, the QAOA method could be adopted to maximize the cost function $\overline{C}(\bar{\boldsymbol{\sigma}}) = \bar{\boldsymbol{\sigma}}^\top \overline{\boldsymbol{Y}} \bar{\boldsymbol{\sigma}} / n^{(q-1)/2}$ with decision variable $\bar{\boldsymbol{\sigma}} \in \{\pm 1\}^{n^{q/2}}$. Notice that such a QAOA method needs to be applied to a $n^{q/2}$-qubit system. Effectively, $\overline{C}(\bar{\boldsymbol{\sigma}})$ could be interpreted as the cost function of a spiked 2-tensor model of size $\bar{n} = n^{q/2}$ and with a rescaled signal-to-noise ratio $\bar{\lambda}_n = \lambda_n/n^{(q-2)/4}$. According to Claim 3.7, $p$-step QAOA outputs a long bit-string $\bar{z} \in \{\pm 1\}^{n^{q/2}}$ overlapping with the signal $\bar{\boldsymbol{u}}$ as long as $\bar{\lambda}_n = \bar{n}^{\varepsilon_p/2}$ for $\varepsilon_p = 1/p$. Translating to the scaling of $\lambda_n$, the computational threshold for QAOA with tensor unfolding is $\lambda_n = \Omega(n^{(q-2+\varepsilon_p')/4})$ where $\varepsilon_p' = q/p$. This recovers the classical $\Theta(n^{(q-2)/4})$ threshold as $p \to \infty$.

# 4 Numerical simulations

We now validate our theoretical results by conducting numerical simulations of the QAOA through classical computers. In this section, we focus on the case of 1-step QAOA ($p = 1$) for the spiked

matrix model ($q = 2$), where we can obtain an explicit formula the expected squared overlap at any finite problem dimension $n$ (see Appendix F for a derivation):

$$\mathbb{E}_{\boldsymbol{Y}}[\langle \mathcal{R}^2_{\mathrm{QAOA}}\rangle_{\gamma,\beta}] = \frac{n-1}{2n}e^{-8\gamma^2(n-2)/n}\sin^2(2\beta)[1 - \cos^{n-2}(8\lambda\gamma/n)]$$
$$+ \frac{n-1}{n}e^{-4\gamma^2(n-1)/n}\sin(4\beta)\sin(4\lambda\gamma/n)\cos^{n-2}(4\lambda\gamma/n) + \frac{1}{n}. \tag{4.1}$$

In Fig. 2(a), we report the overlap distribution of 1-step QAOA ($p = 1$) for the spiked matrix model ($q = 2$) where the SNR is chosen as $\lambda_n = n^{1/2}$. The histogram shows the Monte Carlo simulation results following the predicted sine-Gaussian law. The dashed gray lines are from the simulations of the QAOA using classical algorithms for $n = 26$, each corresponding to one of 40 instances. Note that simulating QAOA classically has complexity $O(2^n)$, which limits us to $n = 26$. We see

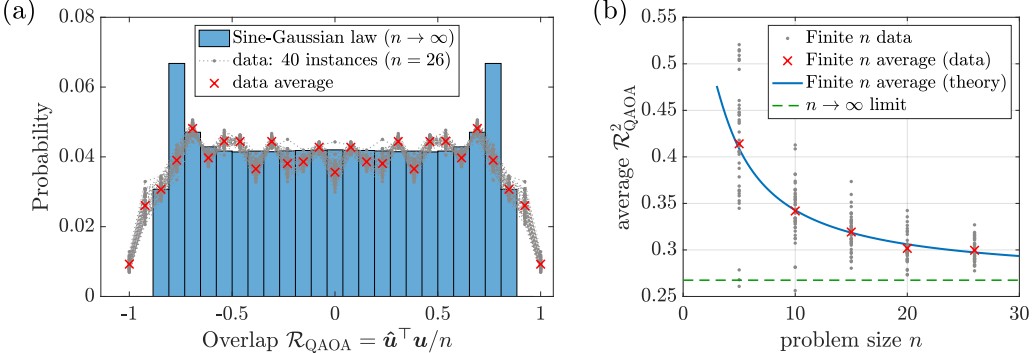

Figure 2: (a) Example overlap distribution from 1-step QAOA for the spiked matrix model ($q = 2$), where simulation data is collected from 40 random generated instances with $n = 26$ bits. The signal-to-noise ratio is chosen to be $\lambda_n = n^{1/2}$, and $(\gamma, \beta) = (\sqrt{\ln 5/32}, \pi/4)$. Dash gray lines connect data from the same instance. (b) Average of squared overlap $\langle \mathcal{R}^2_{\mathrm{QAOA}}\rangle_{\gamma,\beta}$ from the QAOA output distribution for 40 random instances generated at various problem dimensions.

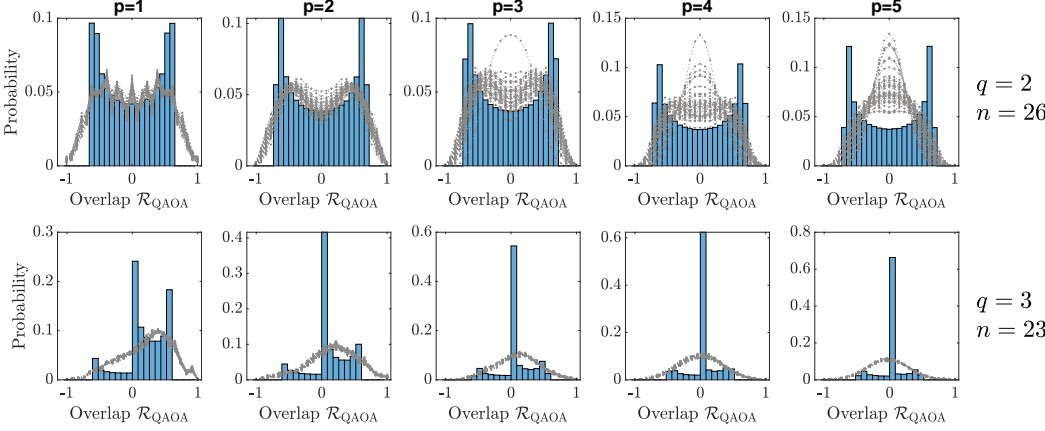

Figure 3: Example overlap distributions from $p$-step QAOA for the spiked tensor model for $1 \leq p \leq 5$. The top row shows data from 40 random 26-bit instances with $q = 2$ and $\lambda_n = n^{1/(2p)}$. The bottom row shows data from 40 random 23-bit instances with $q = 3$ and $\lambda_n = n^{[1+1/(2^p-1)]/2}$. Different columns correspond to different $p$, using the QAOA parameters $(\boldsymbol{\gamma}, \boldsymbol{\beta})$ that optimized $|a_p b_p|^{\varepsilon_p}$ in Table 1. Dash gray lines connect data from the same instance. Blue histograms are the theoretical sine-Gaussian distributions in the $n \to \infty$ limit, where $\mathcal{R}_{\mathrm{QAOA}} \sim a_p \sin[b_p G^{(q-1)^p}]$ according to Claim 3.7. (Note here $\Lambda = 1$.)

that, despite some finite sample effects, the predicted sine-Gaussian distribution matches the QAOA simulation.

Fig. 2(b) reports the expected squared overlap from the QAOA simulations. The green dashed line is the theoretical prediction in the $n \to \infty$ limit. The blue solid line is the finite $n$ theoretical prediction from Eq. (4.1). The gray dots are the squared overlaps from individual QAOA instances simulated classically. The average over instances (red crosses) agrees well with the finite $n$ theory prediction, which converges to the $n \to \infty$ limit with order $1/n$ deviation.

We also perform simulations for $1 \leq p \leq 5$ and $q = 2, 3$. Fig. 3 plots the overlap distribution for $p$-step QAOA. The simulation curves follow the shape of the theoretical histograms for $p \leq 2$. For $p \geq 3$, the shapes of the simulated and theoretical overlap distributions do not match well, likely due to finite size effects (simulations for large $n > 26$ are computationally challenging).

In Appendix G, we present additional numerical simulation results on higher $(p, q)$ and find the second moment of the QAOA overlap converges to our theoretical predictions up to $O(1/n)$ deviations. We also describe more details of the simulation methods.

An interesting phenomenon apparent from Fig. 2(a) and Fig. 3 is that the output distribution of the QAOA appears to concentrate over the randomness of instances $\boldsymbol{Y}$, but not over the quantum measurements. This is in stark contrast to previous concentration results on the QAOA where concentration over measurements were shown, e.g., for spin-glass models in [24, 26, 32]. We note that such anti-concentration is also expected in the limit of zero noise ($\lambda \to \infty$), where it is known the constant-$p$ QAOA can prepare the GHZ state [48]. Since existing limitations of both classical [29] and quantum algorithms [26, 30–32] on various problems over random structures rely heavily on concentration, extending these negative results to the QAOA for the spiked tensor model do not seem possible due to the absence of concentration. Nevertheless, our analysis shows that the constant-$p$ QAOA is unable to improve the recovery threshold in the spiked tensor model achieved by classical algorithms by more than a constant factor.

## 5  Discussion

In this paper, we have investigated the power of quantum algorithms for the spiked tensor model, a canonical problem in statistical inference with a large computational-statistical gap that has so far eluded classical algorithms. We gave the first rigorous study of a polynomial-time quantum algorithm on this problem by analyzing the performance of the QAOA, a popular variational quantum algorithm that has been implemented on current quantum computing hardware. We showed that $p$-step QAOA achieves the same asymptotic SNR threshold for weak recovery as $p$-step tensor power iteration. A heuristic analysis showed that multi-step QAOA with tensor unfolding could achieve, but not surpass, the classical computation threshold $\Theta(n^{(q-2)/4})$. This implies that achieving a strong quantum advantage via the QAOA requires using a number of steps $p$ that grows with $n$. However, we revealed that the asymptotic overlap distribution of QAOA exhibits an intriguing sine-Gaussian law, distinct from tensor power iteration. For certain parameters $(p, q)$, the QAOA effectively has a recovery threshold that is a constant factor better, indicating a modest quantum advantage over the classical power iteration. Overall, while achieving identical scalings as power iteration, the QAOA demonstrates qualitative differences and potential for quantum speedups.

There are many interesting questions that remain open. One worthy challenge would be a rigorous proof for the $p > 1$ analysis without relying on heuristic arguments. Additionally, it would be interesting to prove that the sine-Gaussian distribution is concentrated over problem instances but not over measurements, as suggested by our simulations. This is in contrast to recent results showing that the low-depth QAOA is concentrated over measurements [26, 32], a seemingly essential ingredient for many proofs of algorithmic limitations [26, 29–32]. Despite the absence of concentration in the spiked tensor setting, our results show that the constant-$p$ QAOA has limited power, similar to the message of recent works [26, 30–32, 49–51] proving limitations up to $p = O(\log n)$. This suggests that demonstrating strong quantum advantage requires analyzing super-logarithmic depth QAOA, which remains an outstanding open question. Finally, it would be interesting to study quantum algorithms in other statistical inference models that classically exhibit computational-statistical gaps, including planted clique, Bayesian linear models, and sparse PCA. Overcoming any such gap with a polynomial-time quantum algorithm would be an exciting superpolynomial quantum speedup with practical relevance.

## Acknowledgments

We thank David Gamarnik for insightful discussions and Stuart Hadfield for detailed comments on the manuscript. We thank Yuchen Wu for providing the proof of Proposition 3.9 and Ruixiang Zhang for the helpful discussion on the potential for making Claim 3.7 rigorous. LZ acknowledges funding from the Walter Burke Institute for Theoretical Physics at Caltech. JB is partially supported by a grant from the Simons Foundation under Award No. 825053 and the NASA Ames Research Center, from NASA Academic Mission Services (NAMS) under Contract No. NNA16BD14C, and from the DARPA ONISQ program under interagency agreement IAA 8839, Annex 114. SM is supported by NSF DMS-2210827, CCF-2315725, CAREER DMS-2339904, ONR N00014-24-S-B001, an Amazon Research Award, a Google Research Scholar Award, and an Okawa Foundation Research Grant.

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

# Contents

## A   Additional background

### A.1   Review of quantum computing terminology

For the convenience of readers who are unfamiliar with quantum computing terminology, we briefly introduce relevant linear algebra concepts. A quantum state of an $n$-qubit system is a $2^n$-dimensional unit complex vector $\psi \in \mathbb{C}^{2^n}$ satisfying $\sum_{i \in [2^n]} |\psi_i|^2 = 1$. Each bit-string $z \in \{\pm 1\}^n$ associates with a quantum state $|z\rangle \in \mathbb{C}^{2^n}$, representing the $|z|$'th canonical basis vector $[0, \cdots, 0, 1, 0, \cdots, 0]^\top \in \mathbb{C}^{2^n}$, where only position $|z|$ equals 1 (with $|z| = 1 + \sum_{j \in [n]} 2^{j-1}(1 - z_j)/2$ denoting the rank of bit-string $z$). Therefore, $\psi = \sum_{z \in \{\pm 1\}} \psi_{|z|} |z\rangle$ where $|\psi_{|z|}|^2$ gives the probability of observing $z$ upon measurement. This represents $\psi$ as a probability distribution over all $2^n$ bit-strings in $\{\pm 1\}^n$.

The Pauli operators $\sigma_x, \sigma_y, \sigma_z$ on a single qubit are represented as $2 \times 2$ complex matrices:

$$I = \begin{bmatrix} 1 & 0 \\ 0 & 1 \end{bmatrix}, \quad \sigma_x = \begin{bmatrix} 0 & 1 \\ 1 & 0 \end{bmatrix}, \quad \sigma_y = \begin{bmatrix} 0 & -i \\ i & 0 \end{bmatrix}, \quad \sigma_z = \begin{bmatrix} 1 & 0 \\ 0 & -1 \end{bmatrix}. \tag{A.1}$$

In an $n$-qubit system, the Pauli operators $\{X_k, Y_k, Z_k\} \in \mathbb{C}^{2^n \times 2^n}$ associated to the $k$-th qubit are defined by $I^{\otimes(k-1)} \otimes \{\sigma_x, \sigma_y, \sigma_z\} \otimes I^{\otimes(n-k)} \in \mathbb{C}^{2^n \times 2^n}$, where $\otimes$ is the Kronecker product operator.

### A.2   Related literature on the QAOA

In terms of theoretical analysis of its computational complexity, the performance of the QAOA has been studied for various models, including the Sherrington-Kirkpatrick model [24], MaxCut [25, 27], the Max-$q$-XORSAT for regular hypergraphs [25], $q$-spin spin-glass models [26, 52], the ferromagnetic Ising model [53], and random constraint satisfaction problems [28]. While [24] shows promising evidence for the QAOA to achieve the ground state energy of the Sherrington-Kirkpatrick model [54], [26] proves that constant-step QAOA cannot achieve the ground state for $q$-spin spin-glass models in general.

There is also a line of work aiming to prove computational hardness results for the QAOA and related quantum algorithms. [30, 31, 50, 51] studied the limitation of local quantum algorithms like the QAOA for solving combinatorial optimization problems on sparse random graphs, using the bounded light-cone of the algorithms at sufficiently low depths. This limitation was translated to the dense spin-glass models in [26]. Furthermore, [32, 49] proved hardness results for the QAOA by exploiting the symmetry of the problem. Note all previously known limitations of the QAOA in the average-case setting [26, 30–32] have relied on concentration of the output distribution in the Hamming weight basis, which is not present in the spiked tensor model.

Our work studies the QAOA for a statistical inference problem, distinct from these existing results. Furthermore, we develop new techniques for analyzing the QAOA that do not exist in prior work.

# B    Moment generating function of the QAOA overlap at $p = 1$

We dedicate this section to derive a combinatorial expression for expected moment-generating function of the QAOA overlap, defined as

$$M_n(\zeta; \gamma_n, \beta_n, \lambda_n) := \langle e^{\zeta \widehat{\mathcal{R}}} \rangle_{\gamma_n, \beta_n}. \tag{B.1}$$

We write $M_n(\zeta) = M_n(\zeta; \gamma_n, \beta_n, \lambda_n)$ for short. This quantity will be used in future derivations.

We use the techniques and conventions first introduced in [24]. First, we define bistrings $\boldsymbol{a} \in B := \{\pm 1\}^3$ indexed as $\boldsymbol{a} = (a_1, a_\mathrm{m}, a_2)$. Since $p = 1$, we write $\beta = \beta_1$, $\gamma = \gamma_1$. Additionally, define the quantities given by

$$Q_{\boldsymbol{a}} = \frac{1}{2} \langle a_1 | e^{i\beta X} | 1 \rangle \langle 1 | e^{-i\beta X} | a_2 \rangle, \tag{B.2}$$

$$\Phi_{\boldsymbol{a}} = \gamma(a_1 - a_2). \tag{B.3}$$

We may also write $\{n_{\boldsymbol{a}}\}_{\boldsymbol{a} \in B} \subseteq \mathbb{Z}^{|B|}$ where $\sum_{\boldsymbol{a} \in A} n_{\boldsymbol{a}} = n$ to assign a count to each bit-string. If we underscore the bit-string, we mean $\underline{\boldsymbol{a}} = (\boldsymbol{a}_1, \boldsymbol{a}_2, \ldots, \boldsymbol{a}_q) \in B^q$. We also write $\Phi_{\underline{\boldsymbol{a}}} = \Phi_{\boldsymbol{a}_1 \boldsymbol{a}_2 \cdots \boldsymbol{a}_q}$.

We now have the notation to state the following lemma.

**Lemma B.1** (QAOA overlap expected moment-generating function in the configuration basis for $p = 1$)**.** *The expectation over the spiked tensor disorder in Eq.* (1.1) *of the moment-generating function defined in Eq.* (B.1) *for $p = 1$ is given by*

$$\mathbb{E}_{\boldsymbol{Y}}[M_n(\zeta)] = \sum_{\{n_{\boldsymbol{a}}\}} \binom{n}{\{n_{\boldsymbol{a}}\}} \prod_{\boldsymbol{a} \in B} Q_{\boldsymbol{a}}^{n_{\boldsymbol{a}}} \exp \left[ -\frac{1}{2n^{q-1}} \sum_{\underline{\boldsymbol{a}} \in B^q} \Phi_{\underline{\boldsymbol{a}}}^2 \prod_{s=1}^{q} n_{\boldsymbol{a}_s} + \frac{i\lambda_n}{n^{q-1}} \sum_{\underline{\boldsymbol{a}} \in B^q} \Phi_{\underline{\boldsymbol{a}}} \prod_{s=1}^{q} (\boldsymbol{a}_s)_\mathrm{m} n_{\boldsymbol{a}_s} \right.$$

$$\left. + \frac{\zeta}{n} \sum_{\boldsymbol{v} \in B} v_\mathrm{m} n_{\boldsymbol{v}} \right]. \tag{B.4}$$

*Proof of Lemma B.1.* Without loss of generality, we assume that $\boldsymbol{u} = \boldsymbol{1}$ and proceed as in [24, Section 5] and [26, Appendix D.2]. By definition, we have that

$$M_n(\zeta) = \langle \boldsymbol{\gamma}, \boldsymbol{\beta} | e^{\zeta \widehat{\mathcal{R}}} | \boldsymbol{\gamma}, \boldsymbol{\beta} \rangle$$

$$= \langle s | e^{i\gamma C} e^{i\beta B} e^{\zeta \widehat{\mathcal{R}}} e^{-i\beta B} e^{i\gamma C} | s \rangle. \tag{B.5}$$

Inserting 3 resolutions of the identity $\mathbb{I} = \sum_{\boldsymbol{z}} |\boldsymbol{z}\rangle \langle \boldsymbol{z}|$ observing that every computation basis state $|\boldsymbol{z}\rangle$ is an eigenvector of $C$ and $\widehat{\mathcal{R}}$, we have that

$$M_n(\zeta) = \sum_{\boldsymbol{z}^1, \boldsymbol{z}^m, \boldsymbol{z}^2} \langle s | e^{i\gamma C} | \boldsymbol{z}^1 \rangle \langle \boldsymbol{z}^1 | e^{i\beta B} e^{\zeta \widehat{\mathcal{R}}} | \boldsymbol{z}^m \rangle \langle \boldsymbol{z}^m | e^{-i\beta B} | \boldsymbol{z}^2 \rangle \langle \boldsymbol{z}^2 | e^{i\gamma C} | s \rangle$$

$$= \frac{1}{2^n} \sum_{\boldsymbol{z}^1, \boldsymbol{z}^m, \boldsymbol{z}^2} \langle \boldsymbol{z}^1 | e^{i\beta B} | \boldsymbol{z}^m \rangle e^{i\gamma C(\boldsymbol{z}^1)} e^{\zeta \widehat{\mathcal{R}}(\boldsymbol{z}^m)} e^{i\gamma C(\boldsymbol{z}^2)} \langle \boldsymbol{z}^m | e^{-i\beta B} | \boldsymbol{z}^2 \rangle$$

$$= \frac{1}{2^n} \sum_{\boldsymbol{z}^1, \boldsymbol{z}^m, \boldsymbol{z}^2} f_\beta^*(\boldsymbol{z}^1 \boldsymbol{z}^m) f_\beta(\boldsymbol{z}^m \boldsymbol{z}^2) \exp \left[ i\gamma(C(\boldsymbol{z}^1) - C(\boldsymbol{z}^2)) + \zeta \mathcal{R}(\boldsymbol{z}^m) \right]$$

$$= \frac{1}{2^n} \sum_{\boldsymbol{z}^1, \boldsymbol{z}^m, \boldsymbol{z}^2} f_\beta^*(\boldsymbol{z}^1 \boldsymbol{z}^m) f_\beta(\boldsymbol{z}^m \boldsymbol{z}^2)$$

$$\times \exp \left[ i\gamma \sum_{i_1, \ldots, i_q = 1}^{n} \left( \frac{\lambda_n}{n^{q-1}} + \frac{W_{i_1, \ldots, i_q}}{n^{(q-1)/2}} \right) (z_{i_1}^1 \cdots z_{i_q}^1 - z_{i_1}^2 \cdots z_{i_q}^2) + \frac{\zeta}{n} \sum_{j=1}^{n} z_j^m \right], \tag{B.6}$$

where we defined $f_\beta(\boldsymbol{z}\boldsymbol{z}') = \langle \boldsymbol{z} | e^{-i\beta B} | \boldsymbol{z}' \rangle$ since this quantity only depends on the bitwise product $\boldsymbol{z}\boldsymbol{z}'$. We also used the definitions of $C(\boldsymbol{z}) = \langle \boldsymbol{Y}, \boldsymbol{z}^{\otimes q} \rangle / n^{(q-2)/2}$ and $\widehat{\mathcal{R}}(\boldsymbol{z}) = \frac{1}{n} \sum_{j=1}^{n} z_j$. Next, we tranform the $\boldsymbol{z}^j$ as follows:

$$\boldsymbol{z}^1 \rightarrow \boldsymbol{z}^1 \boldsymbol{z}^m, \qquad \boldsymbol{z}^2 \rightarrow \boldsymbol{z}^2 \boldsymbol{z}^m. \tag{B.7}$$

This gives

$$
\begin{aligned}
M_n(\zeta) = \frac{1}{2^n} \sum_{\boldsymbol{z}^1, \boldsymbol{z}^m, \boldsymbol{z}^2} & f_\beta^*(\boldsymbol{z}^1) f_\beta(\boldsymbol{z}^2) \\
& \times \exp\left[ i\gamma \sum_{i_1,\ldots,i_q=1}^{n} \left( \frac{\lambda_n}{n^{q-1}} + \frac{W_{i_1,\ldots,i_q}}{n^{(q-1)/2}} \right) z_{i_1}^m \cdots z_{i_q}^m (z_{i_1}^1 \cdots z_{i_q}^1 - z_{i_1}^2 \cdots z_{i_q}^2) + \frac{\zeta}{n} \sum_{j=1}^{n} z_j^m \right] \\
= \frac{1}{2^n} \sum_{\boldsymbol{z}^1, \boldsymbol{z}^m, \boldsymbol{z}^2} & f_\beta^*(\boldsymbol{z}^1) f_\beta(\boldsymbol{z}^2) \\
& \times \exp\left[ i \sum_{i_1,\ldots,i_q=1}^{n} \left( \frac{\lambda_n}{n^{q-1}} + \frac{W_{i_1,\ldots,i_q}}{n^{(q-1)/2}} \right) z_{i_1}^m \cdots z_{i_q}^m \Phi_{i_1,\ldots,i_q}(\boldsymbol{Z}) + \frac{\zeta}{n} \sum_{j=1}^{n} z_j^m \right], \quad \text{(B.8)}
\end{aligned}
$$

where we denoted $\boldsymbol{Z} = (\boldsymbol{z}^1, \boldsymbol{z}^2)$ and

$$
\Phi_{i_1,\ldots,i_q}(\boldsymbol{Z}) = \gamma(z_{i_1}^1 \cdots z_{i_q}^1 - z_{i_1}^2 \cdots z_{i_q}^2). \quad \text{(B.9)}
$$

Hence, the expected moment-generating function is

$$
\begin{aligned}
\mathbb{E}_{\boldsymbol{Y}}[M_n(\zeta)] = \frac{1}{2^n} \sum_{\boldsymbol{z}^1, \boldsymbol{z}^m, \boldsymbol{z}^2} & f_\beta^*(\boldsymbol{z}^1) f_\beta(\boldsymbol{z}^2) \\
& \times \exp\left[ \sum_{i_1,\ldots,i_q=1}^{q} \left( \frac{i\lambda_n}{n^{q-1}} z_{i_1}^m \cdots z_{i_q}^m \Phi_{i_1,\ldots,i_q}(\boldsymbol{Z}) - \frac{1}{2n^{q-1}} \Phi_{i_1,\ldots,i_q}^2(\boldsymbol{Z}) \right) + \frac{\zeta}{n} \sum_{j=1}^{n} z_j^m \right]. \\
& \quad \text{(B.10)}
\end{aligned}
$$

Now we change to the so-called configuration basis. For any bit-string $1 \leq j \leq n$, we look at a new bit-string:

$$
(z_j^1, z_j^m, z_j^2) \in B. \quad \text{(B.11)}
$$

For any $\boldsymbol{a} \in B$, we represent by $n_{\boldsymbol{a}}$ the number of times that configuration $\boldsymbol{a}$ happens. Note that $\sum_{\boldsymbol{a} \in B} = n$. For more details, we again refer the reader to [26, Appendix D.2]. Now, instead of counting over each bit of $\boldsymbol{z}^1, \boldsymbol{z}^m, \boldsymbol{z}^2$, we can count over configurations in $A$:

$$
\begin{aligned}
\mathbb{E}_{\boldsymbol{Y}}[M_n(\zeta)] = \sum_{\{n_{\boldsymbol{a}}\}} \binom{n}{\{n_{\boldsymbol{a}}\}} \prod_{\boldsymbol{a} \in B} Q_{\boldsymbol{a}}^{n_{\boldsymbol{a}}} \exp\Bigg[ & \frac{i\lambda_n}{n^{q-1}} \sum_{\boldsymbol{a}_1,\ldots,\boldsymbol{a}_q \in B} \Phi_{\boldsymbol{a}_1 \cdots \boldsymbol{a}_q}(\boldsymbol{a}_1)_m \cdots (\boldsymbol{a}_q)_m n_{\boldsymbol{a}_1} \cdots n_{\boldsymbol{a}_q} \\
& - \frac{1}{2n^{q-1}} \sum_{\boldsymbol{a}_1,\ldots,\boldsymbol{a}_q \in B} \Phi_{\boldsymbol{a}_1 \cdots \boldsymbol{a}_q}^2 n_{\boldsymbol{a}_1} \cdots n_{\boldsymbol{a}_q} + \frac{\zeta}{n} \sum_{\boldsymbol{a} \in B} a_m n_{\boldsymbol{a}} \Bigg], \quad \text{(B.12)}
\end{aligned}
$$

which finishes the proof of Lemma B.1. $\qquad\square$

## C  Proof of Theorem 1

### C.1  Proof sketch for Theorem 1(b) and emergence of sine-Gaussian law.

Here we briefly sketch the proof of Theorem 1 for 1-step QAOA, explain how the sine-Gaussian law appears, and highlight the technical ideas.

To derive the distribution of the QAOA overlap, we compute its expected moment-generating function. We start by following the steps from [24, 26] to reformulate the expected moment-generating function (MGF). With some algebra, we arrive at the following equation (see Appendix B and Lemma C.1 for

the derivation):

$$\mathbb{E}_{\boldsymbol{Y}}[\langle e^{\zeta\widehat{\mathcal{R}}}\rangle_{\gamma,\beta}] = \sum_{t=0}^{n} \binom{n}{t}\left(\frac{\sin(2\beta)}{4n}\right)^t e^{-\gamma^2[n^q-(n-2t)^q]/n^{q-1}} S_{n,t} \tag{C.1}$$

$$S_{n,t} = n^t \sum_{\sum n_i = t} \binom{t}{\{n_i\}}(-i)^{n_1-n_2+n_3-n_4} e^{(\zeta/n)(n_1+n_2-n_3-n_4)} Z_{n,t}(n_1-n_2-n_3+n_4), \tag{C.2}$$

$$Z_{n,t}(k) = \frac{1}{2^{n-t}} \sum_{\tau_++\tau_-=n-t} \binom{n-t}{\tau_+,\tau_-}(e^{\zeta/n}\cos^2\beta + e^{-\zeta/n}\sin^2\beta)^{\tau_+}(e^{-\zeta/n}\cos^2\beta + e^{\zeta/n}\sin^2\beta)^{\tau_-}$$
$$\times e^{i\Lambda\gamma[((\tau_+-\tau_-)+k)^q-((\tau_+-\tau_-)-k)^q]/n^{(q-1)/2}}. \tag{C.3}$$

Looking upon the term $Z_{n,t}(k)$, we can interpret the summand $\tau_+$ as a binomial variable $\mathrm{Binom}(n-t, 1/2)$, and by the Central Limit Theorem, we have $(\tau_+ - \tau_-)/\sqrt{n} \xrightarrow{d} G \sim \mathcal{N}(0,1)$. This gives

$$\lim_{n\to\infty} Z_{n,t}(k) = \mathbb{E}_{G\sim\mathcal{N}(0,1)}[e^{i2qk\Lambda\gamma G^{q-1}}] =: Z_t(k).$$

This is the step where the power of Gaussian appears. Next, assuming that we can replace $Z_{n,t}$ by $Z_t$ in the expression of $S_{n,t}$ as in (C.2), and using the multinomial theorem, we get

$$S_{n,t} \doteq \mathbb{E}_{G\sim\mathcal{N}(0,1)} n^t \sum_{\sum n_i = t} \binom{t}{\{n_i\}}(-i)^{n_1-n_2+n_3-n_4} e^{(\zeta/n)(n_1+n_2-n_3-n_4)} e^{(n_1-n_2-n_3+n_4)i2q\Lambda\gamma G^{q-1}}$$

$$= \mathbb{E}_{G\sim\mathcal{N}(0,1)}\{[4n\sinh(\zeta/n)\sin(2q\Lambda\gamma G^{q-1})]^t\} \to \mathbb{E}_{G\sim\mathcal{N}(0,1)}\{[4\zeta\sin(2q\Lambda\gamma G^{q-1})]^t\} =: S_t.$$

This is the step where the sine-Gaussian distribution appears. Finally, suppose that we can replace $S_{n,t}$ by $S_t$ in (C.1), and using the Taylor expansion of the exponential function, we get

$$\mathbb{E}_{\boldsymbol{Y}}[\langle e^{\zeta\widehat{\mathcal{R}}}\rangle_{\gamma,\beta}] \doteq \sum_{t=0}^{n} \binom{n}{t}\left(\frac{\sin(2\beta)}{4n}\right)^t e^{-\gamma^2[n^q-(n-2t)^q]/n^{q-1}} \mathbb{E}_{G\sim\mathcal{N}(0,1)}\left\{[4\zeta\sin(2q\Lambda\gamma G^{q-1})]^t\right\}$$

$$\xrightarrow{\cdot} \mathbb{E}\left\{\sum_{t=0}^{\infty} \frac{1}{t!}[\zeta e^{-2q\gamma^2}\sin(2\beta)\sin(2q\Lambda\gamma G^{q-1})]^t\right\} = \mathbb{E}_{G\sim\mathcal{N}(0,1)}\left\{e^{\zeta e^{-2q\gamma^2}\sin(2\beta)\sin(2q\Lambda\gamma G^{q-1})}\right\}.$$

This gives the moment-generating function of the sine-Gaussian law.

We should notice that several steps in the above proof sketch are non-rigorous, in the sense that we could not sequentially take $n \to \infty$ in $Z_{n,t}$, $S_{n,t}$, and the MGF. To make this step rigorous, we use the idea of discrete Fourier transform in Eq. (C.2) to decouple the two terms $e^{(\zeta/n)(n_1+n_2-n_3-n_4)}$ and $Z_{n,t}$ (see Lemma C.1), which allows one to treat the $n \to \infty$ limit of these two terms separately in the expression of $S_{n,t}$. For more details, see the full proof in the following section.

## C.2   Proof of Theorem 1(b)

To prove Theorem 1(b), it suffices to show that the moment-generating function (MGF) of the QAOA overlap converges to the MGF of a sine-Gaussian law as follows:

$$\lim_{n\to\infty} \mathbb{E}_{\boldsymbol{Y}}[M_n(\zeta)] = \mathbb{E}_{G\sim\mathcal{N}(0,1)}\left\{\exp\left[\zeta e^{-2q\gamma^2}\sin(2\beta)\sin(2q\Lambda\gamma G^{q-1})\right]\right\} =: M(\zeta). \tag{C.4}$$

We start the proof of Eq. (C.4) with the following lemma, which obtains a more explicit expression for the MGF that we derived in Section B.

**Lemma C.1** (Expected moment-generating function). *The expected moment-generating function in Eq.* (B.4) *can be evaluated as*

$$\mathbb{E}_{\boldsymbol{Y}}[M_n(\zeta)] = \sum_{t=0}^{n} \binom{n}{t} e^{-\gamma^2[n^q-(n-2t)^q]/n^{q-1}}[\sinh(\zeta/n)\sin(2\beta)]^t \cdot E_{n,t}, \tag{C.5}$$

*where*

$$E_{n,t} = \frac{1}{2t+1} \sum_{\xi=-t}^{t} \sin^t(2\pi\xi/(2t+1)) \hat{Z}_{n,t}(\xi),$$

$$\hat{Z}_{n,t}(\xi) = \sum_{k=-t}^{t} e^{-2\pi i \xi k/(2t+1)} Z_{n,t}(k),$$

$$Z_{n,t}(k) = \frac{1}{2^{n-t}} \sum_{\tau_+ + \tau_- = n-t} \binom{n-t}{\tau_+, \tau_-} (e^{\zeta/n}\cos^2\beta + e^{-\zeta/n}\sin^2\beta)^{\tau_+} (e^{-\zeta/n}\cos^2\beta + e^{\zeta/n}\sin^2\beta)^{\tau_-}$$

$$\times \, e^{i\Lambda_n\gamma[((\tau_+-\tau_-)+k)^q - ((\tau_+-\tau_-)-k)^q]/n^{(q-1)/2}}.$$

(C.6)

*Here we have* $\Lambda_n = \lambda_n/n^{(q-1)/2}$.

The proof of Lemma C.1 is deferred to Section C.2.1. Now we define

$$\Lambda = \lim_{n\to\infty} \Lambda_n,$$

$$I_{n,t} = \binom{n}{t} e^{-\gamma^2[n^q - (n-2t)^q]/n^{q-1}} [\sinh(\zeta/n)\sin(2\beta)]^t \cdot E_{n,t},$$

(C.7)

$$I_t = \frac{1}{t!} \mathbb{E}_{G\sim\mathcal{N}(0,1)} \left[ [\zeta e^{-2q\gamma^2} \sin(2\beta)\sin(2q\Lambda\gamma G^{q-1})]^t \right].$$

Then it is easy to see that

$$\mathbb{E}_{\mathbf{Y}}[M_n(\zeta)] = \sum_{t=0}^{n} I_{n,t}, \quad M(\zeta) = \sum_{t=0}^{\infty} I_t.$$

As a consequence, we have

$$\left| \mathbb{E}_{\mathbf{Y}}[M_n(\zeta)] - M(\zeta) \right| \leq \sum_{t=0}^{T} |I_{n,t} - I_t| + \left| \sum_{t\geq T+1} I_t \right| + \sum_{t=T+1}^{n} |I_{n,t}|.$$

(C.8)

The following lemma gives the limit of $E_{n,t}$ for fixed $t$ as $n \to \infty$, which indicates that $I_t$ is the limit of $I_{n,t}$.

**Lemma C.2.** *For any fixed integer $t$, we have*

$$\lim_{n\to\infty} E_{n,t} = \mathbb{E}_{G\sim\mathcal{N}(0,1)}[\sin^t(2q\Lambda\gamma G^{q-1})] \equiv E_t.$$

*As a consequence, we have*

$$\lim_{n\to\infty} I_{n,t} = I_t.$$

Furthermore, we have the following upper bound of $I_{n,t}$.

**Lemma C.3.** *For any $t \leq n$ and $\zeta \leq n$, we have*

$$|I_{n,t}| \leq \frac{1}{t!}(6|\zeta|)^t(2t+1)e^{|\zeta|} \equiv s_t,$$

*where*

$$\sum_{t=0}^{\infty} s_t < \infty.$$

The proof of Lemma C.2 and C.3 is deferred to Section C.2.2 and C.2.3, respectively. Now we assume that these two lemmas hold. By the fact that $\sum_{t=0}^{\infty} I_t$ is finite and by Lemma C.3, for any $\varepsilon > 0$, there exists $T = T_\varepsilon$ such that

$$\left| \sum_{t\geq T_\varepsilon+1} I_t \right| \leq \varepsilon/3, \quad \sum_{t\geq T_\varepsilon+1} s_t \leq \varepsilon/3.$$

Furthermore, by Lemma C.2, there exists $N = N_\varepsilon$ such that as long as $n \geq N_\varepsilon$, we have

$$\sum_{t=0}^{T_\varepsilon} |I_{n,t} - I_t| \leq \varepsilon/3.$$

As a consequence, by Eq. (C.8), for any $n \geq n_\varepsilon$ and $\zeta \leq n$, we have

$$\left| \mathbb{E}_Y[M_n(\zeta)] - M(\zeta) \right| \leq \sum_{t=0}^{T_\varepsilon} |I_{n,t} - I_t| + \left| \sum_{t \geq T_\varepsilon+1} I_t \right| + \sum_{t=T_\varepsilon+1}^{\infty} s_t \leq \varepsilon. \tag{C.9}$$

This proves Eq. (C.4) as desired, and hence finishes the proof of Theorem 1(b).

### C.2.1 Proof of Lemma C.1

Our starting point is Eq. (B.4), which we can compute explicitly with a careful organization of the sum. To this end, let

$$\begin{aligned}
t_+ &= n_{++-} + n_{-++}, & t_- &= n_{+--} + n_{--+}, \\
d_+ &= n_{++-} - n_{-++}, & d_- &= n_{+--} - n_{--+}, \\
\tau_+ &= n_{+++} + n_{---}, & \tau_- &= n_{+-+} + n_{-+-}, \\
\Delta_+ &= n_{+++} - n_{---}, & \Delta_- &= n_{+-+} - n_{-+-}.
\end{aligned} \tag{C.10}$$

Observe that these 8 variables completely determine $\{n_a : a \in B\}$. Furthermore, let

$$t = t_+ + t_-, \qquad n - t = \tau_+ + \tau_-. \tag{C.11}$$

Then explicit computation shows that

$$\begin{aligned}
\sum_{\underline{a} \in B^q} \Phi_{\underline{a}}^2 \prod_{s=1}^q n_{a_s} &= 4\gamma^2 \sum_{\underline{a}} \mathbb{1}\{a_{11} \cdots a_{q1} \neq a_{12} \cdots a_{q2}\} \prod_{s=1}^q n_{a_q} \\
&= 2\gamma^2 \left[ n^q - (n - 2t)^q \right],
\end{aligned} \tag{C.12}$$

$$\begin{aligned}
\sum_{\underline{a} \in B^q} \Phi_{\underline{a}} \prod_{s=1}^q (a_s)_m n_{a_s} &= \gamma \left[ \left( \sum_a a_1 a_m n_a \right)^q - \left( \sum_a a_2 a_m n_a \right)^q \right] \\
&= \gamma[(\tau_+ - \tau_-) + (d_+ - d_-))^q - ((\tau_+ - \tau_-) - (d_+ - d_-))^q], \tag{C.13}
\end{aligned}$$

$$\sum_{v \in B} v_m n_v = t_+ - t_- + \Delta_+ - \Delta_-. \tag{C.14}$$

Plugging this into Eq. (B.4) and breaking up the sum, we get

$$\begin{aligned}
\mathbb{E}_Y[M_n(\zeta)] = &\sum_{t=0}^n \binom{n}{t} e^{-\gamma^2[n^q - (n-2t)^q]/n^{q-1}} \sum_{t_+ + t_- = t} \binom{t}{t_+, t_-} \sum_{\tau_+ + \tau_- = n-t} \binom{n-t}{\tau_+, \tau_-} \\
&\sum_{\Delta_+} \binom{\tau_+}{n_{+++}} Q_{+++}^{n_{+++}} Q_{---}^{n_{---}} \sum_{\Delta_-} \binom{\tau_-}{n_{+-+}} Q_{+-+}^{n_{+-+}} Q_{-+-}^{n_{-+-}} e^{(\zeta/n)(t_+ - t_- + \Delta_+ - \Delta_-)} \\
&\sum_{d_+} \binom{t_+}{n_{++-}} Q_{++-}^{n_{++-}} Q_{-++}^{n_{-++}} \sum_{d_-} \binom{t_-}{n_{+--}} Q_{+--}^{n_{+--}} Q_{--+}^{n_{--+}} \\
&e^{i\Lambda_n \gamma[(d_+ - d_- + \tau_+ - \tau_-)^q - ((\tau_+ - \tau_-) - (d_+ - d_-))^q]/n^{(q-1)/2}}, \tag{C.15}
\end{aligned}$$

where $\Lambda_n = \lambda_n/n^{(q-1)/2}$ as shorthand. Now, let us evaluate the sum over $\Delta_\pm$. We can use the following identity

$$2^{\tau_+} \sum_{\Delta_+} \binom{\tau_+}{n_{+++}} Q_{+++}^{n_{+++}} Q_{---}^{n_{---}} e^{(\zeta/n)\Delta_+} = (2Q_{+++} e^{\zeta/n} + 2Q_{---} e^{-\zeta/n})^{\tau_+}. \tag{C.16}$$

Applying this to the earlier sum, and note that $Q_{+++} = Q_{+-+}$ and $Q_{---} = Q_{-+-}$, we get:

$$\mathbb{E}_{\boldsymbol{Y}}[M_n(\zeta)] = \sum_{t=0}^n \binom{n}{t} e^{-\gamma^2[n^q-(n-2t)^q]/n^{q-1}} \sum_{t_++t_-=t} \binom{t}{t_+,t_-} \sum_{\tau_++\tau_-=n-t} \binom{n-t}{\tau_+,\tau_-}$$

$$\frac{1}{2^{n-t}}(2Q_{+++}e^{\zeta/n} + 2Q_{---}e^{-\zeta/n})^{\tau_+}(2Q_{+++}e^{-\zeta/n} + 2Q_{---}e^{\zeta/n})^{\tau_-}e^{(\zeta/n)(t_+-t_-)}$$

$$\sum_{d_+} \binom{t_+}{n_{++-}}Q_{++-}^{n_{++-}} Q_{-++}^{n_{-++}} \sum_{d_-} \binom{t_-}{n_{+--}}Q_{+--}^{n_{+--}} Q_{--+}^{n_{--+}}$$

$$e^{i\Lambda_n\gamma[(d_+-d_-+\tau_+-\tau_-)^q-((\tau_+-\tau_-)-(d_+-d_-))^q]/n^{(q-1)/2}}.$$

(C.17)

To further simplify the expression, we define $Z_{n,t}(k)$ as in Eq. (C.6), which we reproduce here:

$$Z_{n,t}(k) = \frac{1}{2^{n-t}} \sum_{\tau_++\tau_-=n-t} \binom{n-t}{\tau_+,\tau_-}(e^{\zeta/n}\cos^2\beta + e^{-\zeta/n}\sin^2\beta)^{\tau_+}(e^{-\zeta/n}\cos^2\beta + e^{\zeta/n}\sin^2\beta)^{\tau_-}$$

$$\times e^{i\Lambda_n\gamma[((\tau_+-\tau_-)+k)^q-((\tau_+-\tau_-)-k)^q]/n^{(q-1)/2}}.$$

(C.18)

Then we have

$$\mathbb{E}_{\boldsymbol{Y}}[M_n(\zeta)] = \sum_{t=0}^n \binom{n}{t} e^{-\gamma^2[n^q-(n-2t)^q]/n^{q-1}} \sum_{t_++t_-=t} \binom{t}{t_+,t_-} \sum_{d_+} \binom{t_+}{n_{++-}}Q_{++-}^{n_{++-}} Q_{-++}^{n_{-++}}$$

$$\times \sum_{d_-} \binom{t_-}{n_{+--}}Q_{+--}^{n_{+--}} Q_{--+}^{n_{--+}}e^{(\zeta/n)(t_+-t_-)} Z_{n,t}(d_+ - d_-).$$

(C.19)

Let $\Omega_t = \{-t, -t+1, \ldots, t-1, t\}$, and let $\{\hat{Z}_{n,t}(\xi)\}_{\xi\in\Omega_t}$ to be the discrete Fourier transform of $Z_{n,t}(k)$ as defined in Eq. (C.6), i.e.,

$$\hat{Z}_{n,t}(\xi) = (\mathcal{F}_t Z_{n,t})(\xi) = \sum_{k=-t}^t e^{-2\pi i\xi k/(2t+1)}Z_{n,t}(k),$$

(C.20)

By the property of Fourier transforms, we have

$$Z_{n,t}(k) = (\mathcal{F}_t^{-1}\hat{Z})(k) = \frac{1}{2t+1}\sum_{\xi=-t}^t e^{2\pi i\xi k/(2t+1)}\hat{Z}_{n,t}(\xi).$$

(C.21)

Plugging Eq. (C.21) into Eq. (C.19), we have

$$\mathbb{E}_{\boldsymbol{Y}}[M_n(\zeta)] = \sum_{t=0}^n \binom{n}{t} e^{-\gamma^2[n^q-(n-2t)^q]/n^{q-1}} \sum_{t_++t_-=t} \binom{t}{t_+,t_-} \sum_{d_+} \binom{t_+}{n_{++-}}Q_{++-}^{n_{++-}} Q_{-++}^{n_{-++}}$$

$$\times \sum_{d_-} \binom{t_-}{n_{+--}}Q_{+--}^{n_{+--}} Q_{--+}^{n_{--+}}e^{(\zeta/n)(t_+-t_-)}\frac{1}{2t+1}\sum_{\xi=-t}^t e^{2\pi i\xi(d_+-d_-)/(2t+1)}\hat{Z}_{n,t}(\xi)$$

(C.22)

$$\overset{(i)}{=} \sum_{t=0}^n \binom{n}{t} e^{-\gamma^2[n^q-(n-2t)^q]/n^{q-1}} \sum_{t_++t_-=t} \binom{t}{t_+,t_-}e^{(\zeta/n)(t_+-t_-)}$$

$$\times (-1)^{t_-}\cdot\frac{1}{2t+1}\sum_{\xi=-t}^t \left(2iQ_{++-}\sin(2\pi\xi/(2t+1))\right)^t\hat{Z}_{n,t}(\xi)$$

(C.23)

$$\overset{(ii)}{=} \sum_{t=0}^n \binom{n}{t} e^{-\gamma^2[n^q-(n-2t)^q]/n^{q-1}}(\sinh(\zeta/n)\sin(2\beta))^t$$

$$\times \frac{1}{2t+1}\sum_{\xi=-t}^t \left(\sin(2\pi\xi/(2t+1))\right)^t\hat{Z}_{n,t}(\xi),$$

(C.24)

where (i) used the equation

$$\sum_{d_+} \binom{t_+}{n_{++-}} Q_{++-}^{n_{++-}} Q_{-++}^{n_{-++}} e^{i\xi d_+} = \left(2iQ_{++-}\sin\xi\right)^{t_+},$$  (C.25)

and (ii) used the equation

$$\sum_{r+s=t} \binom{t}{r,s} (+1)^r(-1)^s \exp\{\zeta(r-s)\} = 2^t \sinh(\zeta)^t.$$  (C.26)

Note we have also used $4iQ_{++-} = \sin 2\beta$ to get rid of the two factors of $2^t$. This completes the proof of Lemma C.1.

### C.2.2 Proof of Lemma C.2

We first look at the limit of $Z_{n,t}(k)$ for fixed integer $-t \le k \le t$ (c.f. Eq. (C.6)). We denote $T_n = (e^{\zeta/n}\cos^2\beta + e^{-\zeta/n}\sin^2\beta)$, $U_n = (e^{-\zeta/n}\cos^2\beta + e^{\zeta/n}\sin^2\beta)$ and $G_n = (\tau_+ - \tau_-)/\sqrt{n}$ with $\tau_+ \sim \mathrm{Bin}(n-t, 1/2))$ to be a random variable. Then we have

$$Z_{n,t}(k) = \mathbb{E}_{G_n}\left[T_n^{\sqrt{n}G_n}(T_nU_n)^{(n-\sqrt{n}G_n)/2} e^{i\Lambda_n\gamma\sqrt{n}[(G_n+k/\sqrt{n})^q-(G_n-k/\sqrt{n})^q])}\right].$$  (C.27)

Note that we have $\lim_{n\to\infty} T_n^{\sqrt{n}} = \lim_{n\to\infty}(T_nU_n)^{-\sqrt{n}/2} = \lim_{n\to\infty}(T_nU_n)^{n/2} = 1$ and by assumption we have $\lim_{n\to\infty} \Lambda_n = \Lambda$. Furthermore, by central limit theorem, we have $G_n \to G \sim \mathcal{N}(0,1)$ so that for any fixed $-t \le k \le t$,

$$\sqrt{n}[(G_n + k/\sqrt{n})^q - (G_n - k/\sqrt{n})^q]) \xrightarrow{d} 2qkG^{q-1}.$$

This implies that

$$\lim_{n\to\infty} Z_{n,t}(k) = \mathbb{E}_{G\sim\mathcal{N}(0,1)}[e^{ik2q\Lambda\gamma G^{q-1}}] \equiv Z(k).$$

As a consequence, we have

$$\lim_{n\to\infty} E_{n,t} = \frac{1}{2t+1} \sum_{\xi=-t}^{t} \sin(2\pi\xi/(2t+1))^t \left(\sum_{k=-t}^{t} e^{-2\pi i\xi k/(2t+1)} \mathbb{E}_{G\sim\mathcal{N}(0,1)}[e^{ik2q\Lambda\gamma G^{q-1}}]\right).$$

Finally, by Lemma C.4 below and noting that $\sin(2\pi\xi/(2t+1))^t$ can be expressed as a degree $t$ polynomial of $(e^{2\pi i\xi/(2t+1)}, e^{-2\pi i\xi/(2t+1)})$, the right hand side of the equation above gives

$$\frac{1}{2t+1} \sum_{\xi=-t}^{t} \sin(2\pi\xi/(2t+1))^t \left(\sum_{k=-t}^{t} e^{-2\pi i\xi k/(2t+1)} \mathbb{E}_{G\sim\mathcal{N}(0,1)}[e^{ik2q\Lambda\gamma G^{q-1}}]\right) = \mathbb{E}_{G\sim\mathcal{N}(0,1)}[\sin(2q\Lambda\gamma G^{q-1})^t].$$

This proves Lemma C.2.

**Lemma C.4.** *Let $t \in \mathbb{Z}_{\ge 0}$ be an integer and let $\Omega_t = \{-t, -t+1, \ldots, t-1, t\}$. For a vector $(Z(k))_{k\in\Omega_t}$, we denote $\mathcal{F}_t : \mathbb{C}^{2t+1} \to \mathbb{C}^{2t+1}$ to be the discrete Fourier transform*

$$(\mathcal{F}_t Z)(\xi) \equiv \sum_{k=-t}^{t} e^{-2\pi i\xi k/(2t+1)} Z(k).$$

*Let $P : \mathbb{C}^2 \to \mathbb{C}$ be any fixed polynomials with degree less or equal to $t \in \mathbb{Z}_{\ge 0}$. Let $X$ be a real-valued random variable. Then we have*

$$\frac{1}{2t+1} \sum_{\xi=-t}^{t} \left[P(e^{2\pi i\xi/(2t+1)}, e^{-2\pi i\xi/(2t+1)})\left(\mathcal{F}_t\left(\mathbb{E}_X[e^{ikX}]\right)\right)(\xi)\right] = \mathbb{E}_X[P(e^{iX}, e^{-iX})].$$  (C.28)

*Proof of Lemma C.4.* By linearity of the expectation operator and the discrete Fourier transform operator, we just need to prove Eq. (C.28) for $P(e^{2\pi i\xi/(2t+1)}, e^{-2\pi i\xi/(2t+1)}) = e^{2\pi ip\xi/(2t+1)}$ for

some integer $-t \leq p \leq t$. Note that we have

$$\frac{1}{2t+1} \sum_{\xi=-t}^{t} \left[ e^{2\pi i p \xi / (2t+1)} \mathcal{F}_t \left( \mathbb{E}_X[e^{ikX}] \right) \right]$$

$$= \frac{1}{2t+1} \sum_{\xi=-t}^{t} \sum_{k=-t}^{t} e^{2\pi i p \xi / (2t+1)} e^{-2\pi i k \xi / (2t+1)} \mathbb{E}_X[e^{ikX}]$$

$$= \mathbb{E}_X \left[ \frac{1}{2t+1} \sum_{\xi=-t}^{t} \sum_{k=-t}^{t} e^{2\pi i (p-k)(\xi/(2t+1)-X/(2\pi))} e^{ipX} \right] = \mathbb{E}_X[e^{ipX}], \qquad (\text{C.29})$$

where the last equality used the fact that

$$\frac{1}{2t+1} \sum_{\xi=-t}^{t} \sum_{k=-t}^{t} e^{2\pi i (p-k)(\xi/(2t+1)-X/(2\pi))} = 1$$

for any integer $-t \leq p \leq t$ and any real $X$. This completes the proof of Lemma C.4. $\qquad \square$

### C.2.3 Proof of Lemma C.3

By the definition of $Z_{n,t}(k)$ as in Eq. (C.6), it is easy to see that

$$|Z_{n,t}(k)| \leq \frac{1}{2^{n-t}} \sum_{\tau_+ + \tau_- = n-t} \binom{n-t}{\tau_+, \tau_-} \left| e^{\zeta/n} \cos^2 \beta + e^{-\zeta/n} \sin^2 \beta \right|^{\tau_+} \left| e^{-\zeta/n} \cos^2 \beta + e^{\zeta/n} \sin^2 \beta \right|^{\tau_-}$$

$$\times \left| e^{i\Lambda_n \gamma [((\tau_+ - \tau_-)+k)^q - ((\tau_+ - \tau_-)-k)^q]/n^{(q-1)/2}} \right|$$

$$\leq \frac{1}{2^{n-t}} \sum_{\tau_+ + \tau_- = n-t} \binom{n-t}{\tau_+, \tau_-} e^{\tau_+ |\zeta|/n} e^{\tau_- |\zeta|/n} \cdot 1$$

$$= e^{(n-t)|\zeta|/n}$$

$$\leq e^{|\zeta|}. \qquad (\text{C.30})$$

As a consequence, we have $|\hat{Z}_{n,t}(\xi)| \leq (2t+1)e^{|\zeta|}$, which gives

$$|E_{n,t}| \leq (2t+1)e^{|\zeta|}.$$

As a consequence, by the definition of $I_{n,t}$ as in Eq. (C.7), we have

$$|I_{n,t}| \leq \frac{n^t}{t!} |\sinh(\zeta/n)|^t (2t+1)e^{|\zeta|}.$$

Note that when $\zeta/n \leq 1$, we have $|\sinh(\zeta/n)| \leq 6|\zeta|/n$. This gives

$$|I_{n,t}| \leq \frac{1}{t!} (6|\zeta|)^t (2t+1)e^{|\zeta|}.$$

This proves Lemma C.3.

### C.3 Proof of Theorem 1(a)

Theorem 1(a) is a combination of the two lemmas below.

**Lemma C.5.** *Take any sequence of $\{\beta_n\}_{n \geq 1} \subseteq [0, 2\pi]$, $\{\gamma_n\}_{n \geq 1} \subseteq \mathbb{R}$ with $\lim_{n \to \infty} \gamma_n = \infty$, and any sequence of $\{\lambda_n\}_{n \geq 1} \subseteq [0, \infty)$. We have*

$$\lim_{n \to \infty} \mathbb{E}_W[\langle \mathcal{R}_{\mathrm{QAOA}}^2 \rangle_{\gamma_n, \beta_n}] = 0. \qquad (\text{C.31})$$

**Lemma C.6.** *Take any sequence of $\{\beta_n\}_{n \geq 1} \subseteq [0, 2\pi]$, $\{\gamma_n\}_{n \geq 1} \subseteq \mathbb{R}$ with $\sup_n \gamma_n < \infty$, and any sequence of $\{\lambda_n\}_{n \geq 1} \subseteq [0, \infty)$ with $\lim_{n \to \infty} \lambda_n / n^{(q-1)/2} = 0$. We have*

$$\lim_{n \to \infty} \mathbb{E}_W[\langle \mathcal{R}_{\mathrm{QAOA}}^2 \rangle_{\gamma_n, \beta_n}] = 0. \qquad (\text{C.32})$$

### C.3.1 Proof of Lemma C.5

*Proof of Lemma C.5.* Denote $\Lambda_n = \lambda_n/n^{(q-1)/2}$. We can write

$$\mathbb{E}_{\boldsymbol{Y}}[\langle \mathcal{R}_{\mathrm{QAOA}}^2 \rangle_{\gamma_n,\beta_n}] = \frac{\partial^2}{\partial\zeta^2}\Big|_{\zeta=0} \mathbb{E}_{\boldsymbol{Y}}[M_n(\zeta;\gamma_n,\beta_n,\lambda_n)]. \tag{C.33}$$

Using Eq. (C.5), we can see that only a few terms depend on $\zeta$, whose derivative gives

$$\partial_\zeta^2\Big|_{\zeta=0}\Big[\big(\sinh(\zeta/n)\sin(2\beta)\big)^t\big(e^{\zeta/n}\cos^2(\beta)+e^{-\zeta/n}\sin^2(\beta)\big)^{\tau_+}\big(e^{-\zeta/n}\cos^2(\beta)+e^{\zeta/n}\sin^2(\beta)\big)^{\tau_-}\Big]$$

$$=\frac{\delta_{t=0}}{2n^2}\big(2t\sin(2\beta)+[(\tau_+-\tau_-)^2-(\tau_++\tau_-)]\cos(4\beta)+[(\tau_+-\tau_-)^2+(\tau_++\tau_-)]\big)$$

$$+\frac{\delta_{t=1}}{n^2}t(\tau_+-\tau_-)\sin(4\beta)$$

$$+\frac{\delta_{t=2}}{n^2}t(t-1)\sin^2(2\beta). \tag{C.34}$$

Hence, only the $t = 0, 1, 2$ terms survive, and we can write

$$\mathbb{E}_{\boldsymbol{Y}}[\langle \mathcal{R}_{\mathrm{QAOA}}^2 \rangle_{\gamma_n,\beta_n}] = T_0 + T_1 + T_2 \tag{C.35}$$

where

$$T_0 = \frac{1}{2^{n+1}n^2}\sum_{\tau_++\tau_-=n}\binom{n}{\tau_+,\tau_-}\big([(\tau_+-\tau_-)^2-(\tau_++\tau_-)]\cos(4\beta)+[(\tau_+-\tau_-)^2+(\tau_++\tau_-)]\big), \tag{C.36}$$

$$T_1 = \frac{\sin(4\beta_n)}{3n\cdot2^{n-1}}e^{-\gamma_n^2[n^q-(n-2)^q]/n^{q-1}}\sum_{\xi\in\{\pm1\}}\sin(2\pi\xi/3)\sum_{k=-1}^{1}e^{-2\pi i\xi k/3}$$

$$\times\sum_{\tau_++\tau_-=n-1}\binom{n-1}{\tau_+,\tau_-}(\tau_+-\tau_-)e^{i\Lambda_n\gamma_n[((\tau_+-\tau_-)+k)^q-((\tau_+-\tau_-)-k)^q]/n^{(q-1)/2}}, \tag{C.37}$$

$$T_2 = \frac{(n-1)\sin^2(2\beta_n)}{10n\cdot2^{n-2}}e^{-\gamma_n^2[n^q-(n-4)^q]/n^{q-1}}\sum_{\xi\in\{\pm1,\pm2\}}\sin^2(2\pi\xi/5)\sum_{k=-2}^{2}e^{-2\pi i\xi k/5}$$

$$\times\sum_{\tau_++\tau_-=n-2}\binom{n-2}{\tau_+,\tau_-}e^{i\Lambda_n\gamma_n[((\tau_+-\tau_-)+k)^q-((\tau_+-\tau_-)-k)^q]/n^{(q-1)/2}}. \tag{C.38}$$

Lemma C.5 then immediately follows from the Lemma C.7 below. $\qquad\square$

**Lemma C.7.** *For any $n \geq n_0$ for some large $n_0$, we have*

$$T_0 = (1+\cos(4\beta_n))/(2n),$$

$$|T_1| \leq 2\sin(4\beta_n)e^{-q\gamma_n^2},$$

$$|T_2| \leq 2\sin^2(2\beta_n)e^{-q\gamma_n^2}.$$

*Proof of Lemma C.7.* We can compute the first term directly as follows. Note that

$$\sum_{\tau_++\tau_-=n}\binom{n}{\tau_+,\tau_-}(\tau_+-\tau_-)^q = \frac{\partial^q}{\partial x^q}\Big|_{x=0}\sum_{\tau_++\tau_-=n}\binom{n}{\tau_+,\tau_-}e^{x(\tau_+-\tau_-)} \tag{C.39}$$

$$= \frac{\partial^q}{\partial x^q}\Big|_{x=0}\big(2\cosh(x)\big)^n. \tag{C.40}$$

In particular,

$$\sum_{\tau_++\tau_-=n}\binom{n}{\tau_+,\tau_-}(\tau_+-\tau_-) = 0, \tag{C.41}$$

$$\sum_{\tau_++\tau_-=n}\binom{n}{\tau_+,\tau_-}(\tau_+-\tau_-)^2 = 2^n n. \tag{C.42}$$

It follows that

$$T_0 = \frac{1}{2^{n+1}n^2}\left((2^n n - 0)\cos(4\beta) + (2^n n + 0)\right) = \frac{\cos(4\beta) + 1}{2n}. \tag{C.43}$$

For the remaining terms, upper bounds suffice:

$$|T_1| \le \frac{\sin(4\beta_n)}{3n \cdot 2^{n-1}} e^{-\gamma_n^2[n^q - (n-2)^q]/n^{q-1}} \sum_{\xi \in \{\pm 1\}} 1 \cdot \sum_{k=-1}^{1} 1 \cdot \sum_{\tau_+ + \tau_- = n-1} \binom{n-1}{\tau_+, \tau_-} \cdot (n-1) \cdot 1$$

$$\le \frac{\sin(4\beta_n)}{3} e^{-q\gamma_n^2} \cdot 2 \cdot 3$$

$$= 2\sin(4\beta_n)e^{-q\gamma_n^2}, \tag{C.44}$$

and

$$|T_2| \le \frac{(n-1)\sin^2(2\beta_n)}{10n \cdot 2^{n-2}} e^{-q\gamma_n^2} \sum_{\xi \in \{\pm 1, \pm 2\}} 1 \cdot \sum_{k=-2}^{2} 1 \cdot \sum_{\tau_+ + \tau_- = n-2} \binom{n-2}{\tau_+, \tau_-} \cdot 1$$

$$\le \frac{\sin^2(2\beta_n)}{10} e^{-q\gamma_n^2} \cdot 4 \cdot 5$$

$$= 2\sin^2(2\beta_n)e^{-q\gamma_n^2}. \tag{C.45}$$

This finishes the proof of Lemma C.7. $\qquad\square$

### C.3.2 Proof of Lemma C.6

Lemma C.6 follows from Theorem 1(b).

## D Derivation for general $p$-step QAOA (Claim 3.7)

### D.1 Sketch of derivation ideas

We now briefly sketch some ideas behind the derivation for Claim 3.7 that characterizes the overlap distribution of the $p$-step QAOA when the SNR ratio scales as in Eq. (3.10). Similar to Theorem 1, our approach is to evaluate the moment-generating function of the QAOA overlap in the $n \to \infty$ limit. As evident in the proof of Theorem 1, as well as in previous analyses of the QAOA applied to spin-glass models [24, 26, 28], the key technical difficulty is handling a "generalized multinomial sum" of the following form:

$$S = \sum_{m_j \ge 0, \ \sum_j m_j = n} \binom{n}{\{m_j\}} \left(\prod_j Q_j^{m_j}\right) \exp[P(\boldsymbol{m})], \tag{D.1}$$

where $P(\boldsymbol{m})$ is a polynomial over entries of $\boldsymbol{m} = (m_j)_j$ with degree $q$. Note the above summation has no analytical simplification when $P$ is not a linear polynomial ($q > 1$). Previous works have evaluated this sum in the $n \to \infty$ limit either by proving a "generalized multinomial theorem" that exploits combinatorial structures of the polynomial $P$ [24, 26], or by employing a Gaussian integration trick and the saddle-point method when $q = 2^\ell$ [28]. However, neither approach is sufficient for the spiked tensor model that we study in the present paper.

Instead, we develop an alternative approach based on the Fourier transform to linearize exponents in the summands. The idea is to replace $\boldsymbol{m}$ with continuous variables $\boldsymbol{\mu}$ via Dirac delta functions, which after Fourier transforms yield exponents that are linear in $\boldsymbol{m}$, enabling us to analytically evaluate the multinomial sum over $\boldsymbol{m}$ as follows:

$$S = \int d\boldsymbol{\mu} \int d\hat{\boldsymbol{\mu}} \sum_{m_j \ge 0, \ \sum_j m_j = n} \binom{n}{\{m_j\}} \left(\prod_j Q_j^{m_j}\right) \exp[P(\boldsymbol{\mu})] e^{i\hat{\boldsymbol{\mu}} \cdot (\boldsymbol{m} - \boldsymbol{\mu})}$$

$$= \int d\boldsymbol{\mu} \int d\hat{\boldsymbol{\mu}} \left(\sum_j Q_j e^{i\hat{\mu}_j}\right)^n e^{P(\boldsymbol{\mu}) - i\hat{\boldsymbol{\mu}} \cdot \boldsymbol{\mu}}. \tag{D.2}$$

See Appendix D.4 for more details. This is a powerful approach to replace the cumbersome multi-nomial sums with simpler integrals. However, it is difficult to make such manipulations involving Dirac delta functions rigorous, which we leave open as future work. Nevertheless, we proceed with the heuristic derivation in the current paper: by writing the variables $(m_j)_j$ in an alternative basis and rescaling them cleverly, we are able to evaluate the integrals to obtain the moment-generating function in the $n \to \infty$ limit.

## D.2  Organizing the finite $n$ sum

Our goal is to evaluate the moment-generating function of the overlap with signal, $M_n(\zeta) = \langle \gamma, \beta | \exp(\zeta \widehat{\mathcal{R}}) | \gamma, \beta \rangle$, for the general $p$-step QAOA. Using the same method as in the $p = 1$ case, we can show that the disorder-averaged moment-generating function can be written as the following combinatorial sum:

$$\mathbb{E}_{\boldsymbol{Y}}[M_n(\zeta)] = \sum_{\{n_{\boldsymbol{a}}\}} \binom{n}{\{n_{\boldsymbol{a}}\}} \prod_{\boldsymbol{a} \in B} Q_{\boldsymbol{a}}^{n_{\boldsymbol{a}}} \exp\left[\mathcal{A} + i\lambda_n \mathcal{B} + \zeta \mathcal{C}\right], \tag{D.3}$$

where

$$\mathcal{A} = -\frac{1}{2n^{q-1}} \sum_{\underline{\boldsymbol{a}} \in B^q} \Phi_{\underline{\boldsymbol{a}}}^2 \prod_{s=1}^q n_{\boldsymbol{a}_s}, \qquad \mathcal{B} = \frac{1}{n^{q-1}} \sum_{\underline{\boldsymbol{a}} \in B^q} \Phi_{\underline{\boldsymbol{a}}} \prod_{s=1}^q (\boldsymbol{a}_s)_{\mathrm{m}} n_{\boldsymbol{a}_s}, \qquad \mathcal{C} = \frac{1}{n} \sum_{\boldsymbol{v} \in B} v_{\mathrm{m}} n_{\boldsymbol{v}}, \tag{D.4}$$

and

$$\begin{aligned}
B &= \{(a_1, a_2, \ldots, a_p, a_{\mathrm{m}}, a_{-p}, \ldots, a_{-1}) : a_j \in \{\pm 1\}\}, \\
Q_{\boldsymbol{a}} &= \tfrac{1}{2} \prod_{r=1}^p (\cos \beta_r)^{1+(a_r+a_{-r})/2} (\sin \beta_r)^{1-(a_r+a_{-r})/2} (i)^{(a_{-r}-a_r)/2}, \\
\Phi_{\boldsymbol{a}} &= \sum_{r=1}^p \gamma_r (a_r a_{r+1} \cdots a_p - a_{-p} \cdots a_{-r-1} a_{-r}), \\
\Phi_{\underline{\boldsymbol{a}}} &= \Phi_{\boldsymbol{a}_1 \boldsymbol{a}_2 \cdots \boldsymbol{a}_q}.
\end{aligned} \tag{D.5}$$

Note $Q_{\boldsymbol{a}}$ and $\Phi_{\boldsymbol{a}}$ are independent of $a_{\mathrm{m}}$.

This is a straightforward generalization of the proof in Appendix B, where we insert $2p+1$ resolutions of the identity instead of 3. This also closely follows the derivation in Ref. [26, Appendix D.2]. $B, Q_{\boldsymbol{a}}, \Phi_{\boldsymbol{a}}$ are also generalizations of the same quantities in Appendix B for $p > 1$.

Define the rank function

$$\ell(\boldsymbol{a}) = \max(\{i : a_{-i} \neq a_i\} \cup \{0\}). \tag{D.6}$$

**A canonical basis.**  We next perform further simplifications that remove the explicit dependence on $a_{\mathrm{m}}$. First we define the set of $2p$-bit strings as

$$A = \{(a_1, a_2, \ldots, a_p, a_{-p}, \ldots, a_{-1}) : a_j \in \{\pm 1\}\},$$

and define $A_0$ and $D$ according to a similar convention as that in [26] as follows:

$$\begin{aligned}
A_0 &:= \{\boldsymbol{a} \in A : \ell(\boldsymbol{a}) = 0\} = \{\boldsymbol{a} \in A : a_{-k} = a_k \text{ for } 1 \leq k \leq p\}, \\
D &:= \left\{\boldsymbol{a} \in A : \ell(\boldsymbol{a}) > 0 \quad \text{and} \quad \textstyle\prod_{j=1}^p a_j = +1\right\}. 
\end{aligned} \tag{D.7}$$

Given the rank function in Eq. (D.6), we can define an ordering on $D$, which we borrow from [26]. For any two distinct element $\boldsymbol{a}_1, \boldsymbol{a}_2 \in D$, we define the $\prec$ relation as following: (1) If $\ell(\boldsymbol{a}_1) < \ell(\boldsymbol{a}_2)$, we let $\boldsymbol{a}_1 \prec \boldsymbol{a}_2$; (2) If $\ell(\boldsymbol{a}_1) > \ell(\boldsymbol{a}_2)$, we let $\boldsymbol{a}_2 \prec \boldsymbol{a}_1$; (3) If $\ell(\boldsymbol{a}_1) = \ell(\boldsymbol{a}_2)$ and if $\boldsymbol{a}_1$ is lexically less than $\boldsymbol{a}_2$, we let $\boldsymbol{a}_1 \prec \boldsymbol{a}_2$; (3) If $\ell(\boldsymbol{a}_1) = \ell(\boldsymbol{a}_2)$ and if $\boldsymbol{a}_1$ is lexically greater than $\boldsymbol{a}_2$, we let $\boldsymbol{a}_2 \prec \boldsymbol{a}_1$ (here lexical order means that, for example, $(-1, -1), (-1, 1), (1, -1), (1, 1)$ are in lexically increasing order). It is easy to see that such $\prec$ relation is a full order, so that we can also define $\preceq, \succeq,$ and $\succ$ accordingly.

For any $\boldsymbol{a} \in A$, we define

$$n_{\boldsymbol{a}\pm} = n_{\boldsymbol{b}} \quad \text{where} \quad \boldsymbol{b} = (a_1, \ldots, a_p, \pm 1, a_{-p}, \ldots, a_{-1}). \tag{D.8}$$

Let

$$t_{\boldsymbol{a}+} = n_{\boldsymbol{a}+} + n_{\bar{\boldsymbol{a}}+}, \qquad t_{\boldsymbol{a}-} = n_{\boldsymbol{a}-} + n_{\bar{\boldsymbol{a}}-}, \qquad \forall \boldsymbol{a} \in D,$$
$$d_{\boldsymbol{a}+} = n_{\boldsymbol{a}+} - n_{\bar{\boldsymbol{a}}+}, \qquad d_{\boldsymbol{a}-} = n_{\boldsymbol{a}-} - n_{\bar{\boldsymbol{a}}-}, \qquad \forall \boldsymbol{a} \in D, \tag{D.9}$$
$$n_{\boldsymbol{a}} = n_{\boldsymbol{a}+} + n_{\boldsymbol{a}-}, \qquad \delta n_{\boldsymbol{a}} = n_{\boldsymbol{a}+} - n_{\boldsymbol{a}-}, \qquad \forall \boldsymbol{a} \in A_0.$$

Furthermore, $\forall \boldsymbol{a} \in D$, let

$$t_{\boldsymbol{a}} = t_{\boldsymbol{a}+} + t_{\boldsymbol{a}-}, \qquad d_{\boldsymbol{a}} = d_{\boldsymbol{a}+} + d_{\boldsymbol{a}-},$$
$$\delta t_{\boldsymbol{a}} = t_{\boldsymbol{a}+} - t_{\boldsymbol{a}-}, \qquad \delta d_{\boldsymbol{a}} = d_{\boldsymbol{a}+} - d_{\boldsymbol{a}-}. \tag{D.10}$$

Observe that these new variables constitute a basis transformation via

$$\{n_{\boldsymbol{a}}\}_{\boldsymbol{a} \in B} \equiv \{t_{\boldsymbol{a}\pm}, d_{\boldsymbol{a}\pm}\}_{\boldsymbol{a} \in D} \cup \{n_{\boldsymbol{a}}, \delta n_{\boldsymbol{a}}\}_{\boldsymbol{a} \in A_0}$$
$$\equiv \{t_{\boldsymbol{a}}, \delta t_{\boldsymbol{a}}, d_{\boldsymbol{a}}, \delta d_{\boldsymbol{a}}\}_{\boldsymbol{a} \in D} \cup \{n_{\boldsymbol{a}}, \delta n_{\boldsymbol{a}}\}_{\boldsymbol{a} \in A_0}. \tag{D.11}$$

We will call the last line as the "canonical basis". As a side note, comparing to the $p = 1$ derivation in Eq. (C.6), we have $k = \delta d_{+-}$ and $\tau_+ - \tau_- = \delta n_{++} - \delta n_{--}$.

In what follows, we will convert all our expressions into the canonical basis. It is also helpful to denote the shorthand

$$t = \sum_{\boldsymbol{a} \in D} t_{\boldsymbol{a}}, \qquad \text{and thus} \quad n - t = \sum_{\boldsymbol{a} \in A_0} n_{\boldsymbol{a}}. \tag{D.12}$$

In this basis, we can rewrite (D.3) as

$$\mathbb{E}_{\boldsymbol{Y}}[M_n(\zeta)] = \sum_{t=0}^{n} \binom{n}{t} \sum_{\{n_{\boldsymbol{a}}\}_{\boldsymbol{a} \in A_0}} \binom{n-t}{\{n_{\boldsymbol{a}}\}} \prod_{\boldsymbol{a} \in A_0} \left\{ Q_{\boldsymbol{a}}^{n_{\boldsymbol{a}}} \sum_{\delta n_{\boldsymbol{a}}} \binom{n_{\boldsymbol{a}}}{n_{\boldsymbol{a}+}} \right\}$$
$$\times \sum_{\{t_{\boldsymbol{a}}\}_{\boldsymbol{a} \in D}} \binom{t}{\{t_{\boldsymbol{a}}\}} \prod_{\boldsymbol{a} \in D} \oiint_{d_{\boldsymbol{a}}, \delta d_{\boldsymbol{a}}, \delta t_{\boldsymbol{a}}}^{t_{\boldsymbol{a}}} \exp\left[ \mathcal{A} + i\lambda_n \mathcal{B} + \zeta \mathcal{C} \right]. \tag{D.13}$$

where we have used the fact that $Q_{\boldsymbol{a}\pm} = Q_{\boldsymbol{a}}$ does not depend on $a_{\mathrm{m}}$ (here we also slightly abused notation allowing $Q_{\boldsymbol{a}}$ to take $\boldsymbol{a} \in A$ as argument). Here we also define, for any $\boldsymbol{a} \in D$ and $t_{\boldsymbol{a}} \in \mathbb{Z}_{\geq 0}$, the **little-sum** operator on functions of $(d_{\boldsymbol{a}}, \delta d_{\boldsymbol{a}}, \delta t_{\boldsymbol{a}})$ as

$$\oiint_{d_{\boldsymbol{a}}, \delta d_{\boldsymbol{a}}, \delta t_{\boldsymbol{a}}}^{t_{\boldsymbol{a}}} (\cdots) := \sum_{t_{\boldsymbol{a}+}, t_{\boldsymbol{a}-}} \binom{t_{\boldsymbol{a}}}{t_{\boldsymbol{a}+}, t_{\boldsymbol{a}-}} \sum_{d_{\boldsymbol{a}+}} \binom{t_{\boldsymbol{a}+}}{n_{\boldsymbol{a}+}} Q_{\boldsymbol{a}}^{n_{\boldsymbol{a}+}} Q_{\bar{\boldsymbol{a}}}^{n_{\bar{\boldsymbol{a}}+}} \sum_{d_{\boldsymbol{a}-}} \binom{t_{\boldsymbol{a}-}}{n_{\boldsymbol{a}-}} Q_{\boldsymbol{a}}^{n_{\boldsymbol{a}-}} Q_{\bar{\boldsymbol{a}}}^{n_{\bar{\boldsymbol{a}}-}} (\cdots). \tag{D.14}$$

Now let us rewrite $\mathcal{B}$ in the canonical basis, and we will show that it is purely a function of $\{\delta d_{\boldsymbol{a}}, \delta t_{\boldsymbol{a}}\}_{\boldsymbol{a} \in D} \cup \{\delta n_{\boldsymbol{c}}\}_{\boldsymbol{c} \in A_0}$. Observe that

$$n^{q-1} \mathcal{B} = \sum_{r=1}^{p} \gamma_r \left[ \left( B_r^+ \right)^q - \left( B_r^- \right)^q \right], \quad \text{where} \quad B_r^{\pm} = \sum_{\boldsymbol{a} \in B} a_{\pm r}^* a_{\mathrm{m}} n_{\boldsymbol{a}}, \tag{D.15}$$

and we have denoted $a_r^* = a_r \cdots a_p$ for any $1 \leq r \leq p$. Note $a_r^* - a_{-r}^* \neq 0$ only if $\ell(\boldsymbol{a}) \geq r$. Hence, we have

$$B_r^+ = \sum_{\boldsymbol{a} \in A_0} a_r^* \delta n_{\boldsymbol{a}} + \sum_{\boldsymbol{a} \in D, \ell(\boldsymbol{a}) \leq r-1} a_r^* \delta t_{\boldsymbol{a}} + \sum_{\boldsymbol{a} \in D, \ell(\boldsymbol{a}) \geq r} a_r^* \delta d_{\boldsymbol{a}},$$
$$B_r^- = \sum_{\boldsymbol{a} \in A_0} a_r^* \delta n_{\boldsymbol{a}} + \sum_{\boldsymbol{a} \in D, \ell(\boldsymbol{a}) \leq r-1} a_r^* \delta t_{\boldsymbol{a}} + \sum_{\boldsymbol{a} \in D, \ell(\boldsymbol{a}) \geq r} a_{-r}^* \delta d_{\boldsymbol{a}}.$$

To reveal additional structures of $\mathcal{B}$, we write

$$n^{q-1} \mathcal{B} = \sum_{r=1}^{p} \gamma_r [(R_r + L_r)^q - (R_r - L_r)^q] = \sum_{r=1}^{p} \gamma_r [2q L_r R_r^{q-1} + 2\binom{q}{3} L_r^3 R_r^{q-3} + \cdots] \tag{D.16}$$

where we have defined

$$L_r = \frac{1}{2}(B_r^+ - B_r^-) = \sum_{\boldsymbol{a} \in D, \ell(\boldsymbol{a}) \geq r} \frac{1}{2}(a_r^* - a_{-r}^*)\delta d_{\boldsymbol{a}}, \tag{D.17}$$

$$R_r = \frac{1}{2}(B_r^+ + B_r^-) = \sum_{\boldsymbol{a} \in A_0} a_r^* \delta n_{\boldsymbol{a}} + \sum_{\boldsymbol{a} \in D, \ell(\boldsymbol{a}) \leq r-1} a_r^* \delta t_{\boldsymbol{a}} + \sum_{\boldsymbol{a} \in D, \ell(\boldsymbol{a}) \geq r} \frac{1}{2}(a_r^* + a_{-r}^*)\delta d_{\boldsymbol{a}}. \tag{D.18}$$

We note here that $\mathcal{B}$ consists of terms that have at least one power of the $\{\delta d_{\boldsymbol{a}}\}_{\boldsymbol{a} \in D}$ variables through the dependence on $L_r$, which is a fact that will become important later.

Proceeding in the same way for $\mathcal{A}$ and $\mathcal{C}$, we can also write them in the canonical basis. We note $\mathcal{A}$ is a polynomial that has appeared in [26], where it can be shown to only depend on $\{t_{\boldsymbol{a}}, d_{\boldsymbol{a}}\}_{\boldsymbol{a} \in D} \cup \{n_{\boldsymbol{c}}\}_{\boldsymbol{c} \in A_0}$. In summary, we note the dependence of $\mathcal{A}, \mathcal{B}, \mathcal{C}$ on the canonical basis variables is as follows:

$$\mathcal{A} = \mathcal{A}\Big(\{t_{\boldsymbol{a}}\}_{\boldsymbol{a} \in D}, \{d_{\boldsymbol{a}}\}_{\boldsymbol{a} \in D}, \{n_{\boldsymbol{c}}\}_{\boldsymbol{c} \in A_0}\Big),$$
$$i\lambda_n \mathcal{B} = i\lambda_n \mathcal{B}\big(\{\delta d_{\boldsymbol{a}}, \delta t_{\boldsymbol{a}}\}_{\boldsymbol{a} \in D} \cup \{\delta n_{\boldsymbol{c}}\}_{\boldsymbol{c}, \in A_0}\big), \tag{D.19}$$
$$\mathcal{C} = \frac{1}{n}\Big(\sum_{\boldsymbol{a} \in A_0} \delta n_{\boldsymbol{a}} + \sum_{\boldsymbol{a} \in D} \delta t_{\boldsymbol{a}}\Big).$$

**Operator shorthands for different parts of the sum.** To streamline notations, we now introduce three operators $\mathbb{T}, \mathbb{S}, \mathbb{U}$ as shorthands for different parts of the sum that appear in Eq. (D.13).

Let us define the $\mathbb{T}_n^t$ operator acting on a function $f(\{t_{\boldsymbol{a}} : \boldsymbol{a} \in D\})$ as

$$\mathbb{T}_n^t f = \frac{t!}{n^t}\binom{n}{t} \sum_{t_{\boldsymbol{a}} \geq 0, \forall \boldsymbol{a} \in D, \sum_{\boldsymbol{a}} t_{\boldsymbol{a}} = t} f(\{t_{\boldsymbol{a}}\}). \tag{D.20}$$

Next, let us define the operator $\mathbb{S}_n^{\{t_{\boldsymbol{a}}\}}$ acting on any function $g(\{d_{\boldsymbol{a}}, \delta d_{\boldsymbol{a}}, \delta t_{\boldsymbol{a}}\}_{\boldsymbol{a} \in D})$ as follows:

$$\mathbb{S}_n^{\{t_{\boldsymbol{a}}\}} g = \prod_{\boldsymbol{a} \in D} \left\{ \frac{n^{t_{\boldsymbol{a}}}}{t_{\boldsymbol{a}}!} \oint_{d_{\boldsymbol{a}}, \delta d_{\boldsymbol{a}}, \delta t_{\boldsymbol{a}}}^{t_{\boldsymbol{a}}} \right\} g$$

$$= \prod_{\boldsymbol{a} \in D} \left\{ \frac{(nQ_{\boldsymbol{a}})^{t_{\boldsymbol{a}}}}{t_{\boldsymbol{a}}!} \sum_{t_{\boldsymbol{a}+}, t_{\boldsymbol{a}-}} \binom{t_{\boldsymbol{a}}}{t_{\boldsymbol{a}+}, t_{\boldsymbol{a}-}} \sum_{d_{\boldsymbol{a}+}} \binom{t_{\boldsymbol{a}+}}{n_{\boldsymbol{a}+}} (+1)^{n_{\boldsymbol{a}+}} (-1)^{n_{\bar{\boldsymbol{a}}+}} \right.$$
$$\left. \sum_{d_{\boldsymbol{a}-}} \binom{t_{\boldsymbol{a}-}}{n_{\boldsymbol{a}-}} (+1)^{n_{\boldsymbol{a}-}} (-1)^{n_{\bar{\boldsymbol{a}}-}} \right\} g. \tag{D.21}$$

Note $S_n^{\{t_{\boldsymbol{a}}\}} 1 = \mathbb{1}\{t_{\boldsymbol{a}} = 0 \; \forall \boldsymbol{a} \in D\}$.

Lastly, we define the $\mathbb{U}_n^t$ operator acting on a function $h(\{n_{\boldsymbol{c}}/n, \delta n_{\boldsymbol{c}}/\sqrt{n} : \boldsymbol{c} \in A_0\})$ as

$$\mathbb{U}_n^t h = \sum_{\{n_{\boldsymbol{a}}\}_{\boldsymbol{a} \in A_0}} \binom{n-t}{\{n_{\boldsymbol{a}}\}} \prod_{\boldsymbol{a} \in A_0} \left\{ Q_{\boldsymbol{a}}^{n_{\boldsymbol{a}}} \sum_{\delta n_{\boldsymbol{a}}} \binom{n_{\boldsymbol{a}}}{n_{\boldsymbol{a}+}} \right\} h. \tag{D.22}$$

With these summing operators defined, we can rewrite (D.13) as

$$\mathbb{E}_{\boldsymbol{Y}}[M_n(\zeta)] = \sum_{t=0}^{n} e_n(t), \tag{D.23}$$

where

$$e_n(t) = \mathbb{U}_n^t \mathbb{T}_n^t \mathbb{S}_n^{\{t_{\boldsymbol{a}}\}}[\exp(\mathcal{A} + i\lambda_n \mathcal{B} + \zeta \mathcal{C})]. \tag{D.24}$$

### D.3 Rescaling the summand for the $n \to \infty$ limit

In the $n \to \infty$ limit, we want to rescale the canonical basis variables $\{t_a, \delta t_a, d_a, \delta d_a\}_{a \in D} \cup \{n_c, \delta n_c\}_{c \in A_0}$ so that the summing operators $(\mathbb{U}_n^t, \mathbb{T}_n^t, \mathbb{S}_n^{\{t_a\}})$ and the summand $\mathcal{A} + i\lambda_n \mathcal{B} + \zeta\mathcal{C}$ converge to simplified forms. To this end, for all $a, b \in D$, $c \in A_0$, we will rescale by defining

$$
\begin{aligned}
t_a &= \tau_a, & \delta t_a/n^{\rho_a} &= \delta\tau_a, \\
d_b/n &= \eta_b, & \delta d_b/n^{1-\rho_b} &= \delta\eta_b \\
n_c/n &= \omega_c, & \delta n_c/\sqrt{n} &= \delta\omega_c,
\end{aligned}
\tag{D.25}
$$

where $(\tau_a, \eta_b, \omega_c, \delta\tau_a, \delta\eta_b, \delta\omega_c)$ are new dimensionless variables that will be integrated over, and $\rho_a$ are scaling exponents which we will define shortly.

The goal of this subsection is to derive the summand in the $n \to \infty$ limit. Specifically, we consider the summand broken into two parts, each as a polynomial of a distinct subset of the rescaled variables as follows:

$$
\Gamma_n(\{t_a, \eta_b, \omega_c\}) := \mathcal{A}(\{t_a, \eta_b n, \omega_c n\}), \tag{D.26}
$$
$$
\Xi_n(\{\delta\tau_a, \delta\eta_b, \delta\omega_c\}) := i\lambda_n \mathcal{B}(\{\delta\tau_a n^{\rho_a}, \delta\eta_b n^{1-\rho_b}, \delta\omega_c\sqrt{n}\}) + \zeta\mathcal{C}(\{\delta\tau_a n^{\rho_a}, \delta\omega_c\sqrt{n}\}), \tag{D.27}
$$

where the subscripts in the arguments implicitly iterate over $a, b \in D$ and $c \in A_0$. We think of $\Gamma_n$ and $\Xi_n$ as polynomials in their arguments, whose coefficients can depend on $n$.

First, we know from [26, Lemma D.2] that with the rescaling specified in Eq. (D.25) and $\gamma_j, \beta_j = \Theta(1)$, we have

$$
\lim_{n \to \infty} \Gamma_n(\{t_a, \eta_b, \omega_c\}_{a,b \in D, c \in A_0}) = \sum_{a \in D} t_a P_a(\{\eta_b\}_{b \prec a}, \{\omega_c\}_{c \in A_0}) =: \Gamma. \tag{D.28}
$$

For the rest of this subsection, we derive the limit of $\Xi_n = i\lambda_n \mathcal{B} + \zeta\mathcal{C}$.

**Choosing the scaling exponents $\rho_a$.** We want to choose the scaling exponents for $(\delta d_a, \delta t_a)$ variables, such that all the terms of $\mathcal{B}$ except those are linear in $\delta d_a$ vanish in the $n \to \infty$ limit. This would then imply the polynomial $\Xi_n$ in the limit would only be at most linear in $\delta\eta_a$, which is very helpful later for evaluating certain integrals as we shall see in Eq. (D.61).

In the general $p$-step QAOA applied to the spiked $q$-tensor model, suppose the SNR parameter $\lambda$ has a scaling as follows

$$
\lambda_n = \Lambda n^{c(p,q)}, \tag{D.29}
$$

where $c(p, q)$ is to be determined. Also suppose that the appropriate scaling for $\delta d_a$ and $\delta t_a$ are

$$
\delta d_a \sim n^{1-\rho_{\ell(a)}}, \quad \delta t_a \sim n^{\rho_{\ell(a)}}, \tag{D.30}
$$

so that they only depend on the rank $\ell(a)$ of $a$. Based on the explicit derivation at $p = 1$, we believe we only care about the terms in $\mathcal{B}$ that look like $\delta d_a \delta n_b^{q-1}$ when $\ell(a) = 1$ and $\ell(b) = 0$, or $\delta d_a \delta t_b^{q-1}$ when $\ell(a) = \ell$ and $\ell(b) = \ell - 1 > 0$. Also recall that $\delta n_b \sim \sqrt{n}$ for $b \in A_0$ from (D.25). For these terms in $\mathcal{B}$, we have

$$
\frac{\lambda_n}{n^{q-1}} \delta d_a \delta n_b^{q-1} \sim n^{c(p,q)+1-\rho_1+(q-1)/2-(q-1)}, \tag{D.31}
$$

$$
\frac{\lambda_n}{n^{q-1}} \delta d_a \delta t_b^{q-1} \sim n^{c(p,q)+1-\rho_\ell+(q-1)\rho_{\ell-1}-(q-1)}. \tag{D.32}
$$

To ensure that all such terms in $\mathcal{B}$ are order 1, we impose the condition that

$$
c(p,q) + 1 - \rho_\ell + (q-1)(\rho_{\ell-1} - 1) = 0, \quad \text{and} \quad \rho_0 = \frac{1}{2}. \tag{D.33}
$$

Solving this recurrence equation, we get that

$$
\rho_\ell = 1 - \frac{(q-1)^\ell}{2} + c(p,q)\frac{(q-1)^\ell - 1}{q - 2}. \tag{D.34}
$$

If we impose the additional condition that $\rho_p = 1$ (so that $\delta t_{\boldsymbol{a}}/n = \Theta(1)$ to yield a nonvanishing overlap in $\mathcal{C}$), this implies that the SNR scaling needs to be

$$c(p,q) = \frac{q-2}{2}\frac{(q-1)^p}{(q-1)^p - 1} = \frac{q-2}{2} + \frac{q-2}{2[(q-1)^p - 1]}. \tag{D.35}$$

Plugging this into Eq. (D.34), we get

$$\rho_\ell = \frac{1}{2}\frac{(q-1)^p + (q-1)^\ell - 2}{(q-1)^p - 1}. \tag{D.36}$$

For the special case of $q = 2$, we have $c(p,2) = \frac{1}{2p}$, and $\rho_\ell = \frac{1}{2} + \frac{\ell}{2p}$.

Note $1/2 \leq \rho_\ell \leq 1$ since $1 \leq (q-1)^\ell \leq (q-1)^p$ and $0 \leq \ell \leq p$. This means $\delta d_{\boldsymbol{a}} = O(n^{1/2})$ and $\delta t_{\boldsymbol{a}} = \Omega(n^{1/2})$. Another property to note is that $\rho_\ell$ is monotonically increasing with $\ell$. In particular, $\rho_0 = 1/2$ and $\rho_p = 1$.

**The limiting expression for $\Xi_n$.** To get the limiting polynomial for $\Xi_n = i\lambda_n\mathcal{B}+\zeta\mathcal{C}$, we substitute $\delta d_{\boldsymbol{a}} = \delta\eta_{\boldsymbol{a}}n^{1-\rho_{\boldsymbol{a}}}$, $\delta t_{\boldsymbol{a}} = \delta\tau_{\boldsymbol{a}}n^{\rho_{\boldsymbol{a}}}$, and $\delta n_{\boldsymbol{c}} = \delta\omega_{\boldsymbol{c}}\sqrt{n}$, and take the $n \to \infty$ limit. We first consider $\mathcal{B}$ as written in Eq. (D.16). In terms of the rescaled dimensionless variables, we have

$$L_r = \sum_{\boldsymbol{a}\in D,\ell(\boldsymbol{a})\geq r} \frac{1}{2}(a_r^* - a_{-r}^*)\delta\eta_{\boldsymbol{a}}n^{1-\rho_{\boldsymbol{a}}},$$

$$R_r = \frac{1}{2}(B_r^+ + B_r^-) = \sum_{\boldsymbol{a}\in A_0} a_r^*\delta\omega_{\boldsymbol{a}}\sqrt{n} + \sum_{\boldsymbol{a}\in D,\ell(\boldsymbol{a})\leq r-1} a_r^*\delta\tau_{\boldsymbol{a}}n^{\rho_{\boldsymbol{a}}} + \sum_{\boldsymbol{a}\in D,\ell(\boldsymbol{a})\geq r} \frac{1}{2}(a_r^* + a_{-r}^*)\delta\eta_{\boldsymbol{a}}n^{1-\rho_{\boldsymbol{a}}}.$$

With the exponents defined in Eq. (D.36), we note that $L_r$ is dominated by $\{\delta\eta_{\boldsymbol{a}} : \ell(\boldsymbol{a}) = r\}$, and $R_r$ is dominated by $\{\delta\omega_{\boldsymbol{a}} : \boldsymbol{a} \in A_0\}$ when $r = 1$ and $\{\delta\tau_{\boldsymbol{a}} : \ell(\boldsymbol{a}) = r-1\}$ when $r > 1$. Thus, the appropriately rescaled $L_r$ and $R_r$ in the limit are

$$\tilde{L}_r := \lim_{n\to\infty}\frac{L_r}{n^{1-\rho_r}} = \sum_{\boldsymbol{a}\in D,\ell(\boldsymbol{a})=r} \frac{1}{2}(a_r^* - a_{-r}^*)\delta\eta_{\boldsymbol{a}} = \sum_{\boldsymbol{a}\in D,\ell(\boldsymbol{a})=r} a_r^*\delta\eta_{\boldsymbol{a}}, \tag{D.37}$$

$$\tilde{R}_r := \lim_{n\to\infty}\frac{R_r}{n^{\rho_r-1}} \simeq \begin{cases} \sum_{\boldsymbol{a}\in A_0} a_r^*\delta\omega_{\boldsymbol{a}}, & r = 1 \\ \sum_{\boldsymbol{a}\in D,\ell(\boldsymbol{a})=r-1} a_r^*\delta\tau_{\boldsymbol{a}}, & r > 1 \end{cases}. \tag{D.38}$$

For $\lambda_n = \Lambda n^{c(p,q)}$, we have

$$i\lambda_n\mathcal{B} = \frac{i\lambda_n}{n^{q-1}}\sum_{r=1}^p \gamma_r\sum_{k\text{ odd}} 2\binom{q}{k}L_r^k R_r^{q-k},$$

$$\lim_{n\to\infty} i\lambda_n\mathcal{B} = \lim_{n\to\infty}\frac{i\Lambda n^{c(p,q)}}{n^{q-1}}\sum_{r=1}^p \gamma_r\sum_{k\text{ odd}} 2\binom{q}{k}\tilde{L}_r^k\tilde{R}_r^{q-k}n^{k(1-\rho_r)+(q-k)\rho_{r-1}}.$$

One can verify that for any $1 \leq r \leq p$,

$$\lim_{n\to\infty}\frac{n^{c(p,q)}}{n^{q-1}}n^{k(1-\rho_r)+(q-k)\rho_{r-1}} = n^{(k-1)(1-\rho_r-\rho_{r-1})} = \begin{cases} 1, & k = 1 \\ 1/n^\epsilon \text{ for some } \epsilon > 0, & k \geq 3 \end{cases} \tag{D.39}$$

Hence, in the $n \to \infty$ limit, only the $k = 1$ term survives, and

$$\lim_{n\to\infty} i\lambda\mathcal{B} = i\Lambda\sum_{r=1}^p 2q\gamma_r\tilde{L}_r\tilde{R}_r^{q-1}. \tag{D.40}$$

Similarly, consider

$$\mathcal{C} = \sum_{\boldsymbol{a}\in A_0}\frac{\delta n_{\boldsymbol{a}}}{n} + \sum_{\boldsymbol{a}\in D}\frac{\delta t_{\boldsymbol{a}}}{n} = \sum_{\boldsymbol{a}\in A_0}\frac{\delta\omega_{\boldsymbol{a}}}{\sqrt{n}} + \sum_{\boldsymbol{a}\in D}\frac{\delta\tau_{\boldsymbol{a}}n^{\rho_{\boldsymbol{a}}}}{n}. \tag{D.41}$$

In the $n \to \infty$ limit, the only terms that survive are $\delta\tau_{\boldsymbol{a}}$ when $\ell(\boldsymbol{a}) = p$ for which $\rho_{\boldsymbol{a}} = 1$.

Combining the two equations above, we have

$$\lim_{n\to\infty} \Xi_n(\{\delta\tau_{\boldsymbol{a}}, \delta\eta_{\boldsymbol{b}}, \delta\omega_{\boldsymbol{c}}\}_{\boldsymbol{a},\boldsymbol{b}\in D,\boldsymbol{c}\in A_0}) = i\Lambda\sum_{r=1}^p 2q\gamma_r\tilde{L}_r\tilde{R}_r^{q-1} + \zeta\sum_{\boldsymbol{a}:\ell(\boldsymbol{a})=p}\delta\tau_{\boldsymbol{a}} =: \Xi. \tag{D.42}$$

## D.4 MGF at general $p$ in the $n \to \infty$ limit to show Claim 3.7

For succinctness, we denote the following vectors of (rescaled) variables

$$\boldsymbol{t} = (t_{\boldsymbol{a}})_{\boldsymbol{a} \in D}, \qquad \boldsymbol{d} = (d_{\boldsymbol{b}}/n)_{\boldsymbol{b} \in D}, \qquad \boldsymbol{n} = (n_{\boldsymbol{c}}/n)_{\boldsymbol{c} \in A_0},$$

$$\boldsymbol{\delta t} = (\delta t_{\boldsymbol{a}}/n^{\rho_{\boldsymbol{a}}})_{\boldsymbol{a} \in D}, \qquad \boldsymbol{\delta d} = (\delta d_{\boldsymbol{b}}/n^{1-\rho_{\boldsymbol{b}}})_{\boldsymbol{b} \in D}, \qquad \boldsymbol{\delta n} = (\delta n_{\boldsymbol{c}}/\sqrt{n})_{\boldsymbol{c} \in A_0}. \qquad (D.43)$$

We can then write the MGF as

$$\mathbb{E}_{\boldsymbol{Y}}[M_n(\zeta)] = \mathbb{E}_{\boldsymbol{Y}}\left[ \langle \boldsymbol{\gamma}, \boldsymbol{\beta}| \exp(\zeta \frac{1}{n} \sum_{i=1}^n Z_i)|\boldsymbol{\gamma}, \boldsymbol{\beta}\rangle \right] = \sum_{t=0}^n e_n(t), \qquad (D.44)$$

where

$$e_n(t) = \mathbb{T}_n^t \mathbb{S}_n^{\{t_{\boldsymbol{a}}\}} \mathbb{U}_n^t \left[ \exp\left( \Gamma_n(\boldsymbol{t}, \boldsymbol{d}, \boldsymbol{n}) + \Xi_n(\boldsymbol{\delta t}, \boldsymbol{\delta d}, \boldsymbol{\delta n}) \right) \right]. \qquad (D.45)$$

Here, $\Gamma_n$ and $\Xi_n$ are polynomials of their arguments whose coefficients can depend on $n$. Furthermore, $\mathbb{T}_n^t$, $\mathbb{S}_n^{\{t_{\boldsymbol{a}}\}}$, and $\mathbb{U}_n^t$ are summing operators defined in Eqs. (D.20), (D.21), (D.22) earlier.

We now introduce dummy variables $(\boldsymbol{\delta\tau}, \boldsymbol{\eta}, \boldsymbol{\delta\eta}, \boldsymbol{\omega}, \boldsymbol{\delta\omega})$ which will replace $(\boldsymbol{\delta t}, \boldsymbol{d}, \boldsymbol{\delta d}, \boldsymbol{n}, \boldsymbol{\delta n})$ via Dirac delta functions:

$$e_n(t) = \int_{\boldsymbol{\delta\tau}, \boldsymbol{\eta}, \boldsymbol{\delta\eta}, \boldsymbol{\omega}, \boldsymbol{\delta\omega}} \mathbb{T}_n^t \mathbb{S}_n^{\{t_{\boldsymbol{a}}\}} \mathbb{U}_n^t \Big[ \exp\left( \Gamma_n(\boldsymbol{t}, \boldsymbol{\eta}, \boldsymbol{\omega}) + \Xi_n(\boldsymbol{\delta\tau}, \boldsymbol{\delta\eta}, \boldsymbol{\delta\omega}) \right)$$

$$\delta(\boldsymbol{\delta t} - \boldsymbol{\delta\tau})\delta(\boldsymbol{d} - \boldsymbol{\eta})\delta(\boldsymbol{\delta d} - \boldsymbol{\delta\eta})\delta(\boldsymbol{n} - \boldsymbol{\omega})\delta(\boldsymbol{\delta n} - \boldsymbol{\delta\omega}) \Big]$$

$$= \int_{\boldsymbol{\delta\tau}, \boldsymbol{\eta}, \boldsymbol{\delta\eta}, \boldsymbol{\omega}, \boldsymbol{\delta\omega}} \int_{\boldsymbol{\delta\hat{\tau}}, \boldsymbol{\hat{\eta}}, \boldsymbol{\delta\hat{\eta}}, \boldsymbol{\hat{\omega}}, \boldsymbol{\delta\hat{\omega}}} \mathbb{T}_n^t \mathbb{S}_n^{\{t_{\boldsymbol{a}}\}} \mathbb{U}_n^t \Big[ \exp\left( \Gamma_n(\boldsymbol{t}, \boldsymbol{\eta}, \boldsymbol{\omega}) + \Xi_n(\boldsymbol{\delta\tau}, \boldsymbol{\delta\eta}, \boldsymbol{\delta\omega}) \right)$$

$$e^{i\boldsymbol{\delta\hat{\tau}}\cdot(\boldsymbol{\delta t}-\boldsymbol{\delta\tau})+i\boldsymbol{\hat{\eta}}\cdot(\boldsymbol{d}-\boldsymbol{\eta})+i\boldsymbol{\delta\hat{\eta}}\cdot(\boldsymbol{\delta d}-\boldsymbol{\delta\eta})+i\boldsymbol{\hat{\omega}}\cdot(\boldsymbol{n}-\boldsymbol{\omega})+i\boldsymbol{\delta\hat{\omega}}\cdot(\boldsymbol{\delta n}-\boldsymbol{\delta\omega})} \Big].$$

where in the last line we used the Fourier representation of delta functions and introduced dual variables $(\boldsymbol{\delta\hat{\tau}}, \boldsymbol{\hat{\eta}}, \boldsymbol{\delta\hat{\eta}}, \boldsymbol{\hat{\omega}}, \boldsymbol{\delta\hat{\omega}})$.

Note that $\mathbb{S}_n^{\{t_{\boldsymbol{a}}\}}$ is a sum over $(\boldsymbol{d}, \boldsymbol{\delta d}, \boldsymbol{\delta t})$ and $\mathbb{U}_n^t$ is a sum over $(\boldsymbol{n}, \boldsymbol{\delta n})$. We can apply them directly to the relevant exponentials since their dependence is now linear, but involves the dual variables.

First, let us evaluate the $\mathbb{S}_n^{\{t_{\boldsymbol{a}}\}}$ sum, which is defined in Eq. (D.21) as a composition of many little-sums. We start by considering a single little-sum with parameters $(\kappa_{\mathrm{I}}, \kappa_{\mathrm{II}}, \kappa_{\mathrm{III}})$ of the following form:

$$F_{\boldsymbol{a}}(\kappa_{\mathrm{I}}, \kappa_{\mathrm{II}}, \kappa_{\mathrm{III}}) := \oint_{d_{\boldsymbol{a}}, \delta d_{\boldsymbol{a}}, \delta t_{\boldsymbol{a}}}^{t_{\boldsymbol{a}}} e^{\kappa_{\mathrm{I}} d_{\boldsymbol{a}} + \kappa_{\mathrm{II}} \delta d_{\boldsymbol{a}} + \kappa_{\mathrm{III}} \delta t_{\boldsymbol{a}}} \qquad (D.46)$$

$$= \sum_{t_{\boldsymbol{a}+}, t_{\boldsymbol{a}-}} \binom{t_{\boldsymbol{a}}}{t_{\boldsymbol{a}+}, t_{\boldsymbol{a}-}} \sum_{d_{\boldsymbol{a}+}} \binom{t_{\boldsymbol{a}+}}{n_{\boldsymbol{a}+}} Q_{\boldsymbol{a}}^{n_{\boldsymbol{a}+}} Q_{\bar{\boldsymbol{a}}}^{n_{\bar{\boldsymbol{a}}+}} \sum_{d_{\boldsymbol{a}-}} \binom{t_{\boldsymbol{a}-}}{n_{\boldsymbol{a}-}} Q_{\boldsymbol{a}}^{n_{\boldsymbol{a}-}} Q_{\bar{\boldsymbol{a}}}^{n_{\bar{\boldsymbol{a}}-}} e^{\kappa_{\mathrm{I}} d_{\boldsymbol{a}} + \kappa_{\mathrm{II}} \delta d_{\boldsymbol{a}} + \kappa_{\mathrm{III}} \delta t_{\boldsymbol{a}}}.$$

This can be evaluated using $Q_{\bar{\boldsymbol{a}}} = -Q_{\boldsymbol{a}}$ and the basic identity $\sum_{d_{\boldsymbol{a}+}} \binom{t_{\boldsymbol{a}+}}{n_{\boldsymbol{a}+}} (+1)^{n_{\boldsymbol{a}+}} (-1)^{n_{\bar{\boldsymbol{a}}+}} e^{\kappa d_{\boldsymbol{a}+}} = [2 \sinh \kappa]^{t_{\boldsymbol{a}+}}$. Applying this to the two inner sums in $F_{\boldsymbol{a}}$, we get that

$$F_{\boldsymbol{a}}(\kappa_{\mathrm{I}}, \kappa_{\mathrm{II}}, \kappa_{\mathrm{III}}) = Q_{\boldsymbol{a}}^{t_{\boldsymbol{a}}} \sum_{t_{\boldsymbol{a}+}, t_{\boldsymbol{a}-}} \binom{t_{\boldsymbol{a}}}{t_{\boldsymbol{a}+}, t_{\boldsymbol{a}-}} [2 \sinh(\kappa_{\mathrm{I}} + \kappa_{\mathrm{II}})]^{t_{\boldsymbol{a}+}} [2 \sinh(\kappa_{\mathrm{I}} - \kappa_{\mathrm{II}})]^{t_{\boldsymbol{a}-}} e^{\kappa_{\mathrm{III}} \delta t_{\boldsymbol{a}}}$$

$$= (2Q_{\boldsymbol{a}})^{t_{\boldsymbol{a}}} [\sinh(\kappa_{\mathrm{I}} + \kappa_{\mathrm{II}}) e^{\kappa_{\mathrm{III}}} + \sinh(\kappa_{\mathrm{I}} - \kappa_{\mathrm{II}}) e^{-\kappa_{\mathrm{III}}}]^{t_{\boldsymbol{a}}}$$

$$= (4Q_{\boldsymbol{a}})^{t_{\boldsymbol{a}}} (\sinh \kappa_{\mathrm{I}} \cosh \kappa_{\mathrm{II}} \cosh \kappa_{\mathrm{III}} + \cosh \kappa_{\mathrm{I}} \sinh \kappa_{\mathrm{II}} \sinh \kappa_{\mathrm{III}})^{t_{\boldsymbol{a}}}. \qquad (D.47)$$

Returning to $\mathbb{S}_n^t$, we get

$$\mathbb{S}_n^{\{t_{\boldsymbol{a}}\}}[e^{i\boldsymbol{\delta\hat{\tau}}\cdot\boldsymbol{\delta t} + i\boldsymbol{\delta\hat{\eta}}\cdot\boldsymbol{\delta d} + i\boldsymbol{\hat{\eta}}\cdot\boldsymbol{d}}] = \prod_{\boldsymbol{a} \in D} \left\{ \frac{n^{t_{\boldsymbol{a}}}}{t_{\boldsymbol{a}}!} \oint_{d_{\boldsymbol{a}}, \delta d_{\boldsymbol{a}}, \delta t_{\boldsymbol{a}}}^{t_{\boldsymbol{a}}} \right\} e^{i\boldsymbol{\delta\hat{\tau}}\cdot\boldsymbol{\delta t} + i\boldsymbol{\delta\hat{\eta}}\cdot\boldsymbol{\delta d} + i\boldsymbol{\hat{\eta}}\cdot\boldsymbol{d}}$$

$$= \prod_{\boldsymbol{a} \in D} \frac{(4nQ_{\boldsymbol{a}})^{t_{\boldsymbol{a}}}}{t_{\boldsymbol{a}}!} \left( i \sin \frac{\hat{\eta}_{\boldsymbol{a}}}{n} \cos \frac{\delta\hat{\eta}_{\boldsymbol{a}}}{n^{1-\rho_{\boldsymbol{a}}}} \cos \frac{\delta\hat{\tau}_{\boldsymbol{a}}}{n^{\rho_{\boldsymbol{a}}}} - \cos \frac{\hat{\eta}_{\boldsymbol{a}}}{n} \sin \frac{\delta\hat{\eta}_{\boldsymbol{a}}}{n^{1-\rho_{\boldsymbol{a}}}} \sin \frac{\delta\hat{\tau}_{\boldsymbol{a}}}{n^{\rho_{\boldsymbol{a}}}} \right)^{t_{\boldsymbol{a}}}. \qquad (D.48)$$

Next, for $\mathbb{U}_n^t$, we have from the multinomial theorem that

$$\mathbb{U}_n^t[e^{i\hat{\boldsymbol{\omega}}\cdot\boldsymbol{n}+i\delta\hat{\boldsymbol{\omega}}\cdot\boldsymbol{\delta n}}] = \sum_{\{n_{\boldsymbol{a}}\}_{\boldsymbol{a}\in A_0}} \binom{n-t}{\{n_{\boldsymbol{a}}\}} \prod_{\boldsymbol{a}\in A_0} \left\{ Q_{\boldsymbol{a}}^{n_{\boldsymbol{a}}} \sum_{\delta n_{\boldsymbol{a}}} \binom{n_{\boldsymbol{a}}}{n_{\boldsymbol{a}+}} \right\} e^{i\hat{\boldsymbol{\omega}}\cdot\boldsymbol{n}+i\delta\hat{\boldsymbol{\omega}}\cdot\boldsymbol{\delta n}}$$

$$= \left( \sum_{\boldsymbol{a}\in A_0} 2Q_{\boldsymbol{a}} e^{i\hat{\omega}_{\boldsymbol{a}}/n} \cos\frac{\delta\hat{\omega}_{\boldsymbol{a}}}{\sqrt{n}} \right)^{n-t}. \tag{D.49}$$

**Take $n \to \infty$ limit of $e_n(t)$.** We now take the $n \to \infty$ limit while keeping $t$ fixed, assuming $\lambda_n = \Lambda n^{c(p,q)}$. Recall the fact from Appendix D.3 that $0 < \rho_{\boldsymbol{a}} < 1$ when $\ell(\boldsymbol{a}) < p$ and $\rho_{\boldsymbol{a}} = 1$ when $\ell(\boldsymbol{a}) = p$. Then taking the $n \to \infty$ limit of (D.48) yields

$$\lim_{n\to\infty} \mathbb{S}_n^{\{t_{\boldsymbol{a}}\}}[e^{i\delta\hat{\boldsymbol{\tau}}\cdot\boldsymbol{\delta t}+i\delta\hat{\boldsymbol{\eta}}\cdot\boldsymbol{\delta d}+i\hat{\boldsymbol{\eta}}\cdot\boldsymbol{d}}] = \prod_{\boldsymbol{a}\in D} \frac{(4Q_{\boldsymbol{a}})^{t_{\boldsymbol{a}}}}{t_{\boldsymbol{a}}!} [g_{\boldsymbol{a}}(\delta\hat{\tau}_{\boldsymbol{a}}, \hat{\eta}_{\boldsymbol{a}}, \delta\hat{\eta}_{\boldsymbol{a}})]^{t_{\boldsymbol{a}}} \tag{D.50}$$

where

$$g_{\boldsymbol{a}}(\delta\hat{\tau}_{\boldsymbol{a}}, \hat{\eta}_{\boldsymbol{a}}, \delta\hat{\eta}_{\boldsymbol{a}}) = \begin{cases} i\hat{\eta}_{\boldsymbol{a}} - \delta\hat{\eta}_{\boldsymbol{a}}\delta\hat{\tau}_{\boldsymbol{a}}, & \ell(\boldsymbol{a}) < p \\ i\hat{\eta}_{\boldsymbol{a}}\cos\delta\hat{\eta}_{\boldsymbol{a}} - \delta\hat{\tau}_{\boldsymbol{a}}\sin\delta\hat{\eta}_{\boldsymbol{a}}, & \ell(\boldsymbol{a}) = p \end{cases}. \tag{D.51}$$

Similarly, taking the $n \to \infty$ limit of (D.49) gives

$$\lim_{n\to\infty} \mathbb{U}_n^t[e^{i\hat{\boldsymbol{\omega}}\cdot\boldsymbol{n}+i\delta\hat{\boldsymbol{\omega}}\cdot\boldsymbol{\delta n}}] = \exp\left[ \sum_{\boldsymbol{a}\in A_0} 2Q_{\boldsymbol{a}}(i\hat{\omega}_{\boldsymbol{a}} - \frac{1}{2}\delta\hat{\omega}_{\boldsymbol{a}}^2) \right], \tag{D.52}$$

where we used the fact that $\sum_{\boldsymbol{a}\in A_0} 2Q_{\boldsymbol{a}} = 1$. We also note that for any sequence of functions $\{f_n(\boldsymbol{t})\}_n$ that pointwise converges to $f(\boldsymbol{t})$, we have

$$\lim_{n\to\infty} \mathbb{T}_n^t f_n(\boldsymbol{t}) = \lim_{n\to\infty} \frac{t!}{n^t}\binom{n}{t} \sum_{t_{\boldsymbol{a}}\geq 0, \forall \boldsymbol{a}\in D, \sum_{\boldsymbol{a}} t_{\boldsymbol{a}}=t} f_n(\boldsymbol{t}) = \sum_{t_{\boldsymbol{a}}\geq 0, \forall \boldsymbol{a}\in D, \sum_{\boldsymbol{a}} t_{\boldsymbol{a}}=t} f(\boldsymbol{t}) =: \mathbb{T}^t f(\boldsymbol{t}). \tag{D.53}$$

Plugging these back into $e_n(t)$, we get in the limit

$$e(t) := \lim_{n\to\infty} e_n(t)$$

$$= \int_{\delta\boldsymbol{\tau},\boldsymbol{\eta},\delta\boldsymbol{\eta},\boldsymbol{\omega},\delta\boldsymbol{\omega}} \int_{\delta\hat{\boldsymbol{\tau}},\hat{\boldsymbol{\eta}},\delta\hat{\boldsymbol{\eta}},\hat{\boldsymbol{\omega}},\delta\hat{\boldsymbol{\omega}}} \mathbb{T}^t \left[ e^{\Gamma(\boldsymbol{t},\boldsymbol{\eta},\boldsymbol{\omega})+\Xi(\delta\boldsymbol{\tau},\delta\boldsymbol{\eta},\delta\boldsymbol{\omega})} e^{-i\delta\hat{\boldsymbol{\tau}}\cdot\delta\boldsymbol{\tau}-i\hat{\boldsymbol{\eta}}\cdot\boldsymbol{\eta}-i\delta\hat{\boldsymbol{\eta}}\cdot\delta\boldsymbol{\eta}-i\hat{\boldsymbol{\omega}}\cdot\boldsymbol{\omega}-i\delta\hat{\boldsymbol{\omega}}\cdot\delta\boldsymbol{\omega}} \right.$$

$$\left. e^{i\hat{\boldsymbol{\omega}}\cdot(2\boldsymbol{Q})-\frac{1}{2}\delta\hat{\boldsymbol{\omega}}\cdot(2\boldsymbol{Q}\;\delta\hat{\boldsymbol{\omega}})} \prod_{\boldsymbol{a}\in D} \frac{(4Q_{\boldsymbol{a}})^{t_{\boldsymbol{a}}}}{t_{\boldsymbol{a}}!} [g_{\boldsymbol{a}}(\delta\hat{\tau}_{\boldsymbol{a}}, \hat{\eta}_{\boldsymbol{a}}, \delta\hat{\eta}_{\boldsymbol{a}})]^{t_{\boldsymbol{a}}} \right]. \tag{D.54}$$

where we denoted the vector $\boldsymbol{Q} = (Q_{\boldsymbol{a}})_{\boldsymbol{a}\in A_0}$, and $(2\boldsymbol{Q}\;\delta\boldsymbol{\omega})_j = 2Q_j\delta\omega_j$ to mean element-wise product.

**Sum over $e(t)$ to get MGF.** Now we perform the sum over $t$ to get the moment-generating function of the overlap distribution, since (heuristically) $\lim_{n\to\infty} \mathbb{E}_{\boldsymbol{Y}}[M_n(\zeta)] = \sum_{t=0}^{\infty} e(t)$. Note that

$$\sum_{t=0}^{\infty} \mathbb{T}^t f(\boldsymbol{t}) = \sum_{t_{\boldsymbol{a}}\geq 0, \boldsymbol{a}\in D} f(\boldsymbol{t}). \tag{D.55}$$

So in the $n \to \infty$ limit, effectively we are summing over $\{t_{\boldsymbol{a}}\}$ independently. We can also use the fact from [26, Lemma D.2] that $\Gamma(\boldsymbol{t},\boldsymbol{\eta},\boldsymbol{\omega})$ is linear in $\boldsymbol{t}$,

$$\Gamma(\boldsymbol{t},\boldsymbol{\eta},\boldsymbol{\omega}) = \sum_{\boldsymbol{a}\in D} t_{\boldsymbol{a}} P_{\boldsymbol{a}}(\boldsymbol{\eta},\boldsymbol{\omega}). \tag{D.56}$$

Hence, we have

$$\sum_{t=0}^{\infty} e(t) = \int_{\delta\boldsymbol{\tau},\boldsymbol{\eta},\delta\boldsymbol{\eta},\boldsymbol{\omega},\delta\boldsymbol{\omega}} \int_{\delta\hat{\boldsymbol{\tau}},\hat{\boldsymbol{\eta}},\delta\hat{\boldsymbol{\eta}},\hat{\boldsymbol{\omega}},\delta\hat{\boldsymbol{\omega}}} e^{-i\delta\hat{\boldsymbol{\tau}}\cdot\delta\boldsymbol{\tau}-i\hat{\boldsymbol{\eta}}\cdot\boldsymbol{\eta}-i\delta\hat{\boldsymbol{\eta}}\cdot\delta\boldsymbol{\eta}} e^{i\hat{\boldsymbol{\omega}}\cdot(2\boldsymbol{Q}-\boldsymbol{\omega})} e^{-i\delta\hat{\boldsymbol{\omega}}\cdot\delta\boldsymbol{\omega}-\frac{1}{2}\delta\hat{\boldsymbol{\omega}}\cdot(2\boldsymbol{Q}\;\delta\hat{\boldsymbol{\omega}})}$$

$$\exp\left[ \sum_{\boldsymbol{a}\in D} 4Q_{\boldsymbol{a}} g_{\boldsymbol{a}}(\delta\hat{\boldsymbol{\tau}}, \hat{\boldsymbol{\eta}}, \delta\hat{\boldsymbol{\eta}}) e^{P_{\boldsymbol{a}}(\boldsymbol{\eta},\boldsymbol{\omega})} \right] e^{\Xi(\delta\boldsymbol{\tau},\delta\boldsymbol{\eta},\delta\boldsymbol{\omega})}. \tag{D.57}$$

The integrals over $(\hat{\boldsymbol{\omega}}, \boldsymbol{\omega})$ yield Dirac delta functions that set each $\omega_{\boldsymbol{a}} = 2Q_{\boldsymbol{a}}$. The integral over $\delta\hat{\boldsymbol{\omega}}$ yields a Gaussian density function for $\delta\boldsymbol{\omega}$, each with mean 0 and variance $2Q_{\boldsymbol{a}}$. So we can set $\delta\omega_{\boldsymbol{a}} = G_{\boldsymbol{a}} \sim \mathcal{N}(0, 2Q_{\boldsymbol{a}})$, and replace the integrals over $(\delta\hat{\boldsymbol{\omega}}, \delta\boldsymbol{\omega})$ with an expectation over $\boldsymbol{G} = (G_{\boldsymbol{a}})_{\boldsymbol{a} \in A_0}$. Our expression then simplifies to

$$\sum_{t=0}^{\infty} e(t) = \mathbb{E}_{\boldsymbol{G}} \int_{\delta\boldsymbol{\tau}, \boldsymbol{\eta}, \delta\boldsymbol{\eta}} \int_{\delta\hat{\boldsymbol{\tau}}, \hat{\boldsymbol{\eta}}, \delta\hat{\boldsymbol{\eta}}} e^{-i\delta\hat{\boldsymbol{\tau}} \cdot \delta\boldsymbol{\tau} - i\hat{\boldsymbol{\eta}} \cdot \boldsymbol{\eta} - i\delta\hat{\boldsymbol{\eta}} \cdot \delta\boldsymbol{\eta}} \exp\Big[ \sum_{\boldsymbol{a} \in D} 4Q_{\boldsymbol{a}} g_{\boldsymbol{a}}(\delta\hat{\boldsymbol{\tau}}, \hat{\boldsymbol{\eta}}, \delta\hat{\boldsymbol{\eta}}) e^{P_{\boldsymbol{a}}(\boldsymbol{\eta}, 2\boldsymbol{Q})} \Big] e^{\Xi(\delta\boldsymbol{\tau}, \delta\boldsymbol{\eta}, \boldsymbol{G})}$$

$$=: \mathbb{E}_{\boldsymbol{G}} \int_{\delta\boldsymbol{\tau}, \boldsymbol{\eta}, \delta\boldsymbol{\eta}} \int_{\delta\hat{\boldsymbol{\tau}}, \hat{\boldsymbol{\eta}}, \delta\hat{\boldsymbol{\eta}}} e^{S}. \tag{D.58}$$

To do the remaining integrals, it is necessary to use additional structure of the polynomials $P_{\boldsymbol{a}}$, $g_{\boldsymbol{a}}$ and $\Xi$. From [26], we know there is an ordering $(\prec)$ of the elements of $D$ such that the $\boldsymbol{\eta}$ dependence in $P_{\boldsymbol{a}}$ is only on $\{\eta_{\boldsymbol{b}} : \boldsymbol{b} \prec \boldsymbol{a}\}$. Furthermore, from Appendix D.3, we know $\Xi(\delta\boldsymbol{\tau}, \delta\boldsymbol{\eta}, \delta\boldsymbol{\omega})$ has a particular form:

$$\Xi(\delta\boldsymbol{\tau}, \delta\boldsymbol{\eta}, \delta\boldsymbol{\omega}) = i \sum_{\boldsymbol{a} \in D} \delta\eta_{\boldsymbol{a}} R_{\boldsymbol{a}}(\delta\boldsymbol{\tau}, \delta\boldsymbol{\omega}) + \zeta \sum_{\boldsymbol{b}: \ell(\boldsymbol{b})=p} \delta\tau_{\boldsymbol{b}}. \tag{D.59}$$

We also know that the $\delta\boldsymbol{\tau}$ dependence in $R_{\boldsymbol{a}}$ is only on $\{\delta\tau_{\boldsymbol{b}} : \ell(\boldsymbol{b}) < \ell(\boldsymbol{a})\}$. More explicitly, from Eq. (D.38),

$$R_{\boldsymbol{a}}(\delta\boldsymbol{\tau}, \boldsymbol{G}) = 2q\Lambda\gamma_r a_r^* X_r^{q-1}, \quad \text{where } r = \ell(\boldsymbol{a}) \text{ and } X_r = \begin{cases} \sum_{\boldsymbol{b} \in A_0} b_r^* G_{\boldsymbol{b}}, & r = 1 \\ \sum_{\boldsymbol{b} \in D, \ell(\boldsymbol{b})=r-1} b_r^* \delta\tau_{\boldsymbol{b}}, & r > 1 \end{cases}. \tag{D.60}$$

Let us now write out the exponent $S$ in (D.58) using the form of $g_{\boldsymbol{a}}$ in (D.51) and $\Xi$ in (D.59):

$$S = -i\delta\hat{\boldsymbol{\tau}} \cdot \delta\boldsymbol{\tau} - i\hat{\boldsymbol{\eta}} \cdot \boldsymbol{\eta} - i\delta\hat{\boldsymbol{\eta}} \cdot \delta\boldsymbol{\eta}$$

$$+ \sum_{\boldsymbol{a} \in D: \ell(\boldsymbol{a})<p} 4Q_{\boldsymbol{a}}(i\hat{\eta}_{\boldsymbol{a}} - \delta\hat{\eta}_{\boldsymbol{a}}\delta\hat{\tau}_{\boldsymbol{a}}) e^{P_{\boldsymbol{a}}(\boldsymbol{\eta}, 2\boldsymbol{Q})} + \sum_{\boldsymbol{a} \in D: \ell(\boldsymbol{a})=p} 4Q_{\boldsymbol{a}}(i\hat{\eta}_{\boldsymbol{a}} \cos \delta\hat{\eta}_{\boldsymbol{a}} - \delta\hat{\tau}_{\boldsymbol{a}} \sin \delta\hat{\eta}_{\boldsymbol{a}}) e^{P_{\boldsymbol{a}}(\boldsymbol{\eta}, 2\boldsymbol{Q})}$$

$$+ i \sum_{\boldsymbol{a} \in D} \delta\eta_{\boldsymbol{a}} R_{\boldsymbol{a}}(\delta\boldsymbol{\tau}, \boldsymbol{G}) + \zeta \sum_{\boldsymbol{b}: \ell(\boldsymbol{b})=p} \delta\tau_{\boldsymbol{b}}.$$

Regrouping terms, we have

$$S = \sum_{\boldsymbol{a} \in D: \ell(\boldsymbol{a})<p} i\hat{\eta}_{\boldsymbol{a}}(4Q_{\boldsymbol{a}} e^{P_{\boldsymbol{a}}(\boldsymbol{\eta}, 2\boldsymbol{Q})} - \eta_{\boldsymbol{a}}) + \sum_{\boldsymbol{a} \in D: \ell(\boldsymbol{a})=p} i\hat{\eta}_{\boldsymbol{a}}(4Q_{\boldsymbol{a}} \cos \delta\hat{\eta}_{\boldsymbol{a}} e^{P_{\boldsymbol{a}}(\boldsymbol{\eta}, 2\boldsymbol{Q})} - \eta_{\boldsymbol{a}})$$

$$+ \sum_{\boldsymbol{a} \in D: \ell(\boldsymbol{a})<p} i\delta\hat{\tau}_{\boldsymbol{a}}(i4Q_{\boldsymbol{a}}\delta\hat{\eta}_{\boldsymbol{a}} e^{P_{\boldsymbol{a}}(\boldsymbol{\eta}, 2\boldsymbol{Q})} - \delta\tau_{\boldsymbol{a}}) + \sum_{\boldsymbol{a} \in D: \ell(\boldsymbol{a})=p} i\delta\hat{\tau}_{\boldsymbol{a}}(i4Q_{\boldsymbol{a}} \sin \delta\hat{\eta}_{\boldsymbol{a}} e^{P_{\boldsymbol{a}}(\boldsymbol{\eta}, 2\boldsymbol{Q})} - \delta\tau_{\boldsymbol{a}})$$

$$+ \sum_{\boldsymbol{a} \in D} i\delta\eta_{\boldsymbol{a}}[R_{\boldsymbol{a}}(\delta\boldsymbol{\tau}, \boldsymbol{G}) - \delta\hat{\eta}_{\boldsymbol{a}}] + \zeta \sum_{\boldsymbol{b} \in D: \ell(\boldsymbol{b})=p} \delta\tau_{\boldsymbol{b}}. \tag{D.61}$$

Integrating over $(\hat{\boldsymbol{\eta}}, \boldsymbol{\eta})$ yields delta functions that assign $\eta_{\boldsymbol{a}} = 4Q_{\boldsymbol{a}} e^{P_{\boldsymbol{a}}(\boldsymbol{\eta}, 2\boldsymbol{Q})}$ when $\ell(\boldsymbol{a}) < p$, or $\eta_{\boldsymbol{a}} = 4Q_{\boldsymbol{a}} \cos \delta\hat{\eta}_{\boldsymbol{a}} e^{P_{\boldsymbol{a}}(\boldsymbol{\eta}, 2\boldsymbol{Q})}$ when $\ell(\boldsymbol{a}) = p$. Note $4Q_{\boldsymbol{a}} e^{P_{\boldsymbol{a}}} = 2W_{\boldsymbol{a}}$ where $W_{\boldsymbol{a}}$ is defined the same way for $q$-spin models as in [26], so we will use $W_{\boldsymbol{a}} = 2Q_{\boldsymbol{a}} e^{P_{\boldsymbol{a}}}$ in what follows. Then, integrating over $(\delta\boldsymbol{\eta}, \delta\hat{\boldsymbol{\eta}})$ yields delta functions that assign $\delta\hat{\eta}_{\boldsymbol{a}} = R_{\boldsymbol{a}}(\delta\boldsymbol{\tau}, \boldsymbol{G})$. Note here the linear dependence in $\delta\eta_{\boldsymbol{a}}$ in $\Xi(\delta\boldsymbol{\tau}, \delta\boldsymbol{\eta}, \delta\boldsymbol{\omega})$, as in Eq. (D.59), is important for allowing us to evaluate the integrals. Finally, integrating over $(\delta\hat{\boldsymbol{\tau}}, \delta\boldsymbol{\tau})$ yields delta functions that assign $\delta\tau_{\boldsymbol{a}} = i4Q_{\boldsymbol{a}} R_{\boldsymbol{a}} e^{P_{\boldsymbol{a}}} = i2W_{\boldsymbol{a}} R_{\boldsymbol{a}}$ when $\ell(\boldsymbol{a}) < p$, and $\delta\tau_{\boldsymbol{a}} = i4Q_{\boldsymbol{a}} \sin R_{\boldsymbol{a}} e^{P_{\boldsymbol{a}}} = i2W_{\boldsymbol{a}} \sin R_{\boldsymbol{a}}$ when $\ell(\boldsymbol{a}) = p$. Note that these assignments by delta functions are consistent if we perform the integrals according to the ascending order of the set $D$, since $P_{\boldsymbol{a}}$, $R_{\boldsymbol{a}}$ only depend on the variables $\{(\eta_{\boldsymbol{b}}, \delta\tau_{\boldsymbol{b}}) : \boldsymbol{b} \prec \boldsymbol{a}\}$, which would have already been assigned values from earlier integrals.

The MGF of the overlap distribution is then

$$\lim_{n \to \infty} \mathbb{E}_{\boldsymbol{Y}}[M_n(\zeta)] = \sum_{t=0}^{\infty} e(t) = \mathbb{E}_{\boldsymbol{G}} \left[ \exp\left( \zeta \sum_{\boldsymbol{b} \in D: \ell(\boldsymbol{b})=p} i2W_{\boldsymbol{b}} \sin R_{\boldsymbol{b}}(\boldsymbol{G}) \right) \right]. \tag{D.62}$$

In what follows, let us denote $D_r = \{\boldsymbol{b} \in D : \ell(\boldsymbol{b}) = r\}$ for $1 \leq r \leq p$. Also let $\tilde{\gamma}_r = 2q\Lambda\gamma_r$, and $G = \sum_{\boldsymbol{a} \in A_0} a_1^* G_{\boldsymbol{a}}$. Note $G \sim \mathcal{N}(0,1)$ since $G_{\boldsymbol{a}} \sim \mathcal{N}(0, 2Q_{\boldsymbol{a}})$ and $\sum_{\boldsymbol{a} \in A_0} 2Q_{\boldsymbol{a}} = 1$. To get a sense of the MGF formula, observe that

$$\ell(\boldsymbol{a}) = 1 \quad \Longrightarrow \quad R_{\boldsymbol{a}} = \tilde{\gamma}_1 a_1^* G^{q-1},$$

$$\ell(\boldsymbol{a}) = 2 \quad \Longrightarrow \quad R_{\boldsymbol{a}} = \tilde{\gamma}_2 a_2^* \Big( \sum_{\boldsymbol{b} \in D_1} i2W_{\boldsymbol{b}} R_{\boldsymbol{b}} b_2^* \Big)^{q-1} = \tilde{\gamma}_2 a_2^* \Big( \sum_{\boldsymbol{b} \in D_1} i2W_{\boldsymbol{b}} b_1 \Big)^{q-1} [\tilde{\gamma}_1 G^{q-1}]^{q-1}.$$

Note in the last line we used $b_1^* b_2^* = b_1$. Doing this iteratively, we see that when $\ell(\boldsymbol{a}) = r$, we have

$$R_{\boldsymbol{a}} = a_r^* K_r G^{(q-1)^r}, \qquad \text{where} \quad K_r = \tilde{\gamma}_r \Big( \sum_{\boldsymbol{b} \in D_{r-1}} i2W_{\boldsymbol{b}} b_{r-1} \Big)^{q-1} K_{r-1}^{q-1} \tag{D.63}$$

with initial condition $K_1 = \tilde{\gamma}_1$. Note that $K_r \sim \Lambda^{[(q-1)^r - 1]/(q-2)}$ when $q > 2$ and $K_r \sim \Lambda^r$ when $q = 2$. Furthermore, using the fact that $\sin(aX) = a \sin X$ when $a \in \{\pm 1\}$, we have from Eq. (D.62) that

$$\mathcal{R}_{\text{QAOA}} \xrightarrow{d} \Big( \sum_{\boldsymbol{a} \in D_p} i2W_{\boldsymbol{a}} a_p^* \Big) \sin\big[ K_p G^{(q-1)^p} \big], \tag{D.64}$$

which is indeed of the form of the sine-Gaussian law in Claim 3.7.

We then note that the factors

$$\sum_{\boldsymbol{b} \in D_{r-1}} 2W_{\boldsymbol{b}} b_{r-1}, \qquad \sum_{\boldsymbol{b} \in D_p} 2W_{\boldsymbol{b}} b_p^* \tag{D.65}$$

can be evaluated efficiently using the iterative procedure in [25] due to Theorem 3 in [26]. We give this procedure in the section that immediately follows. This concludes the derivation that shows Claim 3.7.

## D.5 A self contained formula for $a_p(\boldsymbol{\gamma}, \boldsymbol{\beta})$ and $b_p(\boldsymbol{\gamma}, \boldsymbol{\beta})$

In this section, we give a self-contained description of the formula for $(a_p, b_p)$, following Eq. (D.64). Let $B$ be the set of $(2p + 1)$-bit strings indexed as $B = \big\{ (z_1, z_2, \ldots, z_p, z_0, z_{-p}, \ldots, z_{-1}) : z_j \in \{\pm 1\} \big\}$. Define

$$f(\boldsymbol{z}) = \frac{1}{2} \langle z_1 | e^{i\beta_1 X} | z_2 \rangle \cdots \langle z_{p-1} | e^{i\beta_{p-1} X} | z_p \rangle \langle z_p | e^{i\beta_p X} | z_0 \rangle$$

$$\times \langle z_0 | e^{-i\beta_p X} | z_{-p} \rangle \langle z_{-p} | e^{-i\beta_{p-1} X} | z_{-(p-1)} \rangle \cdots \langle z_{-2} | e^{-i\beta_1 X} | z_{-1} \rangle \tag{D.66}$$

where $z_i \in \{+1, -1\}$, and $\langle z_1 | e^{i\beta X} | z_2 \rangle = \cos\beta$ if $z_1 = z_2$, or $i\sin(\beta)$ otherwise. Define matrices $\boldsymbol{H}^{[m]} \in \mathbb{C}^{(2p+1) \times (2p+1)}$ for $0 \leq m \leq p$ as follows. For $j, k \in \{1, \ldots, p, 0, -p, \ldots, -1\}$, let $H_{j,k}^{[0]} = \sum_{\boldsymbol{z} \in B} f(\boldsymbol{z}) z_j z_k$, and

$$H_{j,k}^{[m]} = \sum_{\boldsymbol{z} \in B} f(\boldsymbol{z}) z_j z_k \exp\Big( -\frac{q}{2} \sum_{j',k'=-p}^{p} \big( H_{j',k'}^{[m-1]} \big)^{q-1} \gamma_{j'} \gamma_{k'} z_{j'} z_{k'} \Big) \quad \text{for } 1 \leq m \leq p, \tag{D.67}$$

where we use the convention that $\gamma_{-r} = -\gamma_r$ for $1 \leq r \leq p$, and $\gamma_0 = 0$. Note these matrices first appeared in [25] in the context of assessing the performance of the QAOA on locally treelike Max-$q$-XORSAT problems and can be evaluated in $O(p^2 4^p)$ time.

Once we have the matrix $\boldsymbol{H}^{[p]}$, we compute for $1 \leq r \leq p$,

$$a_r = i \sum_{\boldsymbol{z} \in B} f(\boldsymbol{z}) \frac{z_r z_{r+1} - z_{-r} z_{-(r+1)}}{2} \prod_{s=r+1}^{p} \frac{1 + z_s z_{-s}}{2} \exp\Big( -\frac{q}{2} \sum_{j,k=-p}^{p} H_{j,k}^{[p]} \gamma_j \gamma_k z_j z_k \Big). \tag{D.68}$$

Finally, let $b_1 = 2q\gamma_1$, and for $r = 2, 3, \ldots, p$, compute

$$b_r = 2q\gamma_r (a_{r-1} b_{r-1})^{q-1}. \tag{D.69}$$

**Example formula at** $p = 2$. As an example, we now describe the explicit formula at $p = 2$, which applies in the regime where $\lambda_n = \Lambda n^{(q-2+1/q)/2}$ (note here $\varepsilon_{p=2} = 1/q$). We have

$$b_2 = 2^q q^{q-1} e^{-2q(q-1)\gamma_1^2} \gamma_1^{q-1} \gamma_2 \sin^{q-1}(2\beta_1),$$

$$a_2 = -e^{-2q(\gamma_1^2 + \gamma_2^2 + 2\operatorname{Re}[X]\gamma_1\gamma_2)} \sin 2\beta_2 \times$$

$$\left[ \cos^2 \beta_1 + e^{8q\gamma_1\gamma_2 \operatorname{Re}[X]} \sin^2 \beta_1 + e^{2q(\gamma_1^2 + 2\gamma_1\gamma_2 \operatorname{Re}[X])} \sin 2\beta_1 \sin(4q\gamma_1\gamma_2 \operatorname{Im}[X]) \right],$$

where $X = (\cos 2\beta_1 + ie^{-2q\gamma_1^2} \sin 2\beta_1)^{q-1}$. Then the overlap $\mathcal{R} \xrightarrow{d} a_2 \sin(b_2 \Lambda^q G^{(q-1)^2})$.

Although the above formula is complicated, we can understand the scaling with $q$ by considering a simple choice of $\gamma_1 = \gamma_2 = 1/2\sqrt{q}$ and $\beta_1 = \beta_2 = \pi/4$. Then the above simplifies to

$$b_2 = e^{(1-q)/2} q^{q/2},$$

$$a_2 = e^{-1} \cosh[e^{(1-q)/2} \sin(\pi q/2)] - e^{-1/2} \sin[e^{(1-q)/2} \cos(\pi q/2)]. \tag{D.70}$$

# E  Signal boosting with 1-step QAOA

Consider a scenario where we have some prior information about the signal, in the form of a weak estimator that overlaps partially with the true signal. Our goal is to boost the overlap of this weak estimator. We study the SNR threshold of the 1-step of QAOA and compare it to the 1-step of power iteration. For QAOA, we encode the weak estimator into the initial state: rather than initializing with the uniform superposition across all bit-strings $|s\rangle$, we bias a fraction of the qubits toward the signal. For power iteration, instead of starting from a uniform vector, we sample from a Bernoulli distribution biased toward the signal.

More precisely, for QAOA we consider the following initial state:

$$|s_{\text{biased}}\rangle = \bigotimes_{j=1}^{n} \left( \cos\theta_j |u_j\rangle + \sin\theta_j |-u_j\rangle \right), \tag{E.1}$$

where the $\theta_j$ are drawn i.i.d. according to

$$\theta_j = \begin{cases} \pi/4, & \text{with probability } 1 - \frac{k}{n}, \\ \pi/4 - \delta, & \text{with probability } \frac{k}{n}, \end{cases} \tag{E.2}$$

and $\delta > 0$. As in Eq. (2.3), we prepare the 1-step QAOA state as $|\gamma, \beta\rangle_{\text{biased}} = e^{-i\beta B} e^{-i\gamma C} |s_{\text{biased}}\rangle$. Note the spiked tensor model $Y$ is encoded in this state through $C(\sigma) = \langle Y, \sigma^{\otimes q} \rangle / n^{(q-2)/2}$. The following theorem concerns the SNR threshold for weak recovery and the distribution of overlap $\mathcal{R}_{\text{QAOA,biased}} = \hat{u}^\top u/n$ between a sample $\hat{u} \sim |\gamma, \beta\rangle_{\text{biased}}$ and the signal $u$.

**Theorem 2** (Signal boosting with 1-step QAOA). *Consider the biased 1-step QAOA state* $|\gamma, \beta\rangle_{\text{biased}}$ *as defined above. Fix* $\gamma > 0$, $\beta \in [0, 2\pi]$, $\delta \in [0, \pi/4]$, *and let* $k = \Theta(n^c)$ *for* $1/2 < c < 1$. *Suppose*

$$\lim_{n \to \infty} \lambda_n / n^{(1-c)(q-1)} = \Lambda. \tag{E.3}$$

*Then, over the randomness of* $\theta, Y$ *and quantum measurement, the overlap* $\mathcal{R}_{\text{QAOA,biased}}$ *of 1-step QAOA converges in probability to*

$$\mathcal{R}_{\text{QAOA,biased}} \xrightarrow{p} e^{-2q\gamma^2} \sin(2\beta) \sin(2q\Lambda\gamma \sin^{q-1}(2\delta)). \tag{E.4}$$

We give the proof Theorem 2 in Appendix E.1 that follows.

**Remark E.1.** Theorem 2 considers an initial state with a fraction $k/n$ of qubits biased toward the signal vector $u$, representing some side information. It shows that the SNR threshold is $\Theta(n^{(1-c)(q-1)})$, which becomes lower with increasing side information $k/n = n^{c-1}$. In particular, if $k = \Theta(n^{3/4})$, the weak recovery threshold of 1-step QAOA improves to $\Theta(n^{(q-1)/4})$, compared to the $\Theta(n^{(q-1)/2})$ threshold given by Theorem 1 without any initial overlap between the state and planted signal.

**Comparison with classical tensor power iteration.** We compare the boosting produced by the 1-step QAOA to that provided by 1-step power iteration. Recall the 1-step tensor power iteration estimator (2.1) is $\hat{\boldsymbol{u}}_{1,\text{biased}} = \sqrt{n}\boldsymbol{Y}\left[\hat{\boldsymbol{u}}_{0,\text{biased}}^{\otimes(q-1)}\right] / \left\|\boldsymbol{Y}\left[\hat{\boldsymbol{u}}_{0,\text{biased}}^{\otimes(q-1)}\right]\right\|_2$, where in this case, analogously to Eq. (E.1), the initial vector $\hat{\boldsymbol{u}}_{0,\text{biased}}$ has its entry $(\hat{\boldsymbol{u}}_{0,\text{biased}})_j$ sampled as

$$(\hat{\boldsymbol{u}}_{0,\text{biased}})_j \sim \begin{cases} u_j/\sqrt{n}, & \text{with probability } \frac{1}{2}\left[1 + \frac{k}{n}\sin(2\delta)\right], \\ -u_j/\sqrt{n}, & \text{with probability } \frac{1}{2}\left[1 - \frac{k}{n}\sin(2\delta)\right]. \end{cases} \tag{E.5}$$

One can check that $\sqrt{n}\hat{\boldsymbol{u}}_{0,\text{biased}} \sim |s_{\text{biased}}\rangle$ is a sample from the biased initial QAOA state, so that we are making a fair comparison with QAOA. In the following proposition, we show that the required SNR for the 1-step power iteration estimator is also $\Theta(n^{(1-c)(q-1)})$, and we provide the distribution of overlap $\mathcal{R}_{\text{PI,biased}} \equiv \boldsymbol{u}^\top \hat{\boldsymbol{u}}_{1,\text{biased}}/n$ between the power iteration estimator $\hat{\boldsymbol{u}}_{1,\text{biased}}$ and the signal $\boldsymbol{u}$.

**Proposition E.2** (Signal boosting with 1-step tensor power iteration). *Assume that the rescaled signal-to-noise ratio has a limit $\lim_{n\to\infty}\lambda_n/n^{(1-c)(q-1)} = \Lambda$. Then over the randomness of $\boldsymbol{W}$ and initialization $\hat{\boldsymbol{u}}_{0,\text{biased}}$, the overlap $\mathcal{R}_{\text{PI,biased}}$ of the power iteration estimator with the signal converges in probability to*

$$\mathcal{R}_{\text{PI,biased}} \xrightarrow{p} \sin[\arctan(\Lambda \sin^{q-1}(2\delta))]. \tag{E.6}$$

The proof of Proposition E.2 is contained in Appendix H.2. This shows yet again that the QAOA has the same asymptotic computational efficiency as power iteration. Nevertheless, in the $\Lambda \ll 1$ regime, by choosing $\gamma = 1/2\sqrt{q}$ and $\beta = \pi/8$, the QAOA achieves an overlap that is larger than power iteration by a factor $\sqrt{q/e}$.

## E.1 Proof of Theorem 2

Without loss of generality, we assume that $\boldsymbol{u} = \boldsymbol{1}$. Recall that the initial state is given by Eq. (E.1), which we can rewrite as

$$|s_{\text{biased}}\rangle = \sum_{\boldsymbol{z}} \prod_{j=1}^{n} (\cos\theta_j)^{\delta_{z_j=1}} (\sin\theta_j)^{\delta_{z_j=-1}} |\boldsymbol{z}\rangle, \tag{E.7}$$

where $\theta_j = \pi/4$ with probability $1 - k/n$, and $\theta_j = \pi/4 - \delta$ with probability $k/n$.

To prove Theorem 2, it suffices to show that the moment-generating function (MGF) of the QAOA overlap converges to the MGF of a deterministic variable as follows:

$$\lim_{n\to\infty}\mathbb{E}_{\boldsymbol{\theta}}\,\mathbb{E}_{\boldsymbol{Y}}[M_n(\zeta)] = \exp\left[\zeta e^{-2q\gamma^2}\sin(2\beta)\sin(2q\Lambda\gamma\sin(2\delta)^{q-1})\right] =: M(\zeta). \tag{E.8}$$

The argument for the proof is the same as that for Theorem 1(b), except that we must prove analogous versions of Lemma C.1, C.2 and C.3, which become Lemma E.3, E.4 and E.5, respectively.

**Lemma E.3.** *The expected moment-generating function at $p = 1$ for the overlap of the QAOA initialized with $|s_{\text{biased}}\rangle$ is given by*

$$\mathbb{E}_{\boldsymbol{\theta}}\,\mathbb{E}_{\boldsymbol{Y}}[M_n(\zeta)] = \sum_{t=0}^{n}\binom{n}{t}e^{-\gamma^2[n^q-(n-2t)^q]/n^{q-1}}\left[\sinh(\zeta/n)\sin(2\beta)\left(1 - \frac{k}{n} + \frac{k\cos(2\delta)}{n}\right)\right]^t \cdot E_{n,t}, \tag{E.9}$$

*where*

$$E_{n,t} = \frac{1}{2t+1}\sum_{\xi=-t}^{t}\sin^t(2\pi\xi/(2t+1))\hat{Z}_{n,t}(\xi),$$

$$\hat{Z}_{n,t}(\xi) = \sum_{l=-t}^{t}e^{-2\pi i\xi l/(2t+1)}Z_{n,t}(l),$$

$$Z_{n,t}(l) = \frac{1}{2^{n-t}}\sum_{\tau_+ + \tau_- = n-t}\binom{n-t}{\tau_+, \tau_-}(e^{\zeta/n}\cos^2\beta + e^{-\zeta/n}\sin^2\beta)^{\tau_+}(e^{-\zeta/n}\cos^2\beta + e^{\zeta/n}\sin^2\beta)^{\tau_-}$$

$$\times \left(1 + \frac{k\sin(2\delta)}{n}\right)^{\tau_+}\left(1 - \frac{k\sin(2\delta)}{n}\right)^{\tau_-}$$

$$\times e^{i\Lambda_n\gamma[((\tau_+-\tau_-)+l)^q - ((\tau_+-\tau_-)-l)^q]/n^{c(q-1)}}. \tag{E.10}$$

The proof of Lemma E.3 is deferred to Section E.2. Note the only difference from the unbiased case (Lemma C.1) is the presence of the two terms

$$\left(1 - \frac{k}{n} + \frac{k\cos(2\delta)}{n}\right)^t \qquad \text{and} \qquad \left(1 + \frac{k\sin(2\delta)}{n}\right)^{\tau_+}\left(1 - \frac{k\sin(2\delta)}{n}\right)^{\tau_-},$$

and the rescaled power of $n$ in the exponent.

We further define

$$\Lambda = \lim_{n\to\infty} \Lambda_n,$$

$$I_{n,t} = \binom{n}{t} e^{-\gamma^2[n^q - (n-2t)^q]/n^{q-1}} \left[\sinh(\zeta/n)\sin(2\beta)\left(1 - \frac{k}{n} + \frac{k\cos(2\delta)}{n}\right)\right]^t \cdot E_{n,t}, \quad \text{(E.11)}$$

$$I_t = \frac{1}{t!}\left[[\zeta e^{-2q\gamma^2}\sin(2\beta)\sin(2q\Lambda\gamma\sin^{q-1}(2\delta))]^t\right],$$

where the definition of $E_{n,t}$ is given in Eq. (E.10). Then it is easy to see that

$$\mathbb{E}_{\boldsymbol{\theta}}\,\mathbb{E}_{\boldsymbol{Y}}[M_n(\zeta)] = \sum_{t=0}^{n} I_{n,t}, \quad M(\zeta) = \sum_{t=0}^{\infty} I_t.$$

As a consequence, we have

$$\left|\mathbb{E}_{\boldsymbol{Y}}[M_n(\zeta)] - M(\zeta)\right| \leq \sum_{t=0}^{T} |I_{n,t} - I_t| + \left|\sum_{t\geq T+1} I_t\right| + \sum_{t=T+1}^{n} |I_{n,t}|. \quad \text{(E.12)}$$

The following lemma gives the limit of $E_{n,t}$ for fixed $t$ as $n \to \infty$, which indicates that $I_t$ is the limit of $I_{n,t}$.

**Lemma E.4.** *For any fixed integer t, we have*

$$\lim_{n\to\infty} E_{n,t} = \sin^t(2q\Lambda\gamma\sin^{q-1}(2\delta)) \equiv E_t. \quad \text{(E.13)}$$

*As a consequence, we have*

$$\lim_{n\to\infty} I_{n,t} = I_t.$$

**Lemma E.5.** *For any $t \leq n$ and $\zeta \leq n$, we have*

$$|I_{n,t}| \leq \frac{1}{t!}(18|\zeta|)^t(2t+1)e^{|\zeta|} \equiv s_t, \quad \text{(E.14)}$$

*where*

$$\sum_{t=0}^{\infty} s_t < \infty. \quad \text{(E.15)}$$

The proof of Lemma E.4 and E.5 is deferred to Section E.3 and E.4, respectively. Now we assume that these two lemmas hold. By the fact that $\sum_{t=0}^{\infty} I_t$ is finite and by Lemma E.5, for any $\varepsilon > 0$, there exists $T = T_\varepsilon$ such that

$$\left|\sum_{t\geq T_\varepsilon+1} I_t\right| \leq \varepsilon/3, \quad \sum_{t\geq T_\varepsilon+1} s_t \leq \varepsilon/3.$$

Furthermore, by Lemma E.4, there exists $N = N_\varepsilon$ such that as long as $n \geq N_\varepsilon$, we have

$$\sum_{t=0}^{T_\varepsilon} |I_{n,t} - I_t| \leq \varepsilon/3.$$

As a consequence, by Eq. (E.12), for any $n \geq n_\varepsilon$ and $\zeta \leq n$, we have

$$\left|\mathbb{E}_{\boldsymbol{Y}}[M_n(\zeta)] - M(\zeta)\right| \leq \sum_{t=0}^{T_\varepsilon} |I_{n,t} - I_t| + \left|\sum_{t\geq T_\varepsilon+1} I_t\right| + \sum_{t=T_\varepsilon+1}^{\infty} s_t \leq \varepsilon. \quad \text{(E.16)}$$

This proves Eq. (E.8) as desired, and hence finishes the proof of Theorem 2.

## E.2 Proof of Lemma E.3

With an added expectation over $\boldsymbol{\theta}$, Eq. (B.4) still holds with a modified $Q_{\boldsymbol{a}}$:

$$\mathbb{E}_{\boldsymbol{\theta}} \, \mathbb{E}_{\boldsymbol{Y}} [M_n(\zeta)] = \sum_{\{n_{\boldsymbol{a}}\}} \binom{n}{\{n_{\boldsymbol{a}}\}} \prod_{\boldsymbol{a} \in B} Q_{\boldsymbol{a}}^{n_{\boldsymbol{a}}} \exp\Big[ -\frac{1}{2n^{q-1}} \sum_{\underline{\boldsymbol{a}} \in B^q} \Phi_{\underline{\boldsymbol{a}}}^2 \prod_{s=1}^{q} n_{\boldsymbol{a}_s}$$

$$+ \frac{i\lambda_n}{n^{q-1}} \sum_{\underline{\boldsymbol{a}} \in B^q} \Phi_{\underline{\boldsymbol{a}}} \prod_{s=1}^{q} (\boldsymbol{a}_s)_m n_{\boldsymbol{a}_s} + \frac{\zeta}{n} \sum_{\boldsymbol{v} \in B} v_m n_{\boldsymbol{v}} \Big], \tag{E.17}$$

and

$$Q_{(a_1, a_m, a_2)} = f_{\beta, k, \delta}(a_1, a_m, a_2) \tag{E.18}$$

with $f$ defined below:

$$f_{\beta, k, \delta}(z_j^1, z_j^m, z_j^2) = \begin{cases} \frac{1}{2}\Big(1 + \frac{k \sin(2\delta)}{n}\Big) \cos^2 \beta, & \text{if } (z_j^1, z_j^m, z_j^2) = (1, 1, 1), \\ -\frac{1}{2}\Big(1 - \frac{k}{n} + \frac{k \cos(2\delta)}{n}\Big) i \sin \beta \cos \beta, & \text{if } (z_j^1, z_j^m, z_j^2) = (1, 1, -1), \\ \frac{1}{2}\Big(1 - \frac{k \sin(2\delta)}{n}\Big) \cos^2 \beta, & \text{if } (z_j^1, z_j^m, z_j^2) = (1, -1, 1), \\ -\frac{1}{2}\Big(1 - \frac{k}{n} + \frac{k \cos(2\delta)}{n}\Big) i \sin \beta \cos \beta, & \text{if } (z_j^1, z_j^m, z_j^2) = (1, -1, -1), \\ \frac{1}{2}\Big(1 - \frac{k}{n} + \frac{k \cos(2\delta)}{n}\Big) i \sin \beta \cos \beta, & \text{if } (z_j^1, z_j^m, z_j^2) = (-1, 1, 1), \\ \frac{1}{2}\Big(1 - \frac{k \sin(2\delta)}{n}\Big) \sin^2 \beta, & \text{if } (z_j^1, z_j^m, z_j^2) = (-1, 1, -1), \\ \frac{1}{2}\Big(1 - \frac{k}{n} + \frac{k \cos(2\delta)}{n}\Big) i \sin \beta \cos \beta, & \text{if } (z_j^1, z_j^m, z_j^2) = (-1, -1, 1), \\ \frac{1}{2}\Big(1 + \frac{k \sin(2\delta)}{n}\Big) \sin^2 \beta, & \text{if } (z_j^1, z_j^m, z_j^2) = (-1, -1, -1). \end{cases}$$
$$\tag{E.19}$$

This proof follows very closely that of Theorem 1(b) in Appendix C. From the change of variables in Eq. (C.10) to the breaking up in Eq. (C.15), the same expression still hold, except that we redefine $\Lambda_n = \lambda_n / n^{(1-c)(q-1)}$, which amounts to the power of $n$ in the exponential changing: when compared to Eq. (C.15):

$$\mathbb{E}_{\boldsymbol{\theta}} \, \mathbb{E}_{\boldsymbol{Y}} [M_n(\zeta)] = \sum_{t=0}^{n} \binom{n}{t} e^{-\gamma^2 [n^q - (n-2t)^q]/n^{q-1}} \sum_{t_+ + t_- = t} \binom{t}{t_+, t_-} \sum_{\tau_+ + \tau_- = n-t} \binom{n-t}{\tau_+, \tau_-}$$

$$\sum_{\Delta_+} \binom{\tau_+}{n_{+++}} Q_{+++}^{n_{+++}} Q_{---}^{n_{---}} \sum_{\Delta_-} \binom{\tau_-}{n_{+-+}} Q_{+-+}^{n_{+-+}} Q_{-+-}^{n_{-+-}} e^{(\zeta/n)(t_+ - t_- + \Delta_+ - \Delta_-)}$$

$$\sum_{d_+} \binom{t_+}{n_{++-}} Q_{++-}^{n_{++-}} Q_{-++}^{n_{-++}} \sum_{d_-} \binom{t_-}{n_{+--}} Q_{+--}^{n_{+--}} Q_{--+}^{n_{--+}}$$

$$e^{i\Lambda_n \gamma [(d_+ - d_- + \tau_+ - \tau_-)^q - ((\tau_+ - \tau_-) - (d_+ - d_-))^q]/n^{c(q-1)}}. \tag{E.20}$$

However, it is not true anymore that $Q_{+++} = Q_{+-+}$ and $Q_{---} = Q_{-+-}$ in general. We use the identity in Eq. (C.16) to write

$$\mathbb{E}_{\boldsymbol{\theta}} \, \mathbb{E}_{\boldsymbol{Y}} [M_n(\zeta)] = \sum_{t=0}^{n} \binom{n}{t} e^{-\gamma^2 [n^q - (n-2t)^q]/n^{q-1}} \sum_{t_+ + t_- = t} \binom{t}{t_+, t_-} \sum_{\tau_+ + \tau_- = n-t} \binom{n-t}{\tau_+, \tau_-}$$

$$\frac{1}{2^{n-t}} (2Q_{+++} e^{\zeta/n} + 2Q_{---} e^{-\zeta/n})^{\tau_+} (2Q_{+-+} e^{-\zeta/n} + 2Q_{-+-} e^{\zeta/n})^{\tau_-} e^{(\zeta/n)(t_+ - t_-)}$$

$$\sum_{d_+} \binom{t_+}{n_{++-}} Q_{++-}^{n_{++-}} Q_{-++}^{n_{-++}} \sum_{d_-} \binom{t_-}{n_{+--}} Q_{+--}^{n_{+--}} Q_{--+}^{n_{--+}}$$

$$e^{i\Lambda_n \gamma [(d_+ - d_- + \tau_+ - \tau_-)^q - ((\tau_+ - \tau_-) - (d_+ - d_-))^q]/n^{c(q-1)}}.$$

Then we redefine

$$Z_{n,t}(l) = \frac{1}{2^{n-t}} \sum_{\tau_+ + \tau_- = n-t} \binom{n-t}{\tau_+, \tau_-} (e^{\zeta/n} \cos^2 \beta + e^{-\zeta/n} \sin^2 \beta)^{\tau_+} (e^{-\zeta/n} \cos^2 \beta + e^{\zeta/n} \sin^2 \beta)^{\tau_-}$$

$$\left(1 + \frac{k \sin(2\delta)}{n}\right)^{\tau_+} \left(1 - \frac{k \sin(2\delta)}{n}\right)^{\tau_-} e^{i\Lambda_n \gamma [((\tau_+ - \tau_-) + l)^q - ((\tau_+ - \tau_-) - l)^q]/n^{c(q-1)}}$$

$$(\text{E.21})$$

and, analogously to Eq. (C.19) write

$$\mathbb{E}_{\boldsymbol{\theta}} \, \mathbb{E}_{\boldsymbol{Y}}[M_n(\zeta)] = \sum_{t=0}^{n} \binom{n}{t} e^{-\gamma^2[n^q - (n-2t)^q]/n^{q-1}} \sum_{t_+ + t_- = t} \binom{t}{t_+, t_-} \sum_{d_+} \binom{t_+}{n_{++-}} Q_{++-}^{n_{++-}} Q_{-++}^{n_{-++}}$$

$$\times \sum_{d_-} \binom{t_-}{n_{+--}} Q_{+--}^{n_{+--}} Q_{--+}^{n_{--+}} e^{(\zeta/n)(t_+ - t_-)} Z_{n,t}(d_+ - d_-).$$

$$(\text{E.22})$$

Using the discrete Fourier transform, we have

$$\mathbb{E}_{\boldsymbol{\theta}} \, \mathbb{E}_{\boldsymbol{Y}}[M_n(\zeta)] = \sum_{t=0}^{n} \binom{n}{t} e^{-\gamma^2[n^q - (n-2t)^q]/n^{q-1}} \sum_{t_+ + t_- = t} \binom{t}{t_+, t_-} \sum_{d_+} \binom{t_+}{n_{++-}} Q_{++-}^{n_{++-}} Q_{-++}^{n_{-++}}$$

$$\times \sum_{d_-} \binom{t_-}{n_{+--}} Q_{+--}^{n_{+--}} Q_{--+}^{n_{--+}} e^{(\zeta/n)(t_+ - t_-)} \frac{1}{2t+1} \sum_{\xi=-t}^{t} e^{2\pi i \xi (d_+ - d_-)/(2t+1)} \hat{Z}_{n,t}(\xi)$$

$$= \sum_{t=0}^{n} \binom{n}{t} e^{-\gamma^2[n^q - (n-2t)^q]/n^{q-1}} \sum_{t_+ + t_- = t} \binom{t}{t_+, t_-} e^{(\zeta/n)(t_+ - t_-)}$$

$$\times (-1)^{t_-} \cdot \frac{1}{2t+1} \sum_{\xi=-t}^{t} \left(2iQ_{++-} \sin(2\pi\xi/(2t+1))\right)^t \hat{Z}_{n,t}(\xi) \qquad (\text{E.23})$$

since the same relations between $Q_{++-}, Q_{-++}, Q_{+--}, Q_{--+}$ hold. Finally,

$$\mathbb{E}_{\boldsymbol{\theta}} \, \mathbb{E}_{\boldsymbol{Y}}[M_n(\zeta)] = \sum_{t=0}^{n} \binom{n}{t} e^{-\gamma^2[n^q - (n-2t)^q]/n^{q-1}} (\sinh(\zeta/n) \sin(2\beta))^t \left(1 - \frac{k}{n} + \frac{k \cos(2\delta)}{n}\right)^t$$

$$\times \frac{1}{2t+1} \sum_{\xi=-t}^{t} \left(\sin(2\pi\xi/(2t+1))\right)^t \hat{Z}_{n,t}(\xi), \qquad (\text{E.24})$$

which is analogous to Eq. (C.24). This completes the proof of Lemma E.3.

## E.3 Proof of Lemma E.4

We first look at the limit of $Z_{n,t}(l)$ for fixed integer $-t \le l \le t$. Letting $T_n = (e^{\zeta/n} \cos^2 \beta + e^{-\zeta/n} \sin^2 \beta)$, $U_n = (e^{-\zeta/n} \cos^2 \beta + e^{\zeta/n} \sin^2 \beta)$ and $\epsilon = k \sin(2\delta)/n$, we can write

$$Z_{n,t}(l) = \frac{1}{2^{n-t}} \sum_{\tau_+ + \tau_- = n-t} \binom{n-t}{\tau_+, \tau_-} T_n^{\tau_+} U_n^{\tau_-} (1+\epsilon)^{\tau_+} (1-\epsilon)^{\tau_-} e^{i\Lambda_n \gamma [((\tau_+ - \tau_-) + l)^q - ((\tau_+ - \tau_-) - l)^q]/n^{c(q-1)}}.$$

$$(\text{E.25})$$

We let $G_n = (\tau_+ - \tau_- + \epsilon n - t(\epsilon - 1))/\sqrt{n}$ so that

$$Z_{n,t}(l) = \mathbb{E}_{G_n} \left[ T_n^{(\sqrt{n}G_n + (\epsilon+1)n - t(\epsilon+1))/2} U_n^{(-\sqrt{n}G_n - (\epsilon-1)n + t(\epsilon+1))/2} \right.$$

$$\left. \times e^{\frac{i\Lambda_n \gamma}{n^{c(q-1)}} [(\sqrt{n}G_n + \epsilon n - t(\epsilon+1) + l)^q - (\sqrt{n}G_n + \epsilon n - t(\epsilon+1) - l)^q]} \right], \qquad (\text{E.26})$$

where $\tau_+ \sim \text{Binom}(n-t, (1+\epsilon)/2)$ so that $G_n \to G \sim \mathcal{N}(0,1)$ by the central limit theorem since $\mathbb{E}_{\tau_+}[\tau_+ - \tau_-] = \epsilon n - t(\epsilon+1)$ and $\text{Var}_{\tau_+}[\tau_+ - \tau_-] = (n-t)(1-\epsilon^2)$.

Recall that $\epsilon = \sin(2\delta)n^{c-1}$ where $1/2 < c < 1$. It follows that $\lim_{n\to\infty} T_n^{((\epsilon+1)n - t(\epsilon+1))/2} = \lim_{n\to\infty} U_n^{(-(\epsilon-1)n + t(\epsilon+1))/2} = 1$ as well as $\lim_{n\to\infty} T_n^{\sqrt{n}/2} = \lim_{n\to\infty} U_n^{\sqrt{n}/2} = 1$. Hence, for any fixed $-t \le l \le t$, it follows that

$$
\frac{1}{n^{c(q-1)}}\left[(\sqrt{n}G_n + \epsilon n - t(\epsilon+1) + l)^q - (\sqrt{n}G_n + \epsilon n - t(\epsilon+1) - l)^q\right]
$$

$$
= \frac{1}{n^{c(q-1)}}\left[(\sqrt{n}G_n + n^c\sin(2\delta) - t(n^{c-1}\sin(2\delta)+1) + l)^q\right.
$$

$$
\left. - (\sqrt{n}G_n + n^c\sin(2\delta) - t(n^{c-1}\sin(2\delta)+1) - l)^q\right]
$$

$$
\to 2ql\sin^{q-1}(2\delta). \tag{E.27}
$$

With this, we can conclude

$$
\lim_{n\to\infty} Z_{n,t}(l) = e^{iq\Lambda\gamma 2l\sin^{q-1}(2\delta)}. \tag{E.28}
$$

Hence

$$
\lim_{n\to\infty} E_{n,t} = \frac{1}{2t+1}\sum_{\xi=-t}^{t}\sin(2\pi\xi/(2t+1))^t\left(\sum_{l=-t}^{t}e^{-2\pi i\xi l/(2t+1)}e^{i\Lambda\gamma 2l\sin^{q-1}(2\delta)}\right)
$$

$$
= \sin^t(2q\Lambda\gamma\sin^{q-1}(2\delta)), \tag{E.29}
$$

where we used Lemma C.4 with $X = 1$ with probability 1. This completes the proof of Lemma E.4.

### E.4  Proof of Lemma E.5

We first bound $Z_{n,t}(k)$:

$$
|Z_{n,t}(l)| \le \frac{1}{2^{n-t}}\sum_{\tau_+ + \tau_- = n-t}\binom{n-t}{\tau_+,\tau_-}\left|e^{\zeta/n}\cos^2\beta + e^{-\zeta/n}\sin^2\beta\right|^{\tau_+}\left|e^{-\zeta/n}\cos^2\beta + e^{\zeta/n}\sin^2\beta\right|^{\tau_-}
$$

$$
\times\left(1 - \frac{k\sin(2\delta)}{n}\right)^{\tau_+}\left(1 + \frac{k\sin(2\delta)}{n}\right)^{\tau_-}\left|e^{i\Lambda_n\gamma[((\tau_+-\tau_-)+l)^q - ((\tau_+-\tau_-)-l)^q]/n^{c(q-1)}}\right|
$$

$$
\le \frac{1}{2^{n-t}}\sum_{\tau_+ + \tau_- = n-t}\binom{n-t}{\tau_+,\tau_-}\left(1 - \frac{k\sin(2\delta)}{n}\right)^{\tau_+}\left(1 + \frac{k\sin(2\delta)}{n}\right)^{\tau_-}e^{\tau_+|\zeta|/n}e^{\tau_-|\zeta|/n}\cdot 1
$$

$$
= e^{(n-t)|\zeta|/n}\cdot 1
$$

$$
\le e^{|\zeta|}. \tag{E.30}
$$

The rest of the proof is exactly as the proof of Lemma C.3, except that $I_{n,t}$ involves the following extra factor which we can bound:

$$
\left|\left(1 - \frac{k}{n} + \frac{k\cos(2\delta)}{n}\right)\right| \le 3. \tag{E.31}
$$

So the end bound on $|I_{n,t}|$ ends up with a different constant factor:

$$
|I_{n,t}| \le \frac{1}{t!}(18|\zeta|)^t(2t+1)e^{|\zeta|}. \tag{E.32}
$$

This finishes the proof of Lemma E.5.

## F  Finite $n$ calculation for 1-step QAOA on the spiked matrix ($q = 2$)

In this appendix, we calculate the average squared overlap outputted by the QAOA at any finite problem dimension $n$ and obtain the formula we reported in Eq. (4.1). As done in Appendix B, we first take $\boldsymbol{u} = \mathbf{1}$ to be the all-one vector without loss of generality. The cost function is

$$
C(\boldsymbol{z}) = \sum_{j,k=1}^{n} Y_{j,k}z_j z_k, \qquad \text{where} \qquad Y_{j,k} = \frac{\lambda_n}{n} + \frac{1}{\sqrt{n}}W_{j,k}. \tag{F.1}
$$

Here $W_{j,k} \sim \mathcal{N}(0,1)$.

The QAOA state at level $p = 1$ with this cost function is

$$|\gamma, \beta\rangle = e^{-i\beta B} e^{-i\gamma C} |s\rangle. \tag{F.2}$$

We are interested in the overlap of the QAOA output with the hidden signal $u = 1$. Following the same method as in Appendix B, we can write the disorder-averaged overlap as

$$\mathbb{E}_Y[\langle \mathcal{R}^2_{\text{QAOA}} \rangle_{\gamma,\beta}] = \sum_{\{n_a\}} \binom{n}{\{n_a\}} \prod_{a \in B} Q_a^{n_a} e^{-\frac{1}{2n} \sum_{a,b \in B} \Phi^2_{ab} n_a n_b + \frac{i\lambda n}{n} \sum_{a,b \in B} \Phi_{ab} a_\text{m} b_\text{m} n_a n_b} \left( \frac{1}{n} \sum_{v \in B} v_\text{m} n_v \right)^2, \tag{F.3}$$

where

$$B = \{(a_1, a_\text{m}, a_2) : a_j \in \{\pm 1\}\}, \tag{F.4}$$

$$Q_a = \frac{1}{2} \langle a_1 | e^{i\beta X} |1\rangle \langle 1| e^{-i\beta X} |a_2\rangle, \tag{F.5}$$

and $\quad \Phi_{ab} = \gamma(a_1 b_1 - a_2 b_2). \tag{F.6}$

We can calculate $\mathbb{E}_Y[\langle \mathcal{R}^2_{\text{QAOA}} \rangle_{\gamma,\beta}]$ explicitly with a careful organization of the sum. To this end, similar to what we did in Section C.2.1, we perform a change of variables given by

$$\begin{aligned}
t_+ &= n_{++-} + n_{-++}, & t_- &= n_{+--} + n_{--+}, \\
d_+ &= n_{++-} - n_{-++}, & d_- &= n_{+--} - n_{--+}, \\
\tau_+ &= n_{+++} + n_{---}, & \tau_- &= n_{+-+} + n_{-+-}, \\
\Delta_+ &= n_{+++} - n_{---}, & \Delta_- &= n_{+-+} - n_{-+-}.
\end{aligned} \tag{F.7}$$

Observe that these 8 variables completely determine $\{n_a : a \in B\}$. Furthermore, let

$$t = t_+ + t_-, \qquad \text{and thus} \qquad n - t = \tau_+ + \tau_-. \tag{F.8}$$

Using the identity $a_1 b_1 - a_2 b_2 = [(a_1 + a_2)(b_1 - b_2) + (a_1 - a_2)(b_1 + b_2)]/2$, we can show that

$$\sum_{a,b \in B} \Phi^2_{ab} n_a n_b = 8\gamma^2 t(n - t), \tag{F.9}$$

$$\sum_{a,b \in B} \Phi_{ab} a_\text{m} b_\text{m} n_a n_b = 4\gamma(d_+ - d_-)(\tau_+ - \tau_-), \tag{F.10}$$

$$\sum_{v \in B} v_\text{m} n_v = t_+ - t_- + \Delta_+ - \Delta_-. \tag{F.11}$$

Plugging these into (F.3) and breaking up the sum yield

$$\begin{aligned}
\mathbb{E}_Y[\langle \mathcal{R}^2_{\text{QAOA}} \rangle_{\gamma,\beta}] = \frac{1}{n^2} \sum_{t=0}^n \binom{n}{t} e^{-4\gamma^2 t(n-t)/n} \sum_{t_+ + t_- = t} \binom{t}{t_+, t_-} \sum_{\tau_+ + \tau_- = n-t} \binom{n-t}{\tau_+, \tau_-} \\
\sum_{\Delta_+} \binom{\tau_+}{n_{+++}} Q^{n_{+++}}_{+++} Q^{n_{---}}_{---} \sum_{\Delta_-} \binom{\tau_-}{n_{+-+}} Q^{n_{+-+}}_{+-+} Q^{n_{-+-}}_{-+-} (t_+ - t_- + \Delta_+ - \Delta_-)^2 \\
\sum_{d_+} \binom{t_+}{n_{++-}} Q^{n_{++-}}_{++-} Q^{n_{-++}}_{-++} \sum_{d_-} \binom{t_-}{n_{+--}} Q^{n_{+--}}_{+--} Q^{n_{--+}}_{--+} e^{i\Lambda(d_+ - d_-)(\tau_+ - \tau_-)},
\end{aligned} \tag{F.12}$$

where we've denoted $\Lambda = 4\lambda\gamma/n$ as shorthand. Note we need to perform these sums in a carefully chosen order in order to get a closed-form answer at the end.

We start with the last line, where we sum over $d_\pm$. We can use the fact that $Q_{+x-} = -Q_{-y+} = -\frac{i}{2} \sin\beta \cos\beta$ for any $x, y \in \{\pm\}$. Then, for example we have

$$\sum_{d_+} \binom{t_+}{n_{++-}} Q^{n_{++-}}_{++-} Q^{n_{-++}}_{-++} e^{i\Lambda d_+(\tau_+ - \tau_-)} = \left( 2iQ_{++-} \sin[\Lambda(\tau_+ - \tau_-)] \right)^{t_+}. \tag{F.13}$$

After doing the same thing for the sum over $d_-$, we get

$$\mathbb{E}_{\boldsymbol{Y}}[\langle \mathcal{R}^2_{\mathrm{QAOA}} \rangle_{\gamma,\beta}] = \frac{1}{n^2} \sum_{t=0}^{n} \binom{n}{t} e^{-4\gamma^2 t(n-t)/n} \sum_{t_+ + t_- = t} \binom{t}{t_+, t_-} \sum_{\tau_+ + \tau_- = n-t} \binom{n-t}{\tau_+, \tau_-}$$

$$\sum_{\Delta_+} \binom{\tau_+}{n_{+++}} Q_{+++}^{n_{+++}} Q_{---}^{n_{---}} \sum_{\Delta_-} \binom{\tau_-}{n_{+-+}} Q_{+-+}^{n_{+-+}} Q_{-+-}^{n_{-+-}} (t_+ - t_- + \Delta_+ - \Delta_-)^2$$

$$\left(2iQ_{++-} \sin[\Lambda(\tau_+ - \tau_-)]\right)^t (+1)^{t_+} (-1)^{t_-}.$$

Next, consider the sums over $\{(t_+, t_-) : t_+ + t_- = t\}$. We can use the following identity

$$\sum_{r+s=t} \binom{t}{r, s} (+1)^r (-1)^s (r-s)^k = \begin{cases} \delta_{t=0}, & \text{if } k=0, \\ 2\delta_{t=1}, & \text{if } k=1, \\ 8\delta_{t=2}, & \text{if } k=2. \end{cases} \tag{F.14}$$

Collecting the relevant terms and applying this identity yield

$$\sum_{t_+ + t_- = t} \binom{t}{t_+, t_-} (t_+ - t_- + \Delta_+ - \Delta_-)^2 (+1)^{t_+} (-1)^{t_-}$$

$$= \left[ 8\delta_{t=2} + 4(\Delta_+ - \Delta_-)\delta_{t=1} + (\Delta_+ - \Delta_-)^2 \delta_{t=0} \right]. \tag{F.15}$$

So we have

$$\mathbb{E}_{\boldsymbol{Y}}[\langle \mathcal{R}^2_{\mathrm{QAOA}} \rangle_{\gamma,\beta}] = \frac{1}{n^2} \sum_{t=0}^{n} \binom{n}{t} e^{-4\gamma^2 t(n-t)/n} \sum_{\tau_+ + \tau_- = n-t} \binom{n-t}{\tau_+, \tau_-} \left(2iQ_{++-} \sin[\Lambda(\tau_+ - \tau_-)]\right)^t$$

$$\sum_{\Delta_+} \binom{\tau_+}{n_{+++}} Q_{+++}^{n_{+++}} Q_{---}^{n_{---}} \sum_{\Delta_-} \binom{\tau_-}{n_{+-+}} Q_{+-+}^{n_{+-+}} Q_{-+-}^{n_{-+-}}$$

$$\left[ 8\delta_{t=2} + 4(\Delta_+ - \Delta_-)\delta_{t=1} + (\Delta_+ - \Delta_-)^2 \delta_{t=0} \right]. \tag{F.16}$$

Note the Kronecker deltas will collapse the sum over $t$, so it remains to evaluate the sums over $\Delta_\pm$ and $\tau_\pm$. To perform the sum over $\Delta_\pm$, note that $Q_{+x+} = \frac{1}{2}\cos^2\beta$ and $Q_{-y-} = \frac{1}{2}\sin^2\beta$, for any $x, y \in \{\pm\}$. Thus, we can use the following identity

$$2^{\tau_+} \sum_{\Delta_+} \binom{\tau_+}{n_{+++}} Q_{+++}^{n_{+++}} Q_{---}^{n_{---}} (\Delta_+)^k = \begin{cases} 1, & \text{if } k=0, \\ \tau_+ \cos 2\beta, & \text{if } k=1, \\ \tau_+[1 + (\tau_+ - 1)\cos^2(2\beta)], & \text{if } k=2, \end{cases} \tag{F.17}$$

to write

$$2^{n-t} \sum_{\Delta_+} \binom{\tau_+}{n_{+++}} Q_{+++}^{n_{+++}} Q_{---}^{n_{---}} \sum_{\Delta_-} \binom{\tau_-}{n_{+-+}} Q_{+-+}^{n_{+-+}} Q_{-+-}^{n_{-+-}} (\Delta_+ - \Delta_-)^k$$

$$= \begin{cases} 1, & \text{if } k=0, \\ (\tau_+ - \tau_-)\cos(2\beta), & \text{if } k=1, \\ \tau_+ + \tau_- + \left[\tau_-^2 + \tau_+(\tau_+ - 1) - \tau_-(1 + 2\tau_+)\right]\cos^2(2\beta), & \text{if } k=2. \end{cases} \tag{F.18}$$

Finally, we just need to evaluate the sum over $\tau_\pm$ subject to the three possible values of $t$. Returning to (F.16), we can break $\mathbb{E}_{\boldsymbol{Y}}[\langle \mathcal{R}^2_{\mathrm{QAOA}} \rangle_{\gamma,\beta}] = S_2 + S_1 + S_0$ into three parts, corresponding to $t = 2, 1, 0$,

where

$$S_2 = \frac{8}{n^2}\binom{n}{2}e^{-8\gamma^2(n-2)/n}\sum_{\tau_++\tau_-=n-2}\binom{n-2}{\tau_+,\tau_-}\left(2iQ_{++-}\sin[\Lambda(\tau_+-\tau_-)]\right)^2 2^{-(n-2)}, \quad \text{(F.19)}$$

$$S_1 = \frac{4}{n^2}\binom{n}{1}e^{-4\gamma^2(n-1)/n}\sum_{\tau_++\tau_-=n-1}\binom{n-1}{\tau_+,\tau_-}\left(2iQ_{++-}\sin[\Lambda(\tau_+-\tau_-)]\right)2^{-(n-1)}(\tau_+-\tau_-)\cos 2\beta,$$

$$\text{(F.20)}$$

$$S_0 = \frac{1}{n}. \quad \text{(F.21)}$$

Finally, since, $\sin x = (e^{ix}-e^{-ix})/(2i)$, we have the following identities:

$$\sum_{r+s=m}\binom{m}{r,s}\sin^2[\Lambda(r-s)] = 2^{m-1}[1-\cos^m 2\Lambda], \quad \text{(F.22)}$$

$$\sum_{r+s=m}\binom{m}{r,s}\sin[\Lambda(r-s)](r-s) = 2^m m \sin\Lambda\cos^{m-1}\Lambda. \quad \text{(F.23)}$$

Thus, plugging in $\Lambda = 4\lambda\gamma/n$ and using the fact that $2iQ_{++-} = \frac{1}{2}\sin 2\beta$, we arrive at

$$\begin{aligned}
\mathbb{E}_{\boldsymbol{Y}}[\langle\mathcal{R}_{\mathrm{QAOA}}^2\rangle_{\gamma,\beta}] &= \frac{n-1}{2n}e^{-8\gamma^2(n-2)/n}\sin^2(2\beta)[1-\cos^{n-2}(8\lambda\gamma/n)] \\
&\quad + \frac{n-1}{n}e^{-4\gamma^2(n-1)/n}\sin(4\beta)\sin(4\lambda\gamma/n)\cos^{n-2}(4\lambda\gamma/n)+\frac{1}{n}.
\end{aligned} \quad \text{(F.24)}$$

## G   Additional numerical simulations

In this appendix, complementing the simulation results in Section 4, additional numerical simulations are performed for $1 \le p \le 7$ at $q=2$ and $1 \le p \le 6$ at $q=3$.

Fig. 4 displays the second moment of QAOA overlap versus problem dimension $n$. The y-axis plots the simulated second moment subtracting the theoretical value in the $n \to \infty$ limit. For all demonstrated $(p,q)$ pairs, the simulation appears to converge to the theoretical value with order $1/n$ deviations. This is consistent with the rigorous finite-$n$ formula for $(p,q) = (1,2)$ in Eq. (F.24).

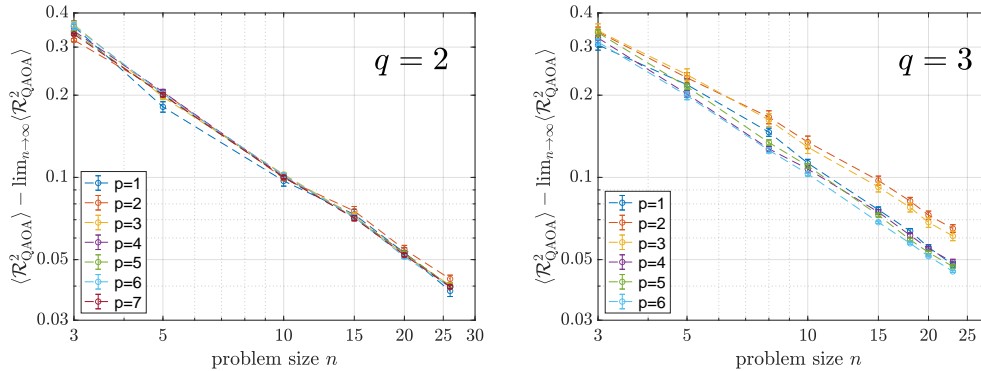

Figure 4: Log-log plots of the difference between observed overlap (averaged over instances and quantum measurements) at various problem dimension $n$ and the predicted value from the sine-Gaussian law in the $n \to \infty$ limit. Different colored lines correspond to different QAOA depth $p$, with parameters $(\boldsymbol{\gamma},\boldsymbol{\beta})$ set to be the same as in Table 1. We choose $\Lambda = 0.2$, $\lambda_n = \Lambda n^{1/(2p)}$ (left), and $\lambda_n = \Lambda n^{[1+1/(2^p-1)]/2}$ (right). Error bars are standard errors of the mean.

Our numerical simulations use the `GenQAOA` library, available at `https://github.com/leologist/GenQAOA`. The simulations are conducted on a laptop (MacBook Pro M2), where the simulation of each 26-qubit instance with $p$-step QAOA for $1 \le p \le 7$ took about 160 seconds. Data used in the figures are available upon request.

# H Analysis of classical power iteration algorithm

## H.1 Proof of Proposition 3.3

Define $\Lambda_n = \lambda_n / n^{(q-1)/2}$, we have

$$\boldsymbol{Y}[\hat{\boldsymbol{u}}_0^{\otimes(q-1)}] = \Lambda_n \langle \hat{\boldsymbol{u}}_0, \boldsymbol{u} \rangle^{q-1} \frac{\boldsymbol{u}}{\sqrt{n}} + \frac{1}{\sqrt{n}} \boldsymbol{W}[\hat{\boldsymbol{u}}_0^{q-1}] \equiv \Lambda_n G_n^{q-1} \frac{\boldsymbol{u}}{\sqrt{n}} + \frac{1}{\sqrt{n}} \boldsymbol{h},$$

where we define $G_n = \langle \hat{\boldsymbol{u}}_0, \boldsymbol{u} \rangle$, and $\boldsymbol{h} = \boldsymbol{W}[\hat{\boldsymbol{u}}_0^{q-1}]$. Then marginally over $\boldsymbol{W}$ and $\hat{\boldsymbol{u}}_0$, we have $G_n$ is independent of $\boldsymbol{h}$, and $G_n$ converges in distribution to a Gaussian random variable $G \sim \mathcal{N}(0,1)$, $\boldsymbol{h} \sim \mathcal{N}(0, I_n)$. As a consequence, we have

$$(G_n, \|\boldsymbol{Y}[\hat{\boldsymbol{u}}_0^{\otimes(q-1)}]\|_2) \xrightarrow{p} (G, \sqrt{1 + \Lambda^2 G^{2q-2}}), \quad n \to \infty.$$

This gives

$$\langle \boldsymbol{u}, \hat{\boldsymbol{u}}_1 \rangle / n = \frac{\Lambda_n G_n^{q-1} + \langle \boldsymbol{u}, \boldsymbol{h} \rangle / n}{\|\boldsymbol{Y}[\hat{\boldsymbol{u}}_0^{\otimes(q-1)}]\|_2} \xrightarrow{p} \frac{\Lambda G^{q-1}}{\sqrt{1 + \Lambda^2 G^{2q-2}}}, \quad n \to \infty.$$

This proves the Proposition 3.3.

## H.2 Proof of Proposition E.2

In this proof, we denote in short $\hat{\boldsymbol{u}}_k = \hat{\boldsymbol{u}}_{k,\text{biased}}$. Define $\Lambda_n = \lambda_n / n^{(1-c)(q-1)}$, we have

$$\boldsymbol{Y}[\hat{\boldsymbol{u}}_0^{\otimes(q-1)}] = \Lambda_n [n^{(1/2)-c} \langle \hat{\boldsymbol{u}}_0, \boldsymbol{u} \rangle]^{q-1} \frac{\boldsymbol{u}}{\sqrt{n}} + \frac{1}{\sqrt{n}} \boldsymbol{W}[\hat{\boldsymbol{u}}_0^{q-1}] \equiv \Lambda_n U_n^{q-1} \frac{\boldsymbol{u}}{\sqrt{n}} + \frac{1}{\sqrt{n}} \boldsymbol{h},$$

where we define $U_n = n^{1/2-c} \langle \hat{\boldsymbol{u}}_0, \boldsymbol{u} \rangle$, and $\boldsymbol{h} = \boldsymbol{W}[\hat{\boldsymbol{u}}_0^{q-1}]$. Then marginally over $\boldsymbol{W}$ and $\hat{\boldsymbol{u}}_0$, we have $U_n \to \sin(2\delta)$, and $\boldsymbol{h} \sim \mathcal{N}(0, I_n)$. As a consequence, we have

$$(U_n, \|\boldsymbol{Y}[\hat{\boldsymbol{u}}_0^{\otimes(q-1)}]\|_2) \xrightarrow{p} (\sin(2\delta), \sqrt{1 + \Lambda^2 \sin(2\delta)^{2q-2}}), \quad n \to \infty.$$

This gives

$$\langle \boldsymbol{u}_1, \hat{\boldsymbol{u}} \rangle / n = \frac{\Lambda_n U_n^{q-1} + \langle \boldsymbol{u}, \boldsymbol{h} \rangle / n}{\|\boldsymbol{Y}[\hat{\boldsymbol{u}}_0^{\otimes(q-1)}]\|_2} \xrightarrow{p} \frac{\Lambda \sin(2\delta)^{q-1}}{\sqrt{1 + \Lambda^2 \sin(2\delta)^{2q-2}}}, \quad n \to \infty.$$

This proves the Proposition E.2.

## H.3 Proof of Proposition 3.9

We prove this proposition using results in [33]. The notations in [33] are slightly different from the notations in this paper, and in the following, we will adopt the notations in the former.

Suppose we observe the spiked tensor model

$$\boldsymbol{T} = \bar{\lambda}_n \boldsymbol{v}^{\otimes q} + \boldsymbol{W}, \tag{H.1}$$

where $\boldsymbol{v} \sim \text{Unif}(\{\pm 1/\sqrt{n}\}^n)$ and each element of $\boldsymbol{W}$ is iid Gaussian. Note that the $\bar{\lambda}_n$ in Eq. (H.1) is different from the $\lambda_n$ in Eq. (1.1). We should take $\bar{\lambda}_n = \sqrt{n} \lambda_n$ so that $\bar{\lambda}_n / n^{(q-1+\varepsilon_p)/2} \to \Lambda$.

Consider the tensor power iteration algorithm with initialization $\boldsymbol{v}^0 = \tilde{\boldsymbol{v}}^0 \sim \text{Unif}(\mathbb{S}^{n-1})$, and

$$\boldsymbol{v}^{t+1} = \boldsymbol{T}[(\tilde{\boldsymbol{v}}^t)^{\otimes(q-1)}] = \bar{\lambda}_n \langle \boldsymbol{v}, \tilde{\boldsymbol{v}}^t \rangle^{q-1} \boldsymbol{v} + \boldsymbol{W}[(\tilde{\boldsymbol{v}}^t)^{\otimes(q-1)}], \quad \tilde{\boldsymbol{v}}^{t+1} = \boldsymbol{v}^{t+1} / \|\boldsymbol{v}^{t+1}\|_2. \tag{H.2}$$

We let $\alpha_t := \bar{\lambda}_n \langle \boldsymbol{v}, \tilde{\boldsymbol{v}}^{t-1} \rangle^{q-1}$. Then [33] shows the following lemma.

**Lemma H.1** (Lemma 3.2 of [33]). *Consider the spiked tensor model as in Eq. (H.1) and consider the tensor power iteration (H.2). For any fixed $\varepsilon \in (1/4, 1/2)$, define the stopping time*

$$T_\varepsilon := \min \{ t \in \mathbb{N}_+ : |\alpha_t| \geq n^\varepsilon \}. \tag{H.3}$$

*Then, there exists an absolute constant $C > 0$, such that with probability no less than $1 - \exp(-C\sqrt{n})$, the following happens: For all $t < \min(T_\varepsilon, n^{1/2(q-1)})$, we have*

$$\alpha_{t+1} \overset{d}{=} (\bar{\lambda}_n n^{-(q-1)/2}) \zeta_t (\alpha_t + b_t + c_t Z_t)^{q-1}, \qquad \alpha_0 = 0, \tag{H.4}$$

*where $Z_t \sim \mathcal{N}(0,1)$ is independent of $(\zeta_t, \alpha_t, b_t, c_t)$,*

$$\zeta_t \in [1 - n^{-1/6}, 1 + n^{-1/6}], \quad |b_t| \leq C n^{1/4 + (q-1)(\varepsilon - 1/2)}, \quad |c_t - 1| \leq C n^{2(q-1)(\varepsilon - 1/2)}, \tag{H.5}$$

We take $\varepsilon \in ([(q-1)^{p-1} - 1]/[2(q-1)^p - 2], 1/2)$ to be fixed. By Lemma H.1, for a fixed $p \in \mathbb{N}_+$, with high probability, we have $T_\varepsilon \geq p$, as well as the upper bounds indicated in Eq. (H.5) for all $t \leq p - 1$. Applying Eq. (H.4) recursively implies that

$$\alpha_p \overset{d}{=} (1 + o_{\mathbb{P}}(1)) \cdot n^{1/2} \Lambda^{1/\varepsilon_p} G^{(q-1)^{p-1}}, \quad G \sim \mathcal{N}(0, 1).$$

By the last equation on page 9 of [33], we see that (for $\boldsymbol{H}_p = \{0, 1, 2, \ldots, p\}^{q-1}$)

$$\boldsymbol{v}^p = \alpha_p \boldsymbol{v} + \sum_{(i_1, i_2, \cdots, i_{q-1}) \in \boldsymbol{H}_{p-1}} \beta^{(p-1)}_{i_1, i_2, \cdots, i_{q-1}} \boldsymbol{w}_{i_1, i_2, \cdots, i_{q-1}},$$

where $\boldsymbol{w}_{i_1, i_2, \cdots, i_{q-1}} \sim_{iid} \mathcal{N}(0, I_n)$, and $\sum_{(i_1, i_2, \cdots, i_{q-1}) \in \boldsymbol{H}_{p-1}} |\beta^{(p-1)}_{i_1, i_2, \cdots, i_{q-1}}|^2 = 1$. Invoking the uniform law of large numbers, we are able to conclude that $\mathcal{R}_{\mathrm{PI}} \overset{d}{=} \langle \boldsymbol{v}^p, \boldsymbol{v} \rangle / \|\boldsymbol{v}^p\|_2 \overset{d}{\to}$ $\sin[\arctan(\Lambda^{1/\varepsilon_p} G^{(q-1)^p})]$. This concludes the proof of Proposition 3.9.

