# OpenReview forum: "Statistical Estimation in the Spiked Tensor Model via the Quantum Approximate Optimization Algorithm"
_NeurIPS.cc/2024/Conference — NeurIPS 2024 spotlight_

### Official Review · Reviewer_R5pw · 2024-07-10

**Soundness:** 4
**Presentation:** 4
**Contribution:** 3
**Rating:** 7
**Confidence:** 3

**Summary:**

The paper studies the performance of the Quantum Approximate Optimization Algorithm (QAOA) for a classical average case problem from high dimensional statistic: tensor principal component analysis (tPCA), which exhibits a computational-statistical gap. The paper investigates if this algorithm can achieve a quantum advantage over its classical counterparts. The paper makes progress towards this question and suggests that the answer to expect is somewhat negative (only for QAOA). The main results are
1. After 1 step of QAOA, it archives weak recovery at the same SNR threshold (up to constants) as achieved by 1 step of clssical tensor power iteration.
2. Using heuristic calculations (but not rigorously), they further showed that even after (some constant) $p$ steps of QAOA, the estimator succeeds at weak recovery at the same SNR as the tensor power iteration. This further suggests that even after using tensor unfolding, QAOA won't be able to surpass the computational threshold for this problem.
3. Along the way, they observe a sine Gaussian law for the asymptotic distribution of the overlap between the estimator (after $p$ steps of QAOA) and the ground truth. Again, this is proven for $p=1$ steps but empirically verified for $p>1$.

Background: The problem is known to have a computational statistical gap. In particular, in the parameterization (1.1) taken in this paper, (a) recovery is possible whenever the SNR $\lambda \gg 1$, (b) but the threshold for efficient algorithms is known to be $\lambda \approx n^{(q-2)/4}$. Several classical algorithms including tensor unfolding, sum-of-squares, or gradient descent with landscape smoothing are known to achieve this. For iterative algorithms, such as the tensor power iteration and vanilla GD, the threshold is further away, requiring $\lambda \approx n^{(q-2)/2}$, and thus, to achieve the computational threshold either tensor unfolding (for power iteration) or landscape smoothing (for GD) is required.

**Strengths:**

1. The paper analyzes one of the important quantum algorithms, for an important problem in high dimensional statistics to seek to answer if there is a quantum advantage. The results in the paper are suggestive that the answer to expect is negative.
2. The paper is well-written, and for heuristic claims, provided clean numerical simulations.

**Weaknesses:**

I do not see any major weaknesses in the paper. Only a small quibble is a place in the introduction in lines 39-40 (and also in the abstract lines 1-3), where the authors present the motivation as seeking whether QAOA has superpolynomial speedup from clssical algorithms. However, I found this motivation slightly hand-wavy. I could not find enough concrete justification for why looking at (the combination of) QAOA for tensor PCA is a promising avenue for demonstrating this. If authors can make this more concrete, that would be helpful.

On the other hand, just studying the performance of QAOA for tensor PCA is an important question in its own right, as very well justified in lines 40-51. The authors do make good progress towards this.

**Questions:**

1. Could the author elaborate on how to combine QAOA with tensor unfolding? In more detail, why do the results after $p$ steps of unfolded tensors suggest that we could not surpass the computational threshold after using QAOA on unfolded almost square matrix?
2. In a non-quantum setup, a more standard is to take the prior to be uniform over a sphere. Can authors describe why it was important to take it to be uniform over the hypercube (in slightly more detail than in footnote 1)? I am just trying to understand the difficulty in the analysis if the prior was uniform over a sphere.

---

> ### Author Rebuttal · Authors · 2024-08-07
>
> We thank the reviewer for their positive comments on the importance of the algorithm and the problem under study. To make the motivation more concrete: We believe that tensor PCA is a promising avenue for demonstrating quantum advantage because the computational-statistical gap in the spiked tensor problem is huge compared to other problems. The gap here is a polynomial factor separating the required $\Omega(n^{(q-2)/4})$ SNR for the best known algorithm vs. the information-theoretically sufficient $O(1)$ SNR; compare this to a constant factor gap in many other cases such as spin-glass optimization [45]. Hence, it is reasonable to conjecture that a better (quantum) algorithm exists for this problem. Our choice of focus on the QAOA is because it is a novel quantum algorithmic primitive that is both realistic for near-term implementations and guaranteed to succeed when the algorithmic depth p grows unboundedly with problem size. We concur with the reviewer that the question of studying QAOA on tensor PCA is an important question on its own, and we will augment the justifications in the main text with the above discussion.
>
> To answer the reviewer’s questions:
> 1) The idea about combining QAOA with tensor unfolding is explained in Remark 3.11, but we will elaborate further here. Essentially, upon an input spiked tensor, we partition the tensor indices into two groups to obtain a 2-tensor which is a spiked matrix that we call the unfolded tensor. We can apply the QAOA to the resultant spiked matrix, with a rescaled SNR $\bar{\lambda}_n = \lambda_n/n^{(q-2)/4}$. Additionally, our Claim 3.7 implies that as $p\to\infty$ (after taking $n\to\infty$), the QAOA can solve the spiked matrix problem with $\bar{\lambda}_n \approx 1$. This implies that the QAOA with tensor unfolding can solve the original spiked tensor problem with $\lambda_n = \Theta(n^{(q-2)/4})$ which achieves but does not surpass the classical computational threshold. Our result does not rule out the possibility that the QAOA can surpass the computational threshold when p grows unboundedly with n, or potentially in combination with a more clever trick.
>
> 2) The issue here is that the QAOA can only output bit-strings, which live on the hypercube. In practice, when the hidden prior is on the sphere, we can still run the QAOA to get an estimator on the hypercube, but this may not be very natural. Our theoretical framework can be certainly used to analyze the overlap between the hypercube estimator and the spherical prior, and we believe the result will be similar. Another potential solution for a spherical prior is to develop a variant of the QAOA for continuous variables, but this is outside the scope of our work.

---

> > ### Comment · Reviewer_R5pw · 2024-08-09
> >
> > Thanks for answering my questions! I would be happy to see this paper appear at the conference!

---

### Official Review · Reviewer_nUte · 2024-07-13

**Soundness:** 3
**Presentation:** 3
**Contribution:** 3
**Rating:** 7
**Confidence:** 3

**Summary:**

The paper investigates the performance of the Quantum Approximate Optimization Algorithm (QAOA) on the spiked tensor model problem. The authors demonstrate that QAOA's weak recovery threshold aligns with that of tensor power iteration and show through heuristic calculations that multi-step QAOA could potentially match but not exceed the classical computation threshold. A notable finding is the sine-Gaussian law for the asymptotic overlap distribution of p-step QAOA verified by simulations, which is distinct from classical methods and suggests a modest quantum advantage. The paper employs novel techniques, including Fourier transforms, to analyze the QAOA's performance and concludes with implications for potential quantum advantage in statistical inference problems.

**Strengths:**

- The paper prove that the weak recovery threshold of 1-step QAOA matches that of 1-step tensor power iteration, which is a new theoretical result for analyzing QAOA.

- The paper uses heuristic calculations to characterize the asymptotic overlap distribution of p-step QAOA, showing that the ability is similar to the multi-step tensor power iterations.

- Their proof techniques includes Fourier transform to handle exponential sums, which may be novel in the analysis of QAOA algorithms.

**Weaknesses:**

- The results indicate that constant-step QAOA does not improve the recovery threshold beyond what is achievable by classical tensor power iteration by more than a constant factor, suggesting that the quantum advantage is modest.

- The paper does not address the performance of QAOA with more circuit depths, which is an open question and could be crucial for demonstrating a strong quantum advantage.

**Questions:**

- Could the authors explain more about the generality of their proof techniques, i.e., could their proof techniques  be used in analysis for QAOA algorithms in other problem settings?

**Limitations:**

- The analysis for p-step QAOA (where p > 1) relies on heuristic arguments from physics, which may not be as rigorous as desired.

---

> ### Author Rebuttal · Authors · 2024-08-07
>
> We thank the reviewer for the positive comments on the novelty of our results. Regarding the weaknesses mentioned, we acknowledge that the quantum advantage is modest. Before our work, the extent of quantum advantage that the QAOA could provide on the spiked tensor problem was unknown, especially since previous hardness results did not apply in this setting. Therefore, understanding the power and limitations of the QAOA for this problem is an important step forward. While our analysis is limited to depths p that do not grow with n, this regime is also the most realistic for near-term quantum computers. Nevertheless, we agree that analyzing this algorithm at super-constant depths is an important open question worthy of future work.
>
> In terms of generalizing our techniques to more problems, we believe our methods can be also applied to study the performance of the QAOA on other problems such as planted clique, sparse PCA, stochastic block model and Bayesian linear models, to name a few. Furthermore, by setting $\lambda=0$, our formula recovers previous results on the QAOA applied to spin-glass models (which can be seen in the derivations in Appendix D.4). This demonstrates the broader applicability of our methods beyond the specific problem studied in this paper.
>
> To address the limitation regarding heuristics arguments, we refer the reviewer to our global rebuttal where we discuss the potential of making our derivation more rigorous. There, we also discuss additional evidence for the correctness of the heuristics (that is, the use of Dirac delta functions and interchanging order of limits), which are further corroborated by our numerical experiments.

---

> > ### Comment · Reviewer_nUte · 2024-08-10
> >
> > Thanks for the detailed response! The authors have addressed my questions satisfactorily, so I adjusted the score accordingly.

---

### Official Review · Reviewer_a21g · 2024-07-13

**Soundness:** 3
**Presentation:** 3
**Contribution:** 3
**Rating:** 7
**Confidence:** 4

**Summary:**

The quantum approximate optimization algorithm is analyzed for the spiked tensor model. Weak recovery
of 1-step QAOA is rigorously shown to matche that of 1-step tensor power iteration. Heuristic calculations for p-step QAOA
matche that of p-step tensor power iteration.

**Strengths:**

There have been many works on tensor revorery fro such statistical models within classical inference. A very small number of works have attacked the quantum algorithmic aspect. This paper is therefore very welcome. The results are interesting and clearly expalined.
Weak recovery of 1-step QAOA is rigorously shown to matche that of 1-step tensor power iteration. Heuristic calculations for p-step QAOA
matche that of p-step tensor power iteration. the paper argues that
that multi-step QAOA with tensor unfolding could achieve,
the asymptotic classical computation threshold of spiked q-tensors.
The asymptotic overlap distribution for p-step QAOA is characterized and
some sort of sine-Gaussian law is observed  (through simulations).

**Weaknesses:**

For p-step QAOA the analysis is not rigorous. The observation of the intriguing sine-Gaussian law is numerical. Further analysis will be needed.

**Questions:**

Maybe authors could discuss the limitations of implementing such algorithms on NISQ devices ? Any realistic prospects ?

**Limitations:**

Maybe authors could discuss the limitations of implementing such algorithms on NISQ devices ? Any realistic prospects ?

---

> ### Author Rebuttal · Authors · 2024-08-07
>
> We thank the reviewer for the positive comments on the importance of understanding the power of quantum algorithms for statistical inference problems. Regarding the weakness mentioned, we remark that our analysis and derivation of the sine-Gaussian law is completely rigorous at p=1. Additionally, for the general-p analysis, we have strong evidence supporting the correctness of our approach, not only from numerical simulation but also from the fact that our framework reproduces known rigorous results in different limits. We explain this in more detail in our global rebuttal.
>
> Addressing the question on implementation, we remark that the QAOA is quite NISQ-friendly and has already been implemented in practice for various problems, as demonstrated for example in Refs. [16-18]. However, some limitations of noisy implementations of QAOA are known, particularly when the problem topology does not match the hardware connectivity (see e.g., [arXiv:2009.05532]). These limitations can be mitigated to some extent with simple forms of error-correction. Nevertheless, our paper is focused on the theoretical analysis of the performance and limitations of the noiseless QAOA. A detailed discussion about potential implementations on near-term quantum devices is beyond the scope of this work.

---

> > ### Comment · Reviewer_a21g · 2024-08-09
> >
> > The authors have answered my question satisfactorily. I suggest adding a few pointers to the implementations of QAOA on curent devices to guide potentially interested readers. I understand this aspect in beyond the scope of the paper but if space allows a few pointers and comments woulb be welcome.

---

> > > ### Author Response · Authors · 2024-08-13
> > >
> > > We thank Reviewer a21g for their positive feedback and suggestion. In line with their recommendation, we will add more discussion about implementations of the QAOA in our revision.

---

### Official Review · Reviewer_9dK4 · 2024-07-23

**Soundness:** 3
**Presentation:** 4
**Contribution:** 3
**Rating:** 8
**Confidence:** 3

**Summary:**

This submission proposed to use quantum approximate optimization algorithm (QAOA) to compute the maximum likelihood estimator in the statistical estimation problem of the spiked tensor model.

Using the overlap between the estimated vector and the original vector, the author(s) obtained rigorous analysis for the so-called weak recovery threshold for 1-step QAOA. Namely, above such a threshold, the overlap will be non-zero with non-trivial probability; otherwise, the overlap will vanish with high probability. The author(s) also showed that the established weak recovery threshold matches the 1-step tensor power iteration classical algorithm.

For the p-step QAOA, the author(s) also obtained the weak recovery threshold based on some heuristic argument, which also matches that of the p-step tensor power iteration algorithm. Numerical experiments show that the QAOA method could achieve the state of the art threshold by combining with tensor unfolding (a technique used in the state of the art algorithm).

As far as I know, the submission proposed a new technical analysis for QAOA (by showing that the overlap exhibits an asymptotic sine-Gaussian distribution), providing a rigorous study of the polynomial-time QAOA.
As such, I believe the work is worth sharing to the community. Hence, I recommend accept.

(I did not have enough time to check all the derivations provided in appendices in details, but the overall proof ideal seems logical to me.)

**Strengths:**

1. Rigorous analysis for the 1-step QAOA, rigorous analysis for the p-step tensor power iteration, and detailed comparison between the two algorithms.

2. Discovering an intriguing sine-Gaussian law also verified through numerical simulations.

3. The manuscript is well written and explained.

**Weaknesses:**

1. The asymptotic analysis requires the number of qubits n approaching infinity, which is practically demanding.

2. The analysis of the p-step QAOA is based on an heuristic calculation.

3. It is not known if QAOA could achieve the same threshold as in the state of the art classical algorithm.
Numerical experiments do provide some potentials of matching the result. Whether there is a quantum advantage remains unclear.

**Questions:**

1. In Remark 3.10, the author(s) claimed that for certain parameter p, the QAOA has a constant factor advantage over the classical power iteration algorithm in the overlap achieved.
The overlap via the p-step tensor is proved in Proposition 3.9. However, the overlap via QAOA given in (3.12) is based on heuristic calculations.
Hence, I'm not sure such an advantage is rigorous. If I did not misunderstand anything, the wording of claiming the constant factor advantage needs to be modified.

2. In Discussion, the author(s) asserted that "This implies that achieving a strong quantum advantage via the QAOA requires using a number of steps p that grows with n."
I understand that multiple (possibly infinite many) steps is needed to achieve a better performance for QAOA.
However, it is still not analytically evident if QAOA can match the state of the art threshold, because a heuristic calculation is required in the analysis.
Hence, whether there is a rigorous advantage for QAOA still remains unclear to me, even p approaching infinity.

3. As above, even if author(s) can rigorously show that p-step QAOA outperforms the p-step tensor power iteration, it probably not accurate to call it a "modest quantum advantage", since the p-step tensor power iteration is not the state of the art algorithm.
I highly recommend the author(s) to be more careful about the phrasing of quantum advantage throughout the paper.

**Limitations:**

The limitations are addressed in the manuscript. I also summarized them in Weaknesses.
Yet, since this submission is a theory work, I think the practical limitation is not a big concern.

---

> ### Author Rebuttal · Authors · 2024-08-07
>
> We thank the reviewer for a positive assessment of our paper. We refer the reviewer to the global rebuttal for our response regarding the weakness of using heuristic calculation. Moreover, our asymptotic results in the $n\to\infty$ limit also show good agreement with numerical experiments at finite $n$. In particular, the average squared overlap with the signal at finite $n$ appears to converge to our infinite-n prediction with order $1/n$ deviations as shown in Figure 2 and 4.
>
> We now address the reviewer’s questions one-by-one:
>
> 1) We stress that the constant factor advantage is in fact rigorous for $p=1$ and $q\ge3$, as seen in the first row of Table 1. However, we acknowledge that the constant factor advantage seen for $p>1$ is currently not fully rigorous, and will revise our manuscript to say “conjectured advantage” in those cases.
>
> 2) We recognize that our results currently do not rigorously show an advantage for the QAOA against the state-of-the-art classical threshold. Although it is rigorously known that the QAOA can compute the MLE and (weakly) recover the signal when the depth p grows unboundedly with n, we do not rigorously know if it can do so in polynomial depths even with SNR at the classical threshold. One possible way to address this is extending previous computational universality results in Ref. [19] to show the QAOA can reproduce classical algorithms for spiked tensors within polynomial depths. Nevertheless, the scope of our present work is focused on analyzing the constant depth regime of QAOA in hopes of obtaining good performance with as little quantum resource as possible.
>
> 3) We acknowledge that the phrasing of “modest quantum advantage” may be misleading. We will revise our manuscript to more clearly indicate that this advantage is only over constant-step power iteration, which is not the best classical algorithm.

---

> > ### Comment · Reviewer_9dK4 · 2024-08-09
> >
> > The authors have addressed my questions. Once the manuscript is revised accordingly, I can support this submission.

---

> > > ### Author Response · Authors · 2024-08-13
> > >
> > > We are grateful to Reviewer 9dK4 for their support. We will revise our manuscript according to their suggestions in the camera-ready version if this submission is accepted.

---

### Official Review · Reviewer_TiV7 · 2024-07-31

**Soundness:** 3
**Presentation:** 4
**Contribution:** 3
**Rating:** 7
**Confidence:** 3

**Summary:**

This paper studies the performance of 1-step and multi-step quantum approximate optimization algorithm (QAOA) for spiked tensor problem. In this problem one observes a q-dimentional tensor which is a properly normalized linear combination of q-th tensor power of unknown vector $u \in \{+1, -1\}^n$ and Gaussian noise $W$: $\lambda u^{\otimes q} / n^{q/2} + 1/sqrt(n)\cdot W$.  The goal is to recover the unknown vector $u$ from the observed tensor. This problem is known to be statistically solvable for $\lambda > T$, for some absolute constant T; however, it is known that under common complexity assumption, the problem has polynomial time classical algorithm only for $\lambda = \Omega(n^{(q-2)/4})$, providing a complexity gap.

This paper studies whether a specific family of quantum algorithms, called QAOA, can achieve a quantum advantage compared to classical algorithm. The paper proved negative result showing that constant-step QAOA applied to weak recovery problem in the spiked tensor model only achieves non-trivial overlap with the signal u when $\lambda = \Omega(n^{(q-1)/2})$ which nearly matches the threshold for the tensor power classical algorithm.

**Strengths:**

The spiked tensor problem is a well-studied problem with important application, hence, understanding the performance of quantum algorithms applied to this problem is an interesting problem. To the best of my knowledge, this is the first paper that studies the performance of QAOA applied to this problem, showing that this family of quantum algorithms does not achieve an advantage over classical algorithms.

The proofs are quite technically involved and combine techniques such as the discrete Fourier transforms and the central limit theorem to handle combinatorial summations. I have not checked the proofs carefully, but skimming through them they look sound.

The authors provide a good overview of the prior work and clearly compare the current paper to prior results.

**Weaknesses:**

I think that the contribution of this paper is somewhat limited in the sense that negative result is obtained only for a very particular family of quantum algorithms, which does not rule out that even small modifications can achieve quantum advantage for this problem. This is in contrast to classical results where gap is established for any classical algorithm under some standard complexity assumptions.

**Questions:**

1) Do authors expect that under some standard assumptions, any BQP algorithm is not able to recover $u$ for $\lambda =o(n^{(q-2)/4})$?
2) Do you see other candidate problems where techniques developed in this paper can potentially be used to study the performance of QAOA?

---

> ### Author Rebuttal · Authors · 2024-08-07
>
> We thank the reviewer for the positive assessment of our paper and agree on the importance of studying quantum algorithms applied to the spiked tensor.
>
> Regarding the mentioned weakness, we would like to point out that the various celebrated classical hardness results for the spiked tensor problem fall into two categories: (i) hardness for all polynomial-time classical algorithms under certain complexity-theoretical assumptions, as in Ref. [13]; or (ii) unconditional hardness results for specific families of classical algorithms, such as tensor power iteration, gradient descent, Langevin dynamics, spectral methods, and message passing algorithms (see Ref. [2-12]). Our work extends the second type of hardness results to the quantum realm by showing an unconditional limitation of a popular family of quantum algorithms on this problem.
>
> To address the reviewer’s questions:
>
> 1) We recognize that our hardness result applies only to a specific family of quantum algorithms, but it does not rely on any assumptions. Although there is a conditional hardness result [13] that rules out all efficient classical algorithms by assuming a generalized version of the planted clique conjecture, it is unclear if this conjecture can be considered as a standard complexity assumption, especially in the quantum setting. While our result is suggestive that weak recovery for $\lambda=o(n^{(q-2)/4})$ is likely difficult for broader classes of quantum algorithms, such as low-depth quantum circuits, we do not have sufficient evidence to rule out all BQP algorithms.
>
> 2) We believe our techniques can be also applied to study the performance of the QAOA on other problems such as planted clique, sparse PCA, stochastic block models, and Bayesian linear models, among others. Furthermore, by setting $\lambda=0$, we recover previous results on the QAOA applied to spin-glass models (which is implicitly discussed in Appendix D.4). Therefore, we believe that our techniques have broader applicability beyond this paper, allowing one to study the QAOA on other problems, as well as other QAOA-like variational quantum algorithms.

---

> > ### Comment · Area_Chair_T7vh · 2024-08-13
> >
> > Dear Reviewer TiV7,
> >
> > The author-reviewer discussion period is ending soon. Please check if the authors' response has addressed your concerns and feel free to adjust your score. If the authors' response is not satisfactory to you, please explain your reason and discuss with the authors *immediately*.
> >
> > Best regards,
> > AC

---

> > ### Comment · Reviewer_TiV7 · 2024-08-13
> >
> > I thank the authors for their response, and I adjusted my score to 7.

---

### Author Rebuttal · Authors · 2024-08-07

In this global rebuttal, we address the concern raised by Reviewers 2, 3, and 4 about our use of physics-style heuristics and the rigor of theoretical results. We emphasize that our result at depth $p=1$ is fully rigorous. While the analysis for $p>1$ is not fully rigorous due to the use of Dirac delta functions and an unjustified change in order of limits, we expect that it could be made rigorous with more advanced harmonic analysis techniques in dealing with Dirac delta functions. For example, Ref. [25] adopted a heuristic approach for analyzing the QAOA performance of another problem, and showed that changing the order of limits can be justified with a series of bound estimates and the dominated convergence theorem.

In any case, we believe there is strong evidence for the correctness of our general $p$-step QAOA analysis because: (1) after setting $p=1$, our general-$p$ framework yields a result identical to our rigorous result at $p=1$ obtained with a different method, and (2) after setting $\lambda=0$, our result agrees with the prior rigorous result for the QAOA’s performance on spin glass models in Ref. [25]. The latter agreement is implicit in the discussion of Appendix D.4. Furthermore, our heuristic derivation can be viewed as an approach to obtain the correct result that is simpler than the rigorous method in Ref. [25], and may be applied to more general problems.

---

### Decision · Program_Chairs · 2024-09-25

**Decision:**

Accept (spotlight)

**Comment:**

Consider statistical estimation in the spiked tensor model. The *weak recovery threshold* refers to the signal-to-noise ratio (SNR) above which weak recovery—an estimator that achieves non-trivial overlap with the signal—is possible. The paper proposes computing the maximum-likelihood estimate via the quantum approximate optimization algorithm (QAOA) and analyzes the weak recovery threshold of $p$-step QAOA.

- It is rigorously proved that 1-step QAOA achieves the same weak recovery threshold as 1-step tensor power iteration.
- It is proved that, if a heuristically proved claim regarding the sine-Gaussian law of the overlap holds, then $p$-step QAOA can be a constant factor better than that of $p$-step tensor power iteration.

Although the results of this paper do not surpass state-of-the-art classical algorithms and are not fully rigorous, it marks the first theoretical work to study QAOA for statistical estimation in the spiked tensor model and provides useful insights into whether there is a quantum advantage for this task. Hence, I suggest accepting this paper as a spotlight.